# Crop rotation and native microbiome inoculation restore soil capacity to suppress a root disease

Yanyan Zhou [1,12], Zhen Yang [2,3,12], Jinguang Liu[2], Xudong Li[1], Xingxiang Wang [2,4], Chuanchao Dai [3], Taolin Zhang[2,4], Víctor J. Carrión [5,6,7,8], Zhong Wei [9], Fuliang Cao[10], Manuel Delgado-Baquerizo [11] & Xiaogang Li [1,2,10] ✉

It is widely known that some soils have strong levels of disease suppression and prevent the establishment of pathogens in the rhizosphere of plants. However, what soils are better suppressing disease, and how management can help us to boost disease suppression remain unclear. Here, we used field, greenhouse and laboratory experiments to investigate the effect of management (monocropping and rotation) on the capacity of rhizosphere microbiomes in suppressing peanut root rot disease. Compared with crop rotations, monocropping resulted in microbial assemblies that were less effective in suppressing root rot diseases. Further, the depletion of key rhizosphere taxa in monocropping, which were at a disadvantage in the competition for limited exudates resources, reduced capacity to protect plants against pathogen invasion. However, the supplementation of depleted strains restored rhizosphere resistance to pathogen. Taken together, our findings highlight the role of native soil microbes in fighting disease and supporting plant health, and indicate the potential of using microbial inocula to regenerate the natural capacity of soil to fight disease.

Some soils have a larger capacity to support disease suppression than others helping to prevent the establishment of pathogens in the rhizosphere of plants[1,2]. However, the characteristics defining these soils are largely unknown. The rhizosphere refers to the soil directly associated with the roots. This complex environment, enriched with carbon and nutrients, is home to a diverse microbial community that plays an essential role in promoting plant growth and health[3,4]. This rhizosphere microbiome serves as the first line of defense against plant pathogens, with direct consequences for plant disease outcomes[5–7]. Because of this, advancing our mechanistic understanding on how variations in the assembly of rhizosphere microbiomes influence plant disease is of paramount important to provide innovative strategies for improving plant health and productivity.

[1]State Key Laboratory of Tree Genetics and Breeding, College of Ecology and Environment, Nanjing Forestry University, Nanjing 210037, China. [2]State Key Laboratory of Soil & Sustainable Agriculture, Institute of Soil Science, Chinese Academy of Sciences, Nanjing 210008, China. [3]Jiangsu Key Laboratory for Microbes and Functional Genomics, College of Life Sciences, Nanjing Normal University, Nanjing 210023 Jiangsu, China. [4]Ecological Experimental Station of Red Soil, Chinese Academy of Sciences, Yingtan 335211, China. [5]Departamento de Microbiología, Facultad de Ciencias, Campus Universitario de Teatinos s/n, Universidad de Málaga, 29010 Málaga, Spain. [6]Instituto de Hortofruticultura Subtropical y Mediterránea La Mayora (IHSM) UMA-CSIC, 29010 Málaga, Spain. [7]Institute of Biology, Leiden University, Sylviusweg 72, 2333 BE Leiden, The Netherlands. [8]Department of Microbial Ecology, Netherlands Institute of Ecology (NIOO-KNAW), Droevendaalsesteeg 10, 6708 PB Wageningen, The Netherlands. [9]College of Resources and Environmental Science, Nanjing Agricultural University, Nanjing 210095, China. [10]Co-Innovation Center for Sustainable Forestry in Southern China, Nanjing Forestry University, Nanjing 210037, China. [11]Laboratorio de Biodiversidad y Funcionamiento Ecosistémico, Instituto de Recursos Naturales y Agrobiología de Sevilla (IRNAS), CSIC, Sevilla, Spain. [12]These authors contributed equally: Yanyan Zhou, Zhen Yang. ✉e-mail: xgli@njfu.edu.cn

Crop management is known to have a critical role in regulating rhizosphere microbiomes. In China, peanut represents one of the most profitable crops. However, the yield and quality of this product have been reported to be greatly compromised by soil-borne diseases, especially under intensive monocropping managements. In subtropical China, 10–40% of crop yield is lost due to increasing disease pressure, with peanut root rot caused by fungal pathogens being a major constraint[8]. Unfortunately, few effective control measures are available, and the use of chemical pesticides is limited by environmental concerns[9]. Crop rotations, including different crop varieties, have been proposed to mitigate the negative impacts of pathogens on crop production, by breaking the link between plant host and pathogens, becoming a non-expensive alternative for disease control. Yet, how management by regulating the impact of rhizosphere microbial communities on peanut root rot diseases remains largely unknown.

Microbial and plant-based tools, such as synthetic microbial communities (SynComs) and root-derived metabolites, are promising strategies to fight against soil-borne pathogens and plant disease, and can help to promote plant health. However, what strategies work best under contrasting management remains poorly understood. Recent studies have shown that rhizosphere microbiome assembly is significantly influenced by the selection effects of plant root metabolisms. As a result, different outcomes of plant health can be fostered depending on which microbial populations are able to take advantage of the root metabolites[10,11]. Thus, agricultural practices could potentially exploit the differences in root metabolisms of various crops to disrupt the directional selection for maintaining the stability of agricultural ecosystem[12,13]. Similarly, SynComs are currently being developed as bio-products to control soil-borne disease. Yet, there are needs for understanding the impacts of these SynComs on crop rhizosphere and as management tools for plant health.

Here, we conducted multiple experiments to unravel the influence of crop management on the capacity of the rhizosphere microbiome to support disease suppression. The findings revealed that under peanut monocropping, there was a selective impoverishment of key bacteria, with weaker responses to root exudates, but that were associated with the capacity of rhizosphere to suppress pathogen invasion. The restoration of these bacteria brought back the capacity of soils to fight disease suppression. Our study advances our knowledge on how to develop innovative strategies to provide favorable support for sustainable agriculture.

## Results

### Monocropping aggravates peanut root rot severity

To determine the effects of different agricultural regimes on peanut health, we conducted a field experiment from 2012 to 2016 aiming to quantify the occurrence of peanut (*Arachis hypogaea* L.) root rot under monocropping and rotation regimes (Fig. 1a). Results indicated that disease index (DI) rapidly increased from 2012 to 2016 in monocropping peanuts that were examined at flowering stage, compared with crops subjected to rotations (DI$_{2012}$ = 2.1, DI$_{2016}$ = 8.0, Fig. 2a, b). In fact, we showed that the DI of peanuts in rotation systems remained stable (DI$_{2012}$ = 1.6, DI$_{2016}$ = 3.1, Fig. 2b). We further showed that the DI of monocropping and rotation began to diverge in 2014, with a significantly higher DI in monocropping compared with crops subjected to rotation (Student's *t*-test, $t = 8.276$, $df = 16$, $P < 0.001$, Fig. 2b). These results indicated that long-term monocropping can aggravate peanut root rot disease.

Based on the difference of DI between monocropped and rotation peanuts at flowering stage, we specifically tracked the changes in DI at different peanut growth stages in 2018. At peanut seedling stage, the DI of peanuts grown on monocropped and rotation plots were very low (DI$_{rotation}$ = 3.87, DI$_{monocropping}$ = 4.38), and we did not find significant differences in DI for the two regimes at this particular stage (Student's *t*-test, $t = -0.660$, $df = 16$, $P = 0.519$, Fig. 2c). However, from

flowering stage, the DI in peanuts under monocropping regime dramatically increased, and was 2.6 times higher than that of peanuts sown under the rotation regime at pod-bearing stage (Student's *t*-test, $t = -7.901$, $df = 16$, $P < 0.001$, Fig. 2c), indicating the amplified severity of root rot during developmental stage.

To identify the pathogen associated with this root rot disease, we characterized the community composition of fungi in healthy and diseased peanut roots using Illumina sequencing (Fig. 1b). A total of 347,214 internal transcribed spacer 1 (ITS1) reads were obtained (range, 31,995–44,592 reads per sample), clustering into 183 fungal OTUs at ≥97% sequence identity. Several taxa from *Fusarium* sp. have been reported with ability to cause wilting symptoms in peanuts, but specific pathogen lying behind peanut root rots remains to be discovered[14,15]. Results indicated OTU177 (taxonomically assigned to *F. oxysporum*) and OTU90 (taxonomically assigned to *F. solani*) were significantly enriched in diseased peanut root. The relative abundance of these two fungal species was 34.1 and 2712.7 times higher in diseased than in healthy peanut roots respectively (Fig. 2d). In order to obtain cultures associated with the potential pathogens, we isolated 38 fungi from the diseased peanut roots. Based on growth morphology, 20 isolates were selected for 18S rRNA sequencing and subsequently were identified as *F. oxysporum* (5), *Penicillium* sp. 196F (1), *F. solani* (3), *Talaromyces pinophilus* (2), *Talaromyces verruculosus* (4), and *Neocosmospora striata* (5) (Supplementary Fig. 1). Next, we confirmed the pathogenicity of the two isolates linked with *F. oxysporum* and *F. solani*. *F. oxysporum* was highly pathogenic (51 ± 11%), resulting in significantly higher disease incidence than that observed upon inoculation with *F. solani* (35 ± 9%) (Student's *t*-test, $t = 5.580$, $df = 58$, $P < 0.001$) (Fig. 2e). Since other Fusarium were not enriched and cultivable, we identified our isolated strain (*F. oxysporum*) as the most likely organism behind the root rot disease observed in the studied peanut fields.

We conducted further quantitative real-time PCR (qPCR) analyses to gain deeper insights into the abundance and dynamics of *F. oxysporum* in the peanut rhizosphere at different growth stages. Consistent with the results of the disease index, there was no significant differences between monocropping and rotation regimes in *F. oxysporum* abundance at the peanut seedling stage (Student's *t*-test, $t = -1.525$, $df = 10$, $P = 0.158$, Fig. 2f). However, *F. oxysporum* abundance in the rhizosphere of monocropped peanut was significantly higher than that of rotation during the next developmental stages (Student's *t*-test, $t = 2.980$, $df = 10$, $P < 0.05$, Fig. 2f). Dynamics in pathogen accumulation after seedling stage in monocropped peanut showed a parallel pattern to that in peanut root rot disease, significantly affecting plant growth and reducing plant yield (Student's *t*-test, $P < 0.05$, Supplementary Fig. 2). Taken together, our results suggested that the effective colonization of *F. oxysporum* in the peanut rhizosphere has critical consequences for plant health in crops subjected to monocropping.

### Crop managements have significant effects on the peanut rhizosphere microbiome

The bacterial microbiome plays a crucial role in influencing the severity of soil-borne diseases compared to fungi. Plant roots can effectively recruit a higher diversity and richness of the rhizosphere bacterial community to suppress the invasion of pathogens, particularly those originating from fungal sources[16–18]. Therefore, we investigated the impacts of crop managements on the bacterial microbiome of peanut rhizosphere at different plant growth stages and bulk soil in our 2018 field experiment. In total, we obtained 1,369,637 sequences from 48 soil samples. Nonmetric multidimensional scaling (NMDS) based on the Bray–Curtis dissimilarity matrix revealed significant differences in bacterial community of monocropping and rotation at all development stages (ANOSIM, $P < 0.001$, Fig. 3a). Sequentially, we compared the variation of rhizosphere bacterial community between monocropping and rotation at seedling stage. Results indicated that

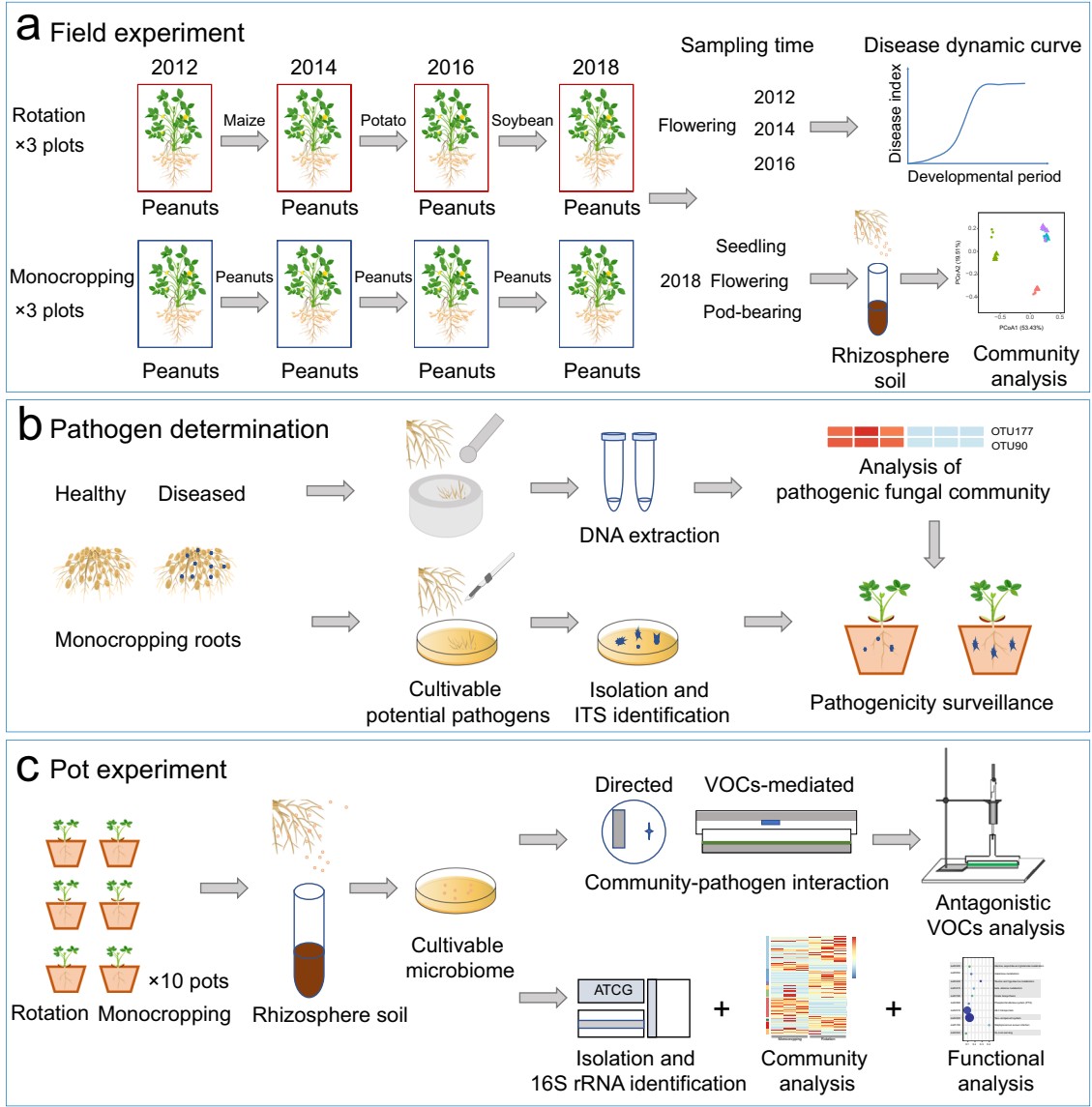

**Fig. 1 | Flow charts of field experiments for peanut disease investigation, identification of potential pathogens and pot experiment. a** From 2012, experimental plots were managed under two cropping regimes (treatments): peanut monocropping and rotation. For monocropping, peanuts were continuously planted from 2012 to 2018. For rotation, peanut was grown first (2012), and then maize (*Zea mays* L.), potato (*Solanum tuberosum*), and soybean (*Glycine max*) were ordinally planted in every other peanut planting year. The disease index of peanut was investigated at the flowering stage in 2012, 2014, and 2016. At 2018, peanut disease was investigated at seedling, flowering, and pod-bearing stage, and the rhizosphere soil was sampled for community analysis. **b** Roots from mature monocropped plants with healthy and diseased were used for pathogenic community analysis and potential pathogen identification. **c** During the end of 2018 planting season, soil samples were collected from six plots and used for pot experiments. The rhizosphere soil was collected at the seedling stage for culturable microbiome–pathogen interaction, community analysis, and functional analysis.

183 OTUs were enriched and 156 OTUs were depleted by monocropping ($P_{adjusted}$ < 0.05, Fig. 3b). Discriminating OTUs mainly clustered in Proteobacteria and Actinobacteriota (Fig. 3c). Further, the average relative abundance of 122 depleted OTUs (account for 82.4% of the discriminating OTUs), and 148 enriched OTUs (account for 83.1% of the discriminating OTUs) in monocropping was <0.1% (Fig. 3c, d).

**Rhizosphere microbiome of crops under rotation suppress the invasion of *F. oxysporum***

To examine the role of rhizosphere community in the occurrence of peanut root rot, we further collected soils from field plots of both monocropping and rotation to grow peanut at a greenhouse experiment (Fig. 1c). The rhizosphere microbiome of peanuts at seedlings was harvested to determine the ability of *F. oxysporum* inhibition. Antifungal activity against *F. oxysporum* was tested in antagonism assay of both volatile organic compounds (VOCs)-mediated and directed microcosms. The results showed that the rhizosphere microbiomes of peanuts grown under rotation regime were able to suppress the growth of *F. oxysporum*, by 45–56% higher than monocropped regime (Fig. 4a, b, Student's *t*-test, *P* < 0.01). This highlighted the significance of antagonism in rhizosphere community against pathogen invasion.

We then investigated the potential role of VOCs in mediating these results. We knew that VOCs produced by the rhizosphere bacterial community can play an important role in preventing fungal pathogen infection[19,20]. Thus, we used gas chromatography–mass spectrometry (GC-MS) to determine the composition of VOCs produced by the rhizosphere microbiome of monocropped and rotation peanut. Of the VOCs detected, dimethyl sulfide, 2,5-dimethylcyclohexanone, and 6-methyl-3,5-pentadien-2-one produced by the

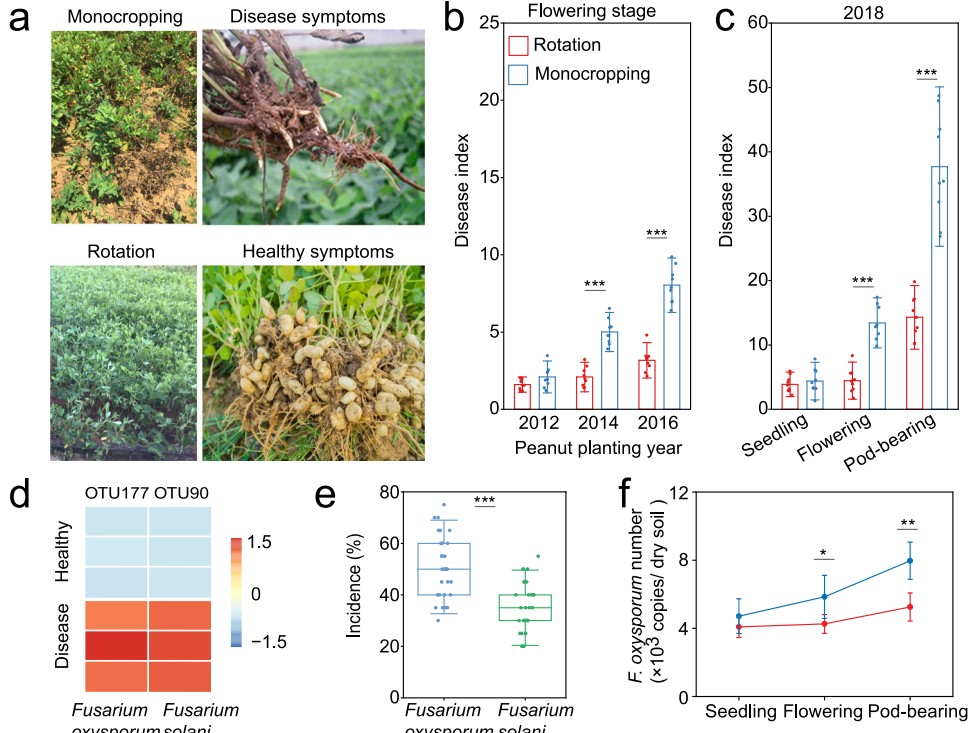

**Fig. 2 | Investigation of peanut disease and identification of pathogen in long-term field experiment. a** Symptoms of healthy and diseased peanuts in the field. **b** Disease index of peanut root rot from 2012 to 2016 at the flowering period ($P < 0.001$, $n = 9$ biologically independent samples). **c** Disease index of peanut root rot in monocropping and rotation regimes at 2018 across different growing periods ($P < 0.001$, $n = 9$ biologically independent samples). **d** Significantly increased OTUs (*Fusarium* sp.) in diseased roots compared with healthy roots. −1.5 to 1.5 represents the relative abundance of normalized OTUs. **e** Test for pathogenicity of the potential pathogens *Fusarium* spp. ($P < 0.001$, $n = 30$ biologically independent samples). Horizontal bars within boxes represent the median. The tops and bottoms of boxes represent 75th and 25th quartiles, respectively. The upper and lower whiskers represent the range of non-outlier data values. **f** The abundance of *F. oxysporum* in peanut rhizosphere of monocropping and rotation regimes in 2018 field experiment ($P < 0.05$, $n = 6$ biologically independent samples). Asterisks above the bars indicate statistically significant differences between the treatments based on two-sided tests by Student's *t*-test (*$P < 0.05$, **$P < 0.01$, ***$P < 0.001$). Each bars represents the mean ± SD.

rhizosphere microbiome from rotation-grown peanut were not detected in the rhizosphere microbiome from monocropped peanut (Supplementary Fig. 3). Further, the relative contents of α-acorenol, dimethyl disulfide, and 1,3-xylene were significantly reduced under the monocropping regime compared with rotation (Student's *t*-test, $P < 0.05$). Further in vitro experiments of pathogen suppression using standard VOCs (dimethyl sulfide, 2,5-dimethylcyclohexanone, 6-methyl-3,5-pentadien-2-one and 1,3-xylene), verified that these specific VOCs detected in peanut rhizosphere of rotation regime significantly inhibited pathogen growth, even at low concentrations (0.5–5.0 µg/mL; Supplementary Fig. 4).

In order to provide a more detailed understanding on the mechanisms behind these results, we further performed transcriptome analysis of cultivable microbiome from monocropping and rotation rhizosphere to understand the associated functional variation for *F. oxysporum* inhibition (Fig. 1c). We extracted mRNA of cultivable microbiome from monocropping and rotation rhizosphere after the VOCs-mediated antagonism assay, and transcribed it into cDNA for sequencing. Pathways enriched in different Kyoto Encyclopedia of Genes and Genomes (KEGG) orthology functional categories in cultivable rhizosphere microbiomes from monocropping and rotation were analyzed. The metabolic pathways that were primarily responsible for the differences in KO functional categories in the monocropping and rotation included ABC transporters and Two-component system (Fig. 4c). To be specific, the expression of genes associated with pathogen inhibition, including Isopentenyl-diphosphate Delta-isomerase (K01823), Glutamate decarboxylase (K01580), and 1-pyrroline-5-carboxylate dehydrogenase (K00294)

were significantly higher in rotation than that in monocropping (Fig. 4d, Student's *t*-test, $P < 0.05$).

## Cultivable rhizosphere microbiome under monocropping and rotation

To understand whether immature microflora was associated with decreased rhizosphere resistance in monocropping seedlings, we first collected cultivable rhizosphere microbiome from the above antagonistic experiments. Considering the difference in inhibition ability of the rhizosphere microbiome against *F. oxysporum* on agar plates between monocropping and rotation, we first compared the characteristics of the whole cultivable microbiome by Illumina sequencing. The number of sequences of the cultivable microbiome ranged from 14,804 to 22,748, with clustering as 714 OTUs at 97% similarity. Since the cultivable microbiome sequence came from all bacterial colonies growing on agar plates, we then determined OTUs depleted from monocropping or rotation rhizosphere cultures. Results indicated 362 OTUs, most belonging to *Bacillus*, *Enterobacter*, *Escherichia–Shigella*, *Pantoea*, *Enterococcus*, *Pseudomonas*, *Lysinibacillus*, *Paenibacillus*, *Kluyvera*, and *Jeotgalibacillus*, were not detected in the rhizosphere of monocropped peanut (Supplementary Fig. 5). By contrast, 257 OTUs, most belonging to *Bacillus*, *Pseudomonas*, *Virgibacillus*, *Burkholderia–Paraburkholderia*, *Ralstonia*, *Escherichia–Shigella*, *Klebsiella*, *Gemmatimonas*, *Acinetobacter*, *Paenibacillus*, and *Lysinibacillus*, were not detected in the rhizosphere of rotation-regime peanut (Supplementary Fig. 5).

In order to obtain the depleted and enriched strains in monocropping rhizosphere, we isolated 173 bacterial strains from agar plates of rhizosphere cultures for 16S rRNA sequencing. These

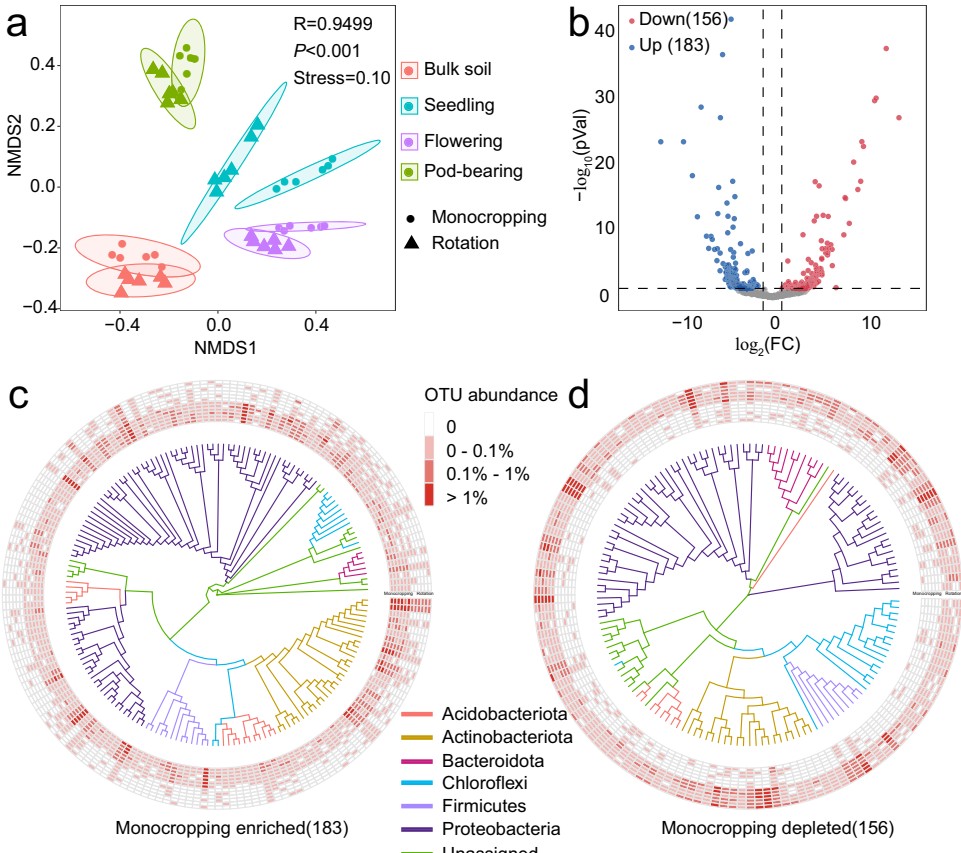

**Fig. 3 | Changes of rhizosphere microbial community in 2018 peanut season.** **a** Non-metric multidimensional scaling (NMDS) of microbial communities in rhizosphere of monocropping and rotation at seedling, flowering and pod-bearing stages. **b** Compared with rotation, enriched or depleted OTUs in monocropping seedlings rhizosphere (DESeq2, $P < 0.05$, FDR adjustment). **c**, **d** Phylogenetic trees constructed by enriched/depleted OTUs in monocropping seedlings rhizosphere and the relative abundance.

sequenced strains were mainly belonged to *Paenibacillus* sp., *Pantoea* sp., *Bacillus* sp., *Fictibacillus* sp., *Lysinibacillus* sp., *Enterobacter* sp., *Sporosarcina* sp., *Pseudomonas* sp., *Burkholderia* sp., *Stenotrophomonas* sp., *Serratia* sp., *Arthrobacter* sp., *Buttiauxella* sp., and *Brevundimons* sp. (Supplementary Fig. 6). Further, in order to determine the consistency of depleted or enriched OTUs with the above strains, we compared the similarity between the 16 S rRNA sequences of the isolated strains and the OTUs that were depleted or enriched in the cultivable microbiome of monocropping rhizosphere (Fig. 4e). When the sequence similarity between isolated strains and depleted or enriched OTUs exceeded 97%, the strain was considered representative of the corresponding OTU. We excluded the 16S rRNA sequence of *Bacillus* sp., a highly abundant genus widely present in the peanut rhizosphere, from the analysis. Among them, strains isolated from the rhizosphere of rotation-grown peanuts, i.e., *Paenibacillus* sp. (R60), *Pantoea* sp. (R05), *Lysinibacillus* sp. (R06), *Enterobacter* sp. (R09), *Sporosarcina* sp. (R07), *Fictibacillus* sp. (R37), and *Pseudomonas* sp. (R26), were 97% similar to OTUs depleted from the rhizosphere of monocropped peanuts, while strains belonging to *Stenotrophomonas* sp. (C20) and *Burkholderia* sp. (C63) were 97% similar to OTUs enriched in the rhizosphere of monocropped peanuts (Fig. 4e). Indeed, the abundance of the OTUs associated with the depleted strains was very low in the rotation plant rhizosphere (Fig. 4e).

### Characteristics of depleted strains responding to peanut root exudates
Plant root exudates are one of the sources that regulate rhizosphere microbial assembly[21]. To understand whether the depletion of low

abundance of taxa was associated with responsiveness to plant root exudates, we collected root exudates of peanuts grown on monocropping and rotation soil, and evaluated the effects of root exudates on bacterial growth in vitro. In this regard, all depleted bacterial strains had less sensitive response to root exudates of monocropped peanut as compared to that from rotation peanut (Student's *t*-test, $P < 0.001$, Fig. 4f). However, monocropping root exudates promoted the growth of *Burkholderia* and *Stenotrophomonas* that were enriched in monocropping rhizosphere (Student's *t*-test, $P < 0.001$, Fig. 4f). Compared to the enriched strains, the growth of depleted strains increased less in responding to root exudates from monocropping peanut.

Next, we determined potential plant growth-promoting properties of the above strains in vitro. In total, depleted strains of monocropping had potential growth-promoting properties. Of these, *Enterobacter* and *Pantoea* produced siderophores, and *Bacillus* and *Sporosarcina* solubilized organic phosphorus (Supplementary Table 1). Accordingly, *Paenibacillus* and *Lysinibacillus* produced more than 25 mg indoleacetic acid (IAA) per L. Except *Pantoea* and *Lysinibacillus*, all the other depleted strains inhibited the growth of *F. oxysporum* (Supplementary Table 1). Nevertheless, *Burkholderia*, a monocropping enriched strain, solubilized phosphorus and produced siderophores and IAA, but did not inhibit the fungal growth.

Finally, we determined the interaction between any two depleted bacteria. No significant enhancement or inhibition was observed between any two strains (Supplementary Fig. 7), indicating independent growth and stable coexistence among depleted strains.

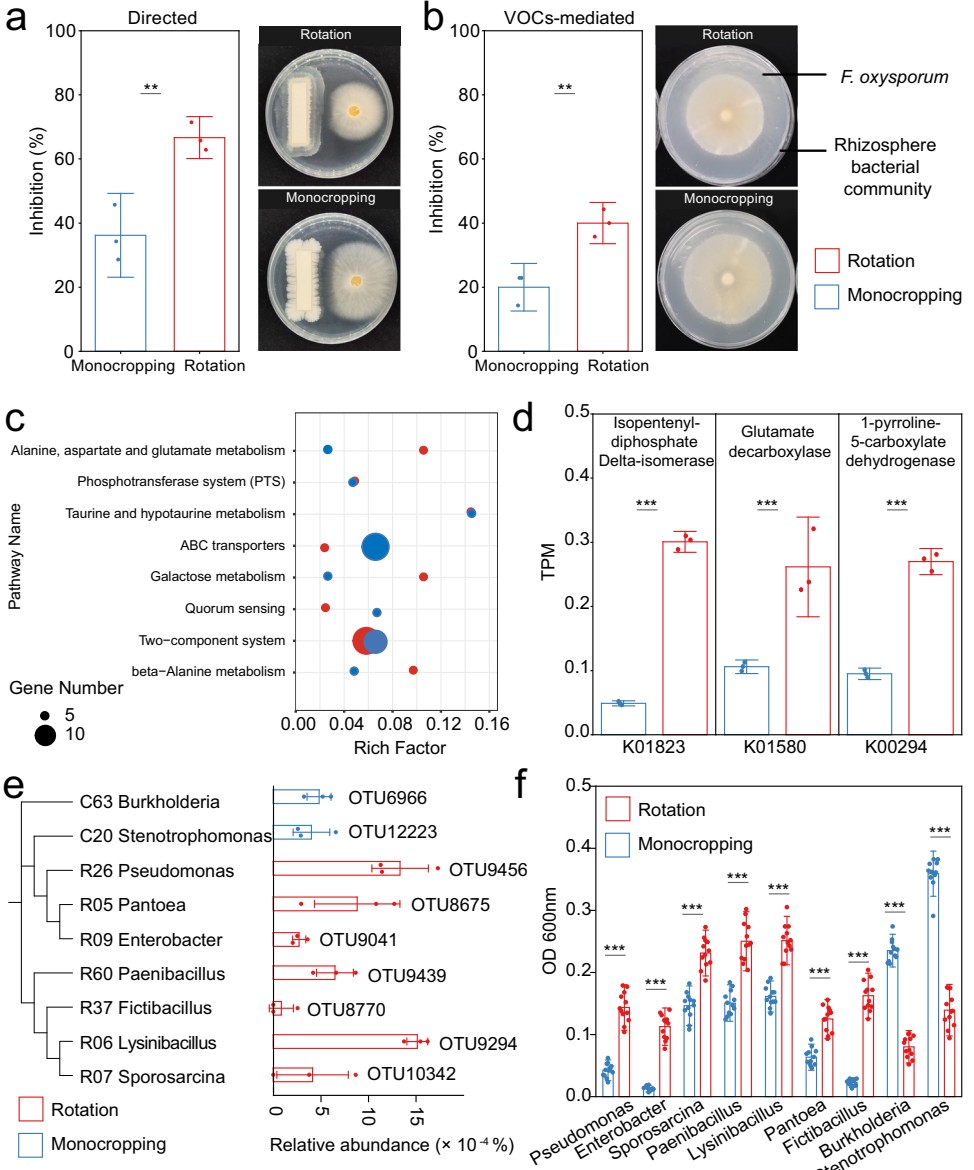

**Fig. 4 | Inhibition of pathogens by rhizosphere microorganisms in mono-cropping and rotation and determination of depleted bacteria in monocropping rhizosphere. a** Directed inhibitory effect of rhizosphere microbial community on *F. oxysporum* ($P = 0.006$, $n = 3$ biologically independent samples). **b** VOCs-mediated inhibitory effect of rhizosphere microbial com=munity on *F. oxysporum* ($P = 0.006$, $n = 3$ biologically independent samples). **c** Enrichment analysis of differential genes of culturable microorganisms in rhizosphere of monocropping and rotation. Rich Factor, the ratio of the number of differential genes in the metabolic pathway to the number of all genes annotated to the pathway ($n = 3$ biologically independent samples). **d** Abundance of representative genes associated with pathogen inhibition in cultivable microorganisms of monocropping and rotation. TPM, transcripts Per million ($P < 0.001$, $n = 3$ biologically independent samples). Each bars represents the mean ± SD. **e** The relative abundances of OTUs

in the monocropping and rotation samples. The blue bars show the relative abundance of monocropping-enriched OTUs in monocropping samples (<0.002% in rotation samples). The red bars show the relative abundance of monocropping-depleted OTUs in rotation samples (<0.002% in monocropping samples). The nine depleted or enriched OTUs with >97% similarity to isolates are shown ($n = 3$ biologically independent samples). Each bars represents the mean ± SEM. **f** Effects of peanut root exudates from monocropping and rotation on growth of monocropping-depleted and enriched bacteria ($P < 0.001$, $n = 12$ biologically independent samples). OD, optical density. Asterisks above the bars indicate statistically significant differences between treatments based on two-sided tests by Student's *t*-test (*$P < 0.05$, **$P < 0.01$, ***$P < 0.001$). Each bars represents the mean ± SD.

## Depleted strains associated with pathogen suppression by rhizosphere microbiome

We next asked why the depletion of specific strains depressed the ability of monocropped peanut rhizosphere to suppress *F. oxysporum*. We first analyzed the ability of different SynComs containing 7, 4, and 2 depleted strains to impact *F. oxysporum* pathogenicity (Fig. 5a and Supplementary Table 2). Overall, SynCom containing the 7, 4, or 2 depleted strains significantly suppressed mycelial development of *F.*

*oxysporum* (ANOVA, $F = 38.483$, $P < 0.001$, Fig. 5b). When the strain was inactivated, the inhibition disappeared. Moreover, the inhibition effect was positively correlated with the diversity of the depleted strains. When there were 7 depleted strains, SynCom showed the best inhibition of *F. oxysporum*. This indicated that all depleted strains participate in the suppression of fungal pathogens via synergistic interactions[22].

Next, we supplemented suspension of SynComs with different depleted strains to rhizosphere microbiome from monocropped

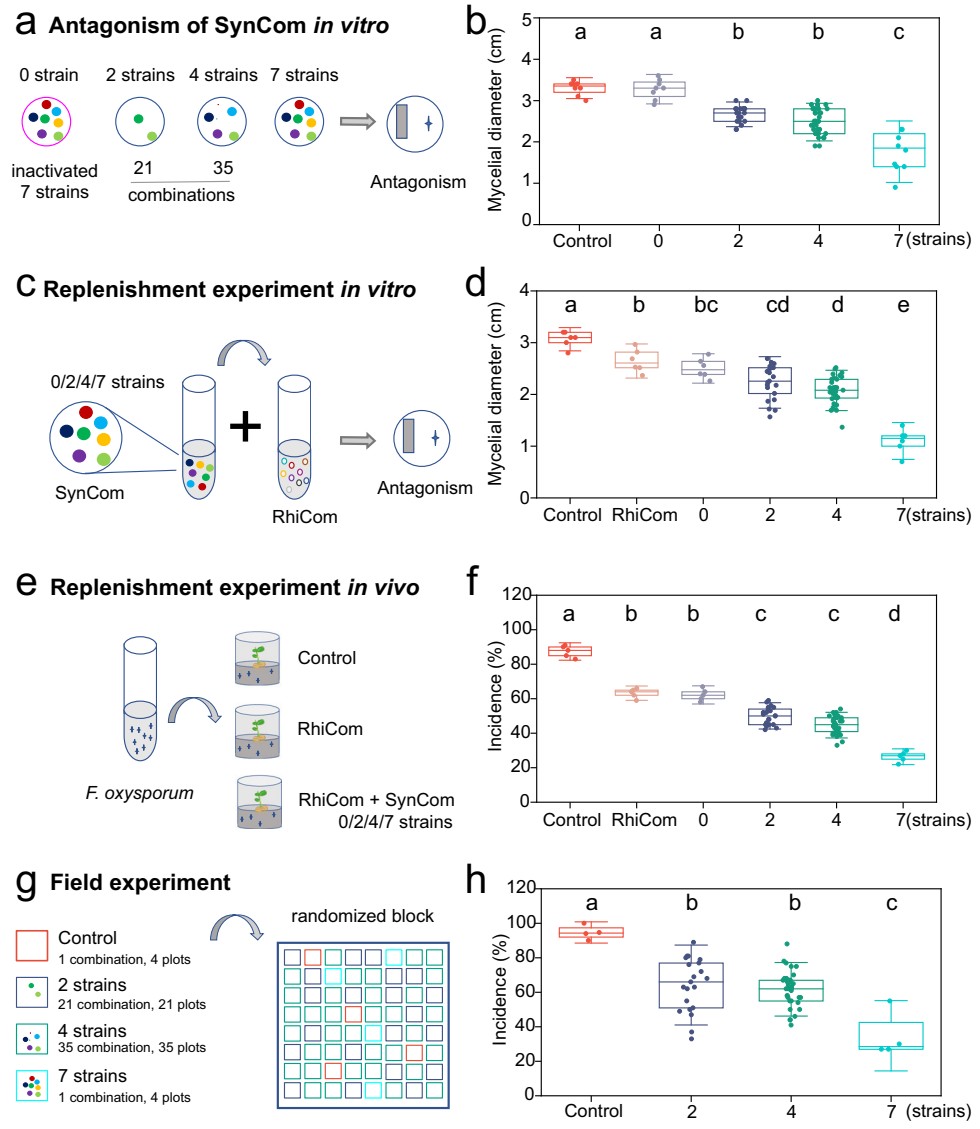

**Fig. 5 | Assessment of suppression ability of monocropping-depleted strains on pathogen development and peanut root rot. a** Overview of experiments performed to determine the supplemental effect of monocropping-depleted strains on pathogen invasion. **b** Effects of re-inoculation of different strain combinations on the growth of *F. oxysporum* ($P < 0.001$, $n_{control} = 8$, $n_7 = 8$, $n_4 = 35 \times 8$, $n_2 = 21 \times 8$, $n_0 = 8$; biologically independent samples). **c** Overview of a replenishment experiment performed to determine the inhibitory effect of the addition of depleted strains on *F. oxysporum* in vitro. **d** Inhibition of *F. oxysporum* by monocropping rhizosphere community supplemented with different strain combinations ($P < 0.001$, $_{control} = 6$, $n_7 = 6$, $n_4 = 35 \times 6$, $n_2 = 21 \times 6$, $n_0 = 6$). **e** Overview of a replenishment experiment performed to determine the disease index of peanuts in vivo. **f** Effect of monocropping rhizosphere community supplemented with different strain combinations on root rot occurrence ($P < 0.001$, $_{control} = 5$, $n_7 = 5$, $n_4 = 35 \times 5$,

$n_2 = 21 \times 5$, $n_0 = 5$). **g** Random block design of a field trial plot. **h** Effects of re-inoculation of different strain combinations on root rot occurrence in the field ($P < 0.001$, $n_{control} = 4$, $n_7 = 4$, $n_4 = 35 \times 4$, $n_2 = 21 \times 4$). In figures **b, d, f** and **h**, different letters above the **b**ars indicate statistically significant differences between treatments. *P* values were calculated using ANOVA's test based on two-sided. RhiCom, treatment with monocropping rhizosphere community; SynCom, supplemented synthetic community composed of monocropping-depleted strains to monocropping rhizosphere community; 0, a SynCom composed of 7 inactivated depleted strains; 2, SynComs composed of 2 depleted strains; 4, SynComs composed of 4 depleted strains; 7, a SynCom composed of 7 depleted strains. Each bars represents the mean ± SD. Horizontal bars within boxes represent the median. The tops and bottoms of boxes represent 75th and 25th quartiles, respectively. The upper and lower whiskers represent the range of non-outlier data values.

peanuts and co-cultured on a plate with *F. oxysporum* (Fig. 5c). When the monocropping rhizosphere suspensions were supplemented with SynComs, the pathogen growth was inhibited by 64.4%, with the inhibition 4-fold higher than that of the control without rhizosphere suspension (ANOVA, $F = 33.501$, $P < 0.001$, Fig. 5d). By contrary, an inactivated SynCom did not make up for the suppression ability of monocropping rhizosphere microbiome (Fig. 5d). The suppression ability on pathogen increased with the supplementation of SynCom diversity, indicating depleted strains compensate the ability of monocropping rhizosphere microbiome to resist pathogen invasion.

Further, we tested the compensatory effect of the depleted strains in experiments with sterile peanut seedlings (Fig. 5e). Compared with seedling inoculation with the monocropping rhizosphere suspension, supplementation with the depleted strains significantly improved rhizosphere resistance to *F. oxysporum* invasion (ANOVA, $F = 71.755$, $P < 0.001$, Fig. 5f). Consistent with in vitro antagonism, the best inhibitory effect was obtained when the SynCom composed of 7 strains was supplemented. In the treatment with no SynCom inoculation, the root rot protection collapsed. However, compared with the treatment with rhizosphere community, supplementation of the depleted strains did not affect plant growth (ANOVA, $P > 0.05$, Supplementary Fig. 8).

Finally, we conducted field experiments to verify the compensatory effect of the depleted strains on the occurrence of root rot in monocropped peanuts (Fig. 5g). Peanut seeds were inoculated with SynComs composed of 7, 4, or 2 strains by soaking. Results indicated compared with the control without their inoculation, SynComs significantly reduced the incidence of peanut root rot (ANOVA, $F = 16.169$, $P < 0.001$, Fig. 5h). SynComs composed of 7 bacterial strains were most effective (Fig. 5h). Similarly, no SynCom tested significantly promoted peanut growth compared to the control (ANOVA, $P > 0.05$), indicating their biocontrol effect in the plant rhizosphere rather than a direct promotion of plant resistance (Supplementary Fig. 9).

## Discussion

In the current study, we investigated the influence of agricultural practices on soil disease suppression. We further manipulated the microbiome of our soils aiming to bring back the capacity of rhizosphere to fight pathogens and disease. We found that the depletion of rhizosphere specific bacteria in response to monocultures of peanut and its associated root exudates can reduce the capacity of soils to suppress disease and further determined the future responses of plant to disease outcomes. On the contrary, crop rotations supported soils with a greater capacity to suppress disease, and we were able to restore rhizosphere communities to protect soils from pathogens using microbial innocula. Our study adds to the consensus that monocultures have a reduced soil disease suppression capacity compared to crop rotations.

Our study revealed that monocultures had a reduced soil disease suppression capacity. In particular, our five-year field peanut experiment, monocropping significantly increased the incidence of root rot disease, compared with fields subjected to rotation-grown peanut. Thus, these contrasting agricultural managements supported opposite capacity to control disease occurrence by preventing soil-borne pathogen invasion. We then tracked root rot occurrence over an entire peanut growing period in 2018. The results showed that disease indices and pathogen abundances were not significantly different under monocropping and rotation regimes at the seedling stage. However, there was a dramatic increase in disease incidence at later growth stages in monocropped peanut compared with rotation-grown peanut. Interestingly, we found significant differences in the rhizosphere microbiome of peanuts in monoculture and crop rotation from seedling stage to the entire plant development. These differences were associated with different disease outcomes, indicating the importance of rhizosphere microbiome for the development of these plants and their disease. Recent studies have shown that differences in rhizosphere microbial composition of seedlings lead to changes in microbial function, and eventually differentiate into two different results (diseased or healthy)[23]. Therefore, our work suggests that strengthening seedling management by focusing on rhizosphere microbiome could contribute to maintaining plant health.

We then sought to understand the mechanisms behind the ability of the rhizosphere microbiome to support pathogen suppression during the seedling stage. As explained above, the ability of the rhizosphere microbiome from monocropped peanuts to inhibit pathogen was significantly lower than that from rotation-grown peanuts. These results indicated that the microflora at seedling stage could determine the suppression ability of plant rhizosphere to pathogen invasion. The underlying mechanisms behind inhibition would associate with the production of volatile organic compounds and antibiotics[24,25]. For instance, dimethyl sulfide, 2,5-dimethylcyclohexanone, and 6-methyl-3,5-pentadien-2-one that were not detected in the rhizosphere microbiome of monocropping had strong inhibition on pathogen growth even at low concentrations. In terms of rhizosphere microbiome function, the expression of genes associated with pathogen inhibition, including Isopentenyl-diphosphate Delta-isomerase, Glutamate decarboxylase, and 1-pyrroline-5-carboxylate dehydrogenase were significantly higher in rotation compared with monocropping[26]. Our results revealed the potential mechanisms behind the greater disease suppression capacity in croplands under rotation.

The negative influence of root exudates in monoculture peanut field on key bacteria supporting suppression may also explain the reduced capacity to protect soils against pathogens and disease. This highlights the importance of low-abundance microbial species regulating plant rhizosphere resistance, as the ability of the community assembly without them in the monocropped peanut rhizosphere to suppress pathogen growth was reduced. In our study, the growth-promoting effect of monocropping root exudates on *Burkholderia* sp. and *Stenotrophomonas* sp. may lead to the occupation of ecological niche, and thus affect the rhizosphere depletion of bacteria with a weak response to monocropping root exudates such as *Paenibacillus* sp., *Pantoea* sp., *Lysinibacillus* sp., *Enterobacter* sp., *Sporosarcina* sp., *Fictibacillus* sp., and *Pseudomonas* sp. Accordingly, the enriched strains preferentially occupy root resources, and the depleted taxa are thereby unable to effectively participate in community assembly in the monocropped rhizosphere.

We then investigated whether reintroducing the depleted microbial taxa could bring back the capacity of soils to resist pathogen invasions and avoid disease. Despite the growing interest in SynComs, the detailed mechanisms by which they operate remain largely unknown. We first determined the inhibitory capacity of microbial communities composed of random combinations of the deleted strains in monoculture compared with rotation croplands. Our result revealed that microbes that were lost under monoculture, participated in the suppression of fungal pathogens via synergistic interactions. We secondly reintroduced these strains into the rhizosphere of monocropped peanuts. The supplementation significantly alleviated pathogen attack of peanut root, strongly supporting the notion that the addition of low abundance microbes during community succession can improve the overall function of the rhizosphere community[27]. The inhibitory capacity of microbial communities composed of random combinations of strains was consistent with an increased phylogenetic diversity, suggesting that all depleted strains participate in the suppression of fungal pathogens via synergistic interactions[22]. Indeed, our microbial inoculation experiments revealed that the reintroduction of depleted strains strongly protected the plant root against pathogen infection, while failing to promote plant growth. Data obtained by different approaches suggests that the substantial participation of the depleted strains in the resident community assembly could determine their success in fulfilling functional services in the rhizosphere[4,28].

Applications of single-function microbes or SynComs composed of multi-functional microbes generally have satisfactory outcomes in the laboratory setting but not in practice. Indeed, lack of consideration of the appropriate ecological niches for functional strains, factors driving the variation in resident microbial interactions, and the impact of environmental factors on microbial colonization ability can have undesirable effects[29]. In the field experiments presented in the current study, SynCom composed of 7 depleted strains showed an excellent ability to prevent root rot. SynComs composed of some depleted strains also controlled root rot, suggesting the stability of their application. Recently, a simplified SynCom, developed based on host-mediated microbial community selection, had effectively captured the dominant members of the maize root microbiota, inhibiting pathogen colonization[30]. However, it is unlikely that the such simplified SynComs would take over all the functions of rhizosphere microbiota[31]. Conversely, specific microbial taxa that drive resident microbial community assembly can amplify rhizosphere community functions. For instance, the addition of biological agents containing *Bacillus amyloliquefaciens* W19 protected banana from fungal pathogen infection by activating specific plant-beneficial bacterial genera (e.g. *Pseudomonas*)

to form a plant-benefitting consortium[27]. Consequently, the current study seeks to rationally control plant disease from a holistic microbial perspective, in that SymComs should switch core from basic functions to activate community functionality.

Taken together, our results indicate that management and rhizosphere microbiomes play a role in supporting soil pathogen suppression against important soil disease. Our work shows that, compared with rotation managements, intensive monocropping strategies weakened the capacity of soils to prevent the entrance of pathogens to the rhizosphere, and promoted the development of soil-borne diseases. Importantly, we provide experimental evidence that pathogen resistance in weakened rhizospheres may be restored by applying targeted microbial inocula. Therefore, our study highlights the role of management to fight against soil crop disease and suggests the active restoration of rhizosphere communities to promote soil disease suppression.

## Methods

### Field site description and disease assessment
From April 2012, we conducted a field experiment at Yujiang County, Jiangxi Province, China. The soil at the study site is classified as Udic Ferrosol (FAO classification), covering an area of $-1 \times 10^8$ ha in southern China. Because of low soil organic matter content and fertility, and climatic suitability, peanut (*Arachis hypogaea* L.) is a particularly popular crop in the region. The field experiment included two cropping regimes (treatments): (1) monocropping with peanut, the same peanut cultivar (Ganhua-5) was consecutively planted for the growing season (April–August) of each year; and (2) rotation, peanut was grown first (2012), and then maize (*Zea mays* L.), potato (*Solanum tuberosum*), and soybean (*Glycine max*) were ordinally planted in every other peanut planting year (Fig. 1a). Three plots of the two cropping regimes were laid out in a randomized block design, and lay fallow after harvest until the following sowing period. Commonly used management practices, including tillage, fertilizer application, and weed control, were applied manually[32].

In the 2012, 2014, and 2016, all plots were planted with peanut, and the severity of root rots was persistently evaluated at flowering stage. In 2018 peanut planting season, we further examined peanut root rots at seedling, flowering and pod-bearing stages (Fig. 1a). For each examination, 30 plants from each plot were carefully removed from the soil, and peanut root rot was evaluated using a five-class rating scale (0 = no lesions, 1 = small root lesions, 2 = central root lesions, 3 = large root lesions, 4 = dead plant)[15]. Overall, 540 plants (6 plots, 3 time points) were removed from the plots for examination. Following a gentle wash with tap water, the plant height, shoot and root fresh weight, root length, and nodule number were determined at each sampling.

### Sample collection and processing
Bulk soil and rhizosphere soil samples were collected before the peanuts harvest in 2018. Before peanut planting, 10 soil cores with a depth of 20 cm were randomly selected from each plot. Every 5 soil cores were fully mixed into a single composite soil sample as bulk soil. Excess soil on the roots was discarded by gently shaking the plants, and the remaining soil particles attached to the root surface were collected as rhizosphere soil. At seedling, flowering, and pod-bearing stage, 10 peanuts were randomly selected from each plot, and rhizosphere soil of every 5 peanut plants was fully mixed into a single composite soil sample. In total, 36 soil samples (6 plots, 3 time points) were collected. Soil DNA of rhizosphere soil collected from rhizosphere and bulk soil ware extracted for microbial community analysis. Roots from healthy or severe disease peanuts at pod-bearing stages were surface-sterilized in 3% hydrogen peroxide, and washed with sterile water and 70% ethanol. Excess fluid on the sterilized root surface was wiped off using sterilized filter papers and root samples were cut into pieces. All the root samples were stored at −80 °C before DNA extraction. In addition, a portion of diseased peanut roots were used to isolate potential pathogens.

### Sequencing analysis of endophytic fungal community
Root samples were ground in liquid nitrogen and DNA was extraction using the MiniBEST Plant Genomic DNA Extraction Kit (Takara, Japan) (Fig. 1b). Specific sequences were amplified using the primer pairs ITS1F(5'-CTTGGTCATTTAGAGGAAGTAA-3')/ITS2(5'-GCTGCGTTCTTC ATCGATGC-3'). The samples were initially denatured for 3 min at 95 °C; this was followed by 27 cycles of denaturation (95 °C, 30 s), annealing (55 °C, 30 s), and elongation (72 °C, 45 s). The PCR program ended with a 10-min incubation at 72 °C. The PCR products were separated by electrophoresis on 1% agarose gel. All samples were pooled in equimolar concentrations and then sequenced with a paired-end protocol at Majorbio Bio-Pharm Technology Co. Ltd. (Shanghai, China) using the Illumina MiSeq platform, according to the manufacturer's instructions. Raw fastq files were quality-filtered using QIIME[33]. Low-quality sequences (<150 bp long, with an average quality score <25) were removed. The reads were trimmed and assigned based on unique 7-base barcodes. The barcode and primer sequences were then removed. The forward and reverse reads were incorporated into full-length sequences based on the thresholds: overlap length >10 bp and mismatch ratio <0.2. After discarding unqualified reads, the OTUs were assigned at 97% identity similarity level using UPARSE[34]. Chimeric sequences were identified and removed using UCHIME[35]. Taxonomic assignment was performed using UNITE database (v7.0) for fungi[36].

### Isolation and identification of potential pathogens
Potential fungal pathogens were isolated from peanut roots that displayed disease symptom[37]. Briefly, a clean knife was used to cut the pathogen-infected roots into sections. The root sections were surface-sterilized (submerged in 4 v/v sodium hypochlorite for 5 min), washed (two times, in sterile distilled water), and placed on a PDA plate containing streptomycin and penicillin (20 µg/mL) to obtain fungal isolates[38,39]. DNA was extracted from each fungal culture by using the FastDNA Spin Kit (MP Biomedicals, Santa Ana, CA), according to the manufacturer's instructions. DNA sequences from the fungal 18S rRNA region were amplified using primers NS1(5'- GTAGTCATATGCT TGTCTC-3') and NS8(5'-TCCG-CAGGTTCACCTACGGA-3')[40]. The PCR mixture contained (per 50 µL) 1.5 U of Taq polymerase (Red Taq, Sigma Chemical Co.) and the following reagents: $1 \times$ Sigma PCR buffer, 0.20 mM PCR nucleotide mix (Promega), 4.0 mM $MgCl_2$, 6.25 mg bovine serum albumin (Roche Diagnostics), and 25 pmol of each primer. For the amplification reaction, the DNA samples were initially denatured for 3 min at 95 °C. This was followed by 35 cycles of denaturation (94 °C, 30 s), annealing (57 °C, 30 s), and elongation (72 °C, 105 s). The PCR program ended with a 2-min incubation at 72 °C[40]. Each fragment was compared phylogenetically to sequences of known species in the GenBank database of the National Center for Biotechnology Information (NCBI) by using BLAST. Phylogenetic trees were analyzed using MEGA v5. Phylogenetic trees were constructed by using the neighbor-joining (NJ) method.

### Determination of pathogenicity of potential pathogens
Potential pathogens were highly enriched isolates from the diseased roots, including *F. oxysporum* and *F. solanum*. A fungal plug (6-mm diameter) was transferred to PDA medium and cultured at 28 °C for 7 days. Obtained fermentation solution was filtered to remove mycelia, and centrifuged for 10 min ($3000 \times g$) to retain precipitation. The precipitated spores were re-suspended with sterile water, and the concentration of spore suspension was adjusted to $10^9$ CFU/mL. To confirm pathogenicity of potential pathogens, peanuts were planted in glasshouse (30 °C, 70% relative humidity, light intensity 500 µM m$^{-2}$ s$^{-1}$). 10 mL spore suspensions of *F. oxysporum* or *F.*

*solanum* were poured to 14-day-old peanuts rhizosphere, and the incidence of disease was recorded 30 d after inoculation.

## Determination of pathogen abundance in peanut rhizosphere

The changes of *F. oxysporum* abundance in the peanut rhizosphere from the seedling to pod-bearing stages were quantitatively analyzed by fluorescence quantitative PCR. The following primer pairs were used: Fa, 5′-TCGTCATCGGCCACGTCGACTCT-3′, and Ra, 5′-CAAT-GACGGTGACATAGTAGCG-3′[41]. The reaction contained 7 μL of double-distilled $H_2O$, 10 μL of 2× SYBR® Green master-mix (Bio-Rad Laboratory, CA, USA), 0.5 μL of each primer (10 μM), and 2 μL of template DNA. The PCR procedure was as follows: 95 °C for 30 s; and 40 cycles of 95 °C for 5 s, 57 °C for 60 s, and 72 °C for 60 s. Each sample was analyzed in three replicates.

## DNA extraction and sequencing from field experiment

Bacterial community analysis was performed on rhizosphere soils (seedling to pod-bearing stage) and bulk soil in field experiments. DNA was extracted from soil and cultivable microbial community using the FastDNA® SPIN Kit for the Soil (MP Biomedicals, Santa Ana, CA). Specific sequences were amplified using the primer pairs 338F (5′-ACTCC-TACGGGAGGCAGCAG-3′)/806 R (5′-GGACTACHVGGGTWTCTAAT-3′), specific for the V4 hypervariable region of the bacterial 16S rRNA gene[42]. The purified PCR products were sequenced on the Illumina MiSeq platform at Majorbio Bio-Pharm Technology Co. Ltd. (Shanghai, China). Raw data were assembled and quality-filtered, and chimeric sequences were removed using the UCHIME tool in USEARCH[35]. After discarding unqualified reads, the remaining effective sequences were clustered into operational taxonomic units (OTUs) at 97% similarity. Taxonomic assignment was performed using SILVA reference database (v12.8) for bacteria[43]. One sample failed quality control in both monocropping and rotation at pod-bearing stage, thus the average of other 5 replicates was used to replace the abandoned sample.

## Peanut seedling cultivation in pot experiment

To isolate the peanut rhizosphere microbes and assess their ability to suppress fungal pathogen invasion, soil samples were collected for pot cultivation experiments after 2018 peanut planting season, which prevented disorganizing the field plot experiment (Fig. 1c). Peanut seeds were surface-sterilized with 0.1% mercury chloride for 5 min, and then rinsed five times with deionized distilled water. For each plot, 20 kg of the soil (0–20 cm layer) was randomly collected into 10 pots, and one surface-disinfected peanut seed (Ganhua-5) was sown in each pot. Overall, there were 30 biological replicate pots for each crop treatment, with 60 experimental units (10 pots × 3 field plots × 2 crop treatments). After cultivation for 28 d (30 °C, 70% relative humidity, light intensity 500 μM m$^{-2}$ s$^{-1}$), the plants were carefully removed from the pots and rhizosphere samples were collected by brushing off the soil adhering to the roots. The rhizosphere soil from 10 pots per plot was pooled. These six independent replicates from monocropping and rotation regimes were used for subsequent isolation of the rhizosphere bacterial colonies, the assessment of fungal pathogen suppression, analysis of cultivable microbiome and metatranscriptome.

## Suppression of peanut rhizosphere community on fungal pathogen development

The suppression of a fungal pathogen by the rhizosphere soil was determined by VOC-mediated and directed microcosm antagonism assays (Fig. 1c). *F. oxysporum* isolated from the peanut root rot was used. Before the cultivation experiment, a fungal plug (6-mm diameter) was transferred to PDA medium and placed in a biochemical incubator at 28 °C for 3 d in the dark. To prepare soil bacterial suspensions, 1 g (dry weight equivalent) of fresh rhizosphere soil was placed in 9 mL of phosphate buffer ($KH_2PO_4$, 1 g/L, pH=6.5), and mixed on a rotary shaker at 4 °C and 150 rpm for 1.5 h. The suspension was

then sonicated for 1 min at 47 kHz, twice, and mixed again for 0.5 h[44]. The suspension was filtered through a 5-μm filter to remove most fungal propagules before testing in vitro.

For the VOC-mediated antifungal activity, an inverted assay was performed using sterile Petri dishes (9-cm diameter)[45]. Briefly, a plug containing pathogenic fungal hyphae was inoculated at the bottom of a Petri dish containing PDA, and incubated at 28 °C for 24 h. A layer of nutrient agar (TSA) (1.5 g/L tryptone, 0.5 g/L soytone, 0.5 g/L NaCl, and 15 g/L agar, pH 7.0) was poured into the lid of the Petri dish. After solidifying, 200 μL aliquots of bacterial suspension were evenly spread on the TSA. As a control, an equal amount of sterile water instead of the bacterial suspension was used. Then, the top of the dish was placed over the matching bottom part, sealed with Parafilm, and incubated at 28 °C until the control PDA plates were filled with mycelia. In this way, the fungi to be tested were exposed to any antifungal substances produced by bacteria in the upper compartment. Each treatment was repeated thirty times. Percentage inhibition was calculated as [(fungal mycelium diameter in the control)−(fungal mycelium diameter in the treatment)] × [100/(fungal mycelium diameter in the control)].

For the directed antifungal activity, 60 μL of bacterial suspension was inoculated evenly on a piece of sterile filter paper (1.8 cm × 4.8 cm) placed on one side of TSA medium. After 3 d, a PDA plug (8-mm diameter) containing fungal hyphae was placed 2 cm away from the filter paper. The plates were sealed with Parafilm and incubated at 28 °C. The fungal pathogens were exposed to the secretions produced by the bacterial communities on the filter paper. The extension of fungal hyphae was measured after 72 h and compared with that on the control plates inoculated with sterile water instead of the bacterial suspension. Each sample contained 10 technique replicates.

## Metatranscriptome sequencing of cultivable rhizosphere microbiome

Following the VOC-mediated antagonism assay, 4 mL of sterile deionized water were added to the plate containing bacterial colonies, and suspensions of rhizosphere microbiome were obtained by mixing with a sterile inoculating loop. Total RNA was extracted from the bacterial suspension using the UltraClean® Microbial RNA Isolation Kit (Qiagen, German), according to the manufacturer's instructions. Then, mRNA was enriched using the MICROBExpress™ Bacterial mRNA Enrichment Kit (Thermo Fisher scientific, USA). The enriched mRNA sample was used as a template in a reverse-transcription reaction to synthesize cDNA. The constructed metatranscriptome library was then sequenced using the Illumina HiSeq 4000 platform at Shanghai Personal Biotechnology Co., Ltd (Shanghai, China). Based on the FastQC quality report, filtered mRNA reads were cleaned using Cutadapt v1.2.1[46]. Specifically, reads <50 bp and 10 bp minimum overlap with maximum 20% mismatch were removed.

For functional analysis, the quality-filtered reads were sorted using SortMeRNA[47], which separated rRNA from non-rRNA and thus, potential mRNA. Trimmed high-quality sequences were assembled de novo using Trinity v2.2.0[48]. Sequences were not assembled into contigs to avoid the problem of chimera formation arising from highly diverse communities[49]. Cluster Database at High Identity with Tolerance (CD-HIT) was used for sample collation after assembly and splicing. Transcripts were merged to eliminate redundancy at a similarity of 0.95 and a minimum coverage of 0.9, and the longest sequence was used as the representative sequence in UniGene. Bowtie2 v2.2.9 and RSEM v1.3.0 were used to functionally annotate Trinity contigs, and to align reads and quantify transcripts. Gene expression levels were determined by RNA sequencing (RNA-seq) as transcripts Per Kilobase of exon model per Million mapped reads (TPM)[50]. Five samples out of six passed quality control after sequencing; sample "C1" yielded very low sequence counts and was not included in the analysis. Therefore, the average value of two monocropping samples was taken for difference analysis. All genes in the catalogue were translated to amino

acid sequences and aligned with data in the Kyoto Encyclopedia of Genes and Genomes (KEGG) database[51].

## Determination of fungal suppressive compounds in rhizosphere community

For VOCs collection, 200 µL of the rhizosphere bacterial suspension was inoculated on TSA medium, sealed with Parafilm, and in the dark in a 28 °C incubator for 3 d. Then, the dish lid was pierced with a solid phase microextraction needle (Supelco, Bellefonte, PA) and the fibers were exposed to the sample headspace for 60 min. After the extraction, the fibers were retracted into the needle, transferred immediately to the injection port of a GC (GC-CP3800, Agilent Technologies, USA), and desorbed at 250 °C for 3 min. The compounds collected in the headspace were analyzed by the GC connected to a mass spectrometer (Saturn 2200, Varian, USA). The column used was 30 m × 0.25 mm internal diameter, with film thickness of 0.25-µm (CP-8, Agilent Technologies, USA). Helium was the carrier gas, at a constant linear velocity of 1 mL/min. The following temperature program was used: 50 °C for 2 min, then ramping up to 260 °C at 5 °C/min, and holding for 2 min. The mass spectra were scanned at 70 eV over a mass range from $m/z$ 35 to 600. VOCs were identified based on their individual mass spectrum, retention index (RI), and retention time (RT) by comparing with those in reference databases (NIST Mass Spectral Data 08' edition). Peak areas of all components were calculated by using Xcalibur 2.0, and relative amounts (RAs) were calculated based on the peak-area ratios of all volatiles[52].

## Determination of suppression capacity of different VOCs on fungal pathogen

The effect of pure VOCs (Dimethyl disulfide, 1,3-Xylene, 2,5-Dimethylcyclohexanone, and 6-Methyl-3,5-heptadiene-2-one) on mycelial growth of *F. oxysporum* was assessed using 5-mm agar plugs from the edge of actively growing mycelial colonies, which were placed downward-faced in one compartment filled with 1/5 PDA medium. Each pure VOC was diluted ten-fold in DMSO (Sigma-Aldrich) and serially diluted in distilled water to obtain the appropriate concentration for each treatment. The respective VOCs (0, 0.05, 0.5, 2.0, and 5.0 mg) were applied to a filter paper disk on the dish lid (without physical contact with *F. oxysporum*), corresponding to a concentration of 0 (control), 0.5, 5.0, 20 and 50 mg/L in air volume (VOC-treated) calculated assuming the complete VOC evaporation from the filter paper. Dishes were sealed with Parafilm and incubated in the dark at 28 °C for 7 d. Each treatment was repeated eight times. Percentage inhibition was calculated as above described.

## Sample preparation, DNA extraction, and sequencing of cultivable microbiome

To obtain cultivable microbial communities, rhizosphere bacterial suspensions prepared as above were plated on TSA, and incubated at 28 °C for 3 d in the dark. To this end, 4 mL of sterile deionized water were added to the plate containing bacterial colonies, and suspensions of rhizosphere microbiome were obtained by mixing with a sterile inoculating loop. Bacterial suspensions were then centrifuged for 10 min at 10000 × $g$ after which the pellets were used for DNA extraction using the FastDNA® SPIN Kit for the Soil (MP Biomedicals, Santa Ana, CA). Sequencing details refer to soil bacterial community analysis.

## Isolation and identification of cultivable bacterial strains

For the experiment, all colonies were picked from plates and further re-streaked on fresh TSA plates for purification. According to the morphology (color and size) of the bacterial colonies, 173 isolates were selected for 16S rRNA gene sequencing for identification. DNA was extracted using the FastDNA® SPIN Kit for the Soil (MP Biomedicals, Santa Ana, CA), according to the manufacturer's instructions. The region of interest was amplified using PCR primers 27F, 5′-AGAGTTTG

ATCCTGGCTCAG-3′, and 1492R, 5′-GGTTACCTTGTTACGACTT-3′[53]. The PCR cycle consisted of an initial denaturation at 94 °C for 5 min; followed by 35 cycles of denaturation at 94 °C for 1 min, annealing at 54 °C for 1 min, and extension at 72 °C for 1 min; and a final extension for 5 min at 72 °C. PCR products were verified on 1% agarose gel and sequenced by Sangon Biotech Co., Ltd. (Shanghai, China). The BLAST search program was used for sequence identification. Highly homologous sequences were aligned, and NJ trees were generated using MEGA v5. To confirm the existence of these bacterial isolates in the peanut rhizosphere, similarity analysis was performed based on the 16S rRNA sequence of the bacterial isolates and gene sequences of OTUs (same genus with isolates). Isolates with the V4 region matching OTUs with more than 97% identity were considered to be the same strain.

## Properties of depleted and enriched strains in monocropping rhizosphere

To investigate why certain strains isolated from the rotation peanut rhizosphere were depleted from the monocropped peanut rhizosphere, their response to root exudates of monocropping and rotation peanuts were determined. To obtain root exudates, sterilized seeds were transferred to pots containing monocropped or rotation soils. An equal dose (10 mL) of sterile 1/2 Hoagland solution every ten days were supplemented. After planted for 30 d, root exudates were collected from 20 seedlings. To do that, plant roots were washed four times with sterile double-distilled water to remove the nutrient solution. Then, each plant was placed into a 50-mL flask and the roots were submerged in 50 mL of sterile double-distilled water. All plants were placed in a growth chamber for 24 h (16-h light/8-h dark photoperiod) at 28 °C with gentle shaking (50 rpm).

To determine bacterial strains growth responding to peanut root exudates from monocropping and rotation, 20 µL of bacterial suspension and 180 µL of root exudates were placed in a sterile 96-well plate. The test was repeated 12 times for each bacterial solution, and a bacterial suspension mixed with sterile water was used as a control. Sample absorbance was measured using a spectrophotometer at optical density (OD) 600 nm every 12 h, and the response of different bacteria to root exudates was determined based on changes in bacterial density.

To understand the functional properties of the monocropping-depleted and enriched bacteria, phosphate solubilization, and siderophore and IAA production capacity of the bacteria were further tested, as well as their inhibitory effect on *F. oxysporum*[54]. Two microliter bacterial suspension of each bacteria was spotted on Pikovskaya (PVK) agar plates, respectively. After 5 days of plate incubation, the bacterial colonies with clear zone were phosphate-soluble bacteria and D (Diameter of zone of clearance)/D (Diameter of colony) represents the capacity of bacteria to fixate nitrogen or dissolve phosphate. Siderophore production was detected using Chrome Azurol-S (CAS) medium according to the method described by Louden[55]. Two microliter bacterial suspension was spotted on the CAS agar plates and incubated at 28 °C for 5 days. The ratio of the diameter of yellow-orange halo around bacteria (D) to the diameter of bacterial colony was considered as an indicator for siderophore production. Stains were inoculated into TSB medium supplemented with tryptophan (5 mmol/L) and incubated at 28 °C for 48 h on a rotary shaker. Bacterial suspension was centrifuged at 10,000 × $g$ for 5 min. One milliliter supernatant was transferred into EP tube containing equal volume of Salkowski reagent. After 30 min, the pink color of the reaction liquid indicated that the corresponding isolate was positive bacterium for producing IAA. The absorbance was determined at 530 nm by an ultraviolet or visible spectrophotometer. A standard curve was also made to calculate the amount of IAA produced by different bacteria[56]. The inhibitory effect on pathogens is referred to the above antagonistic experiment.

### Interactions between monocropping-depleted strains

Bacterial interactions were performed to determine if the depleted strains were mutually reinforcing or antagonistic[10]. *Paenibacillus* sp. R60, *Pantoea* sp. R05, *Lysinibacillus* sp. R06, *Enterobacter* sp. R09, *Sporosarcina* sp. R07, *Fictibacillus* sp. R37, and *Pseudomonas* sp. R26 were inoculated in 5 mL TSA and incubated overnight at 28 °C at 180 rpm. The optical density of the bacterial cultures was adjusted to 0.1 at 600 nm. 1 µL of these dilutions were inoculated seven times in a diagonal row on both sides of a square petri-dish with TSA agar with a multichannel pipet, creating a V-shape of increasingly closer inoculation sites. The plates were sealed with Parafilm and incubated for 2 days at 28 °C.

### Suppression of SynCom on *F. oxysporum*

To investigate whether the monocropping-depleted strains had protection effect on plant disease defenses, three different SynCom types, composed of 7, 4, or 2 depleted strains, were created. Only one combination was used for the SynCom composed of 7 depleted strains, while 35 and 21 combinations of strains were tested in the SynComs composed of 4 and 2 strains, respectively (Supplementary Table 2). Briefly, suspension of depleted strains was prepared by culturing in TSA medium and adjusted density to $OD_{600} = 0.1$ with sterile water. The suspensions of single depleted strains were mixed in equal volume to create a SynCom. Determination of the inhibitory effect of different SynComs on fungal hyphal growth was done by co-cultivation. First, 60 µL of SynCom suspension was inoculated evenly on a piece of sterile filter paper placed on one side of NA medium. After 2 d of co-cultivation at 28 °C, a PDA plug (8-mm diameter) containing fungal hyphae was placed 2 cm away from the filter paper. The growth of fungal hyphae was assessed after 72 h and compared with that on control plates, where sterile water was used instead of the bacterial suspension. The inoculated SynCom consisted of 7 inactivated depleted strains as the negative control. Each SynCom combination was tested in eight replicates.

### Suppression of *F. oxysporum* by SynCom combined with monocropping rhizosphere community in vitro

To determine whether the monocropping-depleted strains increased suppression of monocropping rhizosphere on pathogen, we determined the inhibitory effect of the addition of SynCom consisted of 7, 4, and 2 depleted strains on *F. oxysporum* (Supplementary Table 2). SynComs were prepared as described in the preceding subsection. Following the assay described in "metatranscriptome sequencing of cultivable rhizosphere microbiome", suspensions of rhizosphere microbiome were obtained and adjusted density to $OD_{600} = 1.0$. The SynCom (1 mL) was added to a suspension (2 mL) of rhizosphere microbiome from monocropped peanut. Inhibition of *F. oxysporum* by the rhizosphere microbiome with or without SynCom supplementation was examined. The inoculated SynCom consisted of 7 inactivated depleted strains as the negative control. Each treatment was tested in six replicates.

### Suppression of *F. oxysporum* by SynCom combined with monocropping rhizosphere community in vivo

The protective effect of depleted strain addition on the root and plant growth was assessed. Peanut seedlings were cultivated under sterile conditions[32]. First, surface-disinfected peanut seeds were placed on sterile moist filter paper for five days to germinate. Well-grown and uncontaminated seedlings were planted in 400-mL seedling hole (1 seedling per hole) containing sterile vermiculite. Fifteen 400-mL seedling holes were incubated in a plant growth chamber (30 °C, 70% relative humidity, light intensity 500 µM/m²s¹), covered with transparent lid to prevent microbial contamination. After 3 d of cultivation, 10 mL of SynCom and 10 mL of

monocropping microbiome suspension, or 10 mL of sterile water and 10 mL of monocropping microbiome suspension, were added to the vermiculite in 400-mL holes. Three days later, the vermiculite was inoculated using 10 mL of *F. oxysporum* spore suspension ($10^5$ CFU/mL) or sterile water. 50 mL of sterile Hoagland's nutrient solution (1/4 strength) was poured into the substrate to provide nutrients to the peanuts every 10 days. After 30 d of incubation, the disease incidence and plant growth status, i.e., height, fresh weight, and root length and weight, were determined.

### Biocontrol effect of SynCom application on peanut root rot in field experiment

The effect of different SynComs on plant health and growth was verified in a monocropped peanut field. The field experiment was set up at the Red Soil Ecological Experiment Station of the Chinese Academy of Sciences in Yingtan (Jiangxi Province, China). The control and 7-strain SynCom treatments were arranged in 4 plots (2 m × 2 m). For 4- and 2-strain SynComs, each combination was arranged in one plot, for a total of 64 plots. Before sowing, peanut seeds were soaked in a SynCom suspension for 12 h. Forty-five days after planting, all peanut plants were harvested from each plot to determine their health and growth status (Fig. 5g).

### Statistical analysis

All statistical analyses were performed in the R environment (v4.1.0, http://www.r-project.org/). Student's *t*-tests (two-sided) were performed to compare the statistical significance between pairs of samples and analysis of variance (ANOVA) to determine the statistical significance of multiple comparisons. Non-metric multidimensional scaling(NMDS) based on Bray–Curtis dissimilarity matrix was performed and plotted using the "vegan" package to describe the bacterial community under different treatments and at different sampling times. Differences in bacterial OTUs abundance were examined using DESeq2[57]. *P* values were corrected using the method of FDR. Based on the cultivable rhizosphere microbial community, monocropping-depleted OTUs were defined as OTUs not detected under the monocropping regime, and monocropping-enriched OTUs were defined as OTUs not detected under the rotation regime. Phylogenetic trees of bacterial rhizosphere isolates were constructed using MEGA v5, based on 16S rRNA sequences. Phylogenetic trees were annotated and visualized using iTOL software[58]. KEGG enrichment scatter plot analysis was performed using OmicStudio tools available at https://www.omicstudio.cn/tool/11.

### Reporting summary

Further information on research design is available in the Nature Portfolio Reporting Summary linked to this article.

## Data availability

The raw reads from Illumina sequencing and metatranscriptome generated in this study have been deposited in the NCBI database under accession no. PRJNA1029910 (https://www.ncbi.nlm.nih.gov/bioproject/PRJNA1029910) and PRJNA1029900 (https://www.ncbi.nlm.nih.gov/bioproject/PRJNA1029900), respectively. Additional data generated in this study are provided in the Supplementary Information/Source Data file. The data used for this study are available in 'figshare' with the identifier https://doi.org/10.6084/m9.figshare.24558613. For databases used in this study, SILVA database is available at https://www.arb-silva.de; UNITE database is available at https://unite.ut.ee/repository.php. Source data are provided with this paper.

## Code availability

Codes for statistical analyses are available in the Figshare database (https://doi.org/10.6084/m9.figshare.24433405.v1).

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

## Acknowledgements

This research was supported by the National Key Research and Development Program of China 2022YFD2201900 (Xi.L.), the National Natural Science Foundation of China 32122056, 42011045 (Xi.L.), and the earmarked fund for CARS-13 (X.W.). M.D-B. acknowledges support from TED2021-130908B-C41/AEI/10.13039/501100011033/Unión Europea NextGenerationEU/PRTR and from the Spanish Ministry of Science and Innovation for the I + D + i project PID2020-115813RA-I00 funded by MCIN/AEI/10.13039/501100011033.

## Author contributions

Y.Z. and Z.Y. conducted data curation, methodology, writing-original draft. Xi.L. conceived the project, designed the experiments, and writing-original draft. J.L. and Xu.L. implemented methodology. X.W., C.D., M.D.-B. and T.Z. review the manuscript. V.J.C., F.C., Z.W., and M.D.-B. edited the manuscript. All authors have discussed the results, read and approved the contents of the manuscript.

## Competing interests

The authors declare no competing interests.
