## [Peer Review File · Nature Communications]

Crop rotation and native microbiome inoculation restore soil capacity to suppress a root diseaseReviewers' comments:

Reviewer #1 (Remarks to the Author):

NCOMMS-22-44706

This work aimed to understand the disease biology of the different incidences of the 'root rot' causing agent *F. oxysporum* in peanut under different management regimes, i.e. monocropping and rotation with different crops. The more prominent results are that:

1. The rotation regime reduced the pathogens incidence and the associated microbial communities were studied to understand the origin of this effect

2. The metatranscriptome showed functions derived from these management regimes

3. Depleted strains in the monocrop regime reduced the *F. oxysporum* incidence

The manuscript's main findings are 1) that a rotation field management, rather than monocropping, reduces the pathogen incidence (Figure 2B). This finding has been extensively proved in the literature, but less understood is the mechanism of disease suppression by rotation. The disease suppression assay *in vitro* is ambiguously explained and presented, as commented below. 2) different Symcoms made up of bacteria taxonomically related to the bacteria depleted in the monocrop rhizosphere reduced the pathogen incidence in a number of taxa dependent manner i.e. the more taxa the better pathogen incidence control. I would emphasise this finding as represents the more interesting one. There is a lot of experimental effort, however, this is not presented, analysed and explained in the best way as described below.

General comments:

The introduction is too general and needs significant improvement, as does not include key concepts as the introduction to the system. For example, the host plant i.e. peanuts, the monoculture vs intercropping, the disease under study 'root rot' is not even explained, this is a general term that may refer to different diseases. The reader arrives at the result section ignoring essential information about the research.

In the results section the field experiments are split into two figures, consider dedicating a figure for field results and other for the pot/inhibition experiment as it is complicated to follow a rationale for the experiments. In some parts, the authors explained different taxonomic levels family, genus without a clear justification.

I am concerned about Figure 2D, there are no differences between different disease incidence, meaning that the disease phenotype has somehow disappeared by bringing the experiment to pots and the suppression assay is ambiguously explained in the main text and the caption.

Qiime is less common, however valid, these days compared to DADA2 pipeline; while the former clusters sequences at (typically) 97% identity into Operational Taxonomical Units (OTUs), the latter attempts to reconstruct the exact biological sequences present in the sample, so-called Amplicon Sequence Variants (ASVs). The reference databases for taxonomic information were not included. Furthermore, the differential abundance analysis was performed with a t-test, the microbial community differential abundance analysis of data is multivariate and of compositional nature (<https://microbiomejournal.biomedcentral.com/articles/10.1186/s40168-017-0237-y>, <https://www.frontiersin.org/articles/10.3389/fmicb.2017.02224/full>, <https://academic.oup.com/gigascience/article/8/9/giz107/5572529>).

In the results section culturable microbiome, there is a difficult to read mixture between the whole bacterial community (non-culturable) and the culturable bacteria microbiota as well as in figure 2. The rationale behind the exudate assay (Figure 4A) is unclear, perhaps using exudates from infected plants (with exudates of uninfected plants as control) and testing the effect on differentially abundant microbes, instead just only healthy plant exudates, would have been better justified

There is a poor description of the metatranscriptomic assay, particularly how this was conducted and analyzed. These types of analysis are not well established in the microbiome field, for example, how the per-sample abundance of each gene was calculated? How the gene abundance profiles were converted into functional profiles? How the impact of differential growth of bacterial strains has been corrected?

The discussion section is difficult to follow, I am afraid I could not properly review this section and just a few points were commented on. The discussion is in many instances not supported by references and shows no evidence for the stated arguments (see below 'Discussion section').

The general language and description of terms will benefit from substantial improvement. The different typos are perceived as a poor edition, of course, it is not a final version manuscript, but

does not give the best impression. The structure and how the experiments and figures are lined up is confusing and does not follow a logical progression e.g., the field experiment is separated in Fig 2 and Fig 3; in Figure 2 A and B are field experiments whereas C and D are culturable microbiome. The scripts and the input files used for the analyses should be uploaded to a publicly available repository, were not found as source file. Please, see in detail the comments below:

Title: The title does not mention the management or the system and the key taxa mentioned are vaguely identified in the manuscript

Abstract:

Line 26: typo health

Line 34: 'pest control', perhaps it is pathogen control instead as this research is based on *F. oxysporum*

Introduction:

Line 39: what plant phenotypes do the authors refer to?

Line 41: rhizosphere definition

Line 46: definition of compensatory colonization

Lines 48-49: what does 'the engineering of service function' means?

Line 51: Reference

Line 55: Reference, may be nice to provide some examples

Line 65: References

Line 74: References. In the same line, I would justify better my research question, there is plenty of research conducted for unveiling 'fingerprinting the roles and status of microbial populations in a community assembly, as well as their corresponding modes of action' there is more scope here to defend the novelty of your research. For example lines 101-104.

Line 79: Please mention the specific research carried out, different plant species, or more relevant references, do the authors mean plant varieties or plant species?

Lines 180-181: please specify what is the meaning 'governed by selection effects of plant root metabolism'. Not only does the plant's metabolism governs the microbiota assembly in the rhizosphere, see literature regarding immune system or root system architecture among these factors.

Line 81: 'host morbidity' may refer to a human condition.

Lines 81-84: the text does not mention the agricultural practice described.

Line 84-85: unclear the meaning of 'microbial taxa with sensitive to root metabolisms'

Line 96: Which microbial consortia are affected? What means 'unintended depletion'?

Line 100: please specify which discovered mechanisms underlying plant health.

Results:

Line 109: peanut scientific name

Line 110: provide the data or figure of low index

Line 111: With the advance of plant growth, the growth stage or time after planting, needs to be more concise with descriptions. Add the statistical test.

Line 113: When 'the index in monocrop peanut dramatically increased'?

Line 114: Specify the developmental stage you are describing here. Add the statistical test.

Line 118: how confident by BLAST results the authors are of the fungal species-level identification of fungi, particularly of the *Fusarium* group? Figure 2C

Line 119: culture-independent method

Line 124: I cannot understand where these numbers come from since in 2C figure the disease level range from -1.5 to 1.5. Are 0, 2, and 4 in the figure time points?

Line 125: please rephrase as is difficult to interpret. Also, different experiments, isolates, the whole community sequence, and the re-inoculation should be separated and properly introduced in different paragraphs.

Lines 122-128: are these fungal pathogens putatively the 'root rot'? Take into consideration that *F. oxysporum* is a cryptic species i.e., the same species may contain individuals of different strains with different ecological roles. For example, see:

Larkin RP, Hopkins DL, Martin FN (1996) Suppression of *Fusarium* wilt of watermelon by nonpathogenic *Fusarium oxysporum* and other microorganisms recovered from a disease suppressive soil. *Phytopathology* 86:812-819

The recolonization assay complemented the fungal quantification and demonstrate the pathogenicity

Figure?

Line 133: in the next developmental stages

Line 134: there is a jump in the text that is confusing, Figure 2B was explained in line 114, and the rest of the description is in line 134

Line 135: evident or significant? If I understood figure 2B correctly, the flowering stage is also significant. The figure information is incomplete in the caption, e.g., what represent the lines and what represent the bars.

Line 136: What means (3,8)? What indicates in the rotation regime?

Lines 139-141: this is not a valid interpretation as no rhizosphere data support it

Line 141: The ideal stage for intervention, well interpreted

Line 146: mention what differences

Line 151: briefly explain what these experiments consist of as it is essential to understand the section.

Lines 151-156: I could not understand this experiment. There is a little description. The inhibitory effect seems not to impact the disease level and it is not clear what microbiome was used for the inhibition, controls, etc.

Line 158: Reference

Line 161: In the field or the glasshouse experiment

Line 163: not found? Are they control of non-diseased plants?

Line 166: unclear sentence, likely incomplete?

Line 170: is the pathogen that inhibits the rhizosphere?

Line 173: alpha-diversity? Is significant?

Line 175: Figure?

Line 177: the main question in these results is not mentioned in the main text. Are differences in alpha diversity different between managing regimes?

Line 177: these stats do not feature in the figure. Are the numbers above the bars in extended Fig 1 p-values?

Line 180: Add the statistical test.

Line 182: figure? perhaps Figure 2A, but these specific stats do not feature there

Line 183: Figure?

Line 185: only the seedling stage analysis was found, is there a reason for this?

Line 186: belonging to the families...

Line 188: the differential abundance analysis was performed with a t-test, the microbial community differential abundance analysis must be because it is multivariate and it is compositional (<https://microbiomejournal.biomedcentral.com/articles/10.1186/s40168-017-0237-y> , <https://www.frontiersin.org/articles/10.3389/fmicb.2017.02224/full>, <https://academic.oup.com/gigascience/article/8/9/giz107/5572529>).

Line 194: see the previous comment (Line 188) on differential abundances analyses. Figure? What does this indicate as this is the base of subsequent experiments? This should feature as the main panel

Line 197-198: unclear sentence, please reformulate

Line 202-204: Are these the number of cultured bacteria? Were 6832 OTUs cultured or is this info for a sort of database, please clarify

Line 206: typo significant

Line 212: please refrain from this affirmation as this had not been tested, and causal relationships have not been proved. Which growing stage correspond to these data?

Line 213: were these taxa purified? Were these taxa clustered at 97% of similarity to name them OTUs? Otherwise, change the terminology

Line 214: avoid terms as mainly

Line 217: avoid terms as mainly

Line 222: were not pure cultures the previous paragraph taxa?

Line 224: which biochemical characteristics?

Line 232: Bacillus is a well-known biocontrol agent, perhaps is worthy to include

Line 239: nice approach

Line 242: peanut root exudates from monocrop peanuts? Is there a control with exudates from rotated peanuts?

Line 243: there is essential information missing as was performed in vitro, in pots?

Line 244: Sporosarcina, Lysinibacillus, Pseudomonas, and Fictibacillus are not significantly different according to Figure 4A

Line 248: What does all this indicate?

Line 254: Putative growth promotion properties, as growth promotion per se was not tested
Line 260: is this analysis performed with the field or pots? This might be miss-placed in this section
Line 262: vague interpretation, this information does not add value to the manuscript.
Line 269: how this experiment was carried out?
Line 272: in rotation or monoculture?
Line 274: what does this indicate?
Line 292: how this experiment was carried out, in plates?
Line 295: which strains?
Line 305: I would have included an additional control supplying the RhiCom with a random conformed Syncom
Line 310: poor description
Line 319: was the soil inoculated and after planted?
Line 333: References
Line 334: unclear sentence
Line 337: molecular?
Line 338: manipulate or impact
Line 340: unintended effect? The reviewer could not understand the meaning of this sentence
Line 334: References
Line 347: Where is the mechanistic model for the inhibitory effect? There is no mechanism of community assembly defined. Can this mechanism be explained? how we can use the information provided for other crops?
Line 355: at the seedling stage there were no significant differences in disease index (Fig 2 B)
Line 361: this implies that a 'not mature' microbiome (seedling stage) is not able to prevent infection, whereas a 'mature' assembly in successive developmental stages in monocrop or rotation triggers a different disease outcome
Line 362: likely? could not understand this sentence
Line 364: inhibit the pathogen
Line 371: this experiment was not conducted in roots
Line 371: are these compounds relevant to your investigation?
Line 379: exudates, for consistency. Does this experiment say something about management regimes or disease incidence?
Line 380: environmental filtering?
Line 382: which diversity parameters?
Line 390: where this was shown?
Line 405: depleted in the monocropping regime?
Line 417: colonization ability to interaction relationship?
Line 422: however were present under rotation, therefore able to colonize the rhizosphere environment
Line 428: was this shown in the metatranscriptomics?
Line 434: vigorous metabolism?
Line 441: minor microbes?
Line 454: in the field?
Line 466: cannot understand
Line 475: what means fundamental ecological patterns that govern microbes assembly?
Line 477: pest? The manuscript is about a pathogen
Methods:
Line 487: which crops?
Line 493: Please, briefly mention the management practices here
Lines 502: meaning that the bulk soil and the rhizospheres were collected at different times in the season?
Line 506: bulk soil and rhizosphere?
Line 508: primers used and qPCR program. Refer where to find this information
Line 509: it is not clear if the root does have not soil particles attached with a gentle wash of water that will interfere with the root fresh weight (RFW). Root dry weight is a more accurate measurement
Line 509: Rhizobium?
Line 522: are these primers published? please cite
Line 530: The fungal sequences have a dedicated database UNITE, both databases may be used

for fungal identification. E-values and hit consistency in BLAST outputs need to be mentioned for taxonomical identification of Fungi, as NCBI contains miss-annotated sequences.

Lines 531-532: the phylogenetic tree of the culturable fungal communities? Where it is located?

Line 532: pathogen presence, pathogenicity?

Line 533: all fungi were inoculated, in monocultures, in communities? please indicate the concentration of the inoculants, where this experiment was conducted, field/glasshouse? how the rhizosphere was inoculated?

Line 541: are these primers published? please cite

Line 547: This is rather sample preparation

Line 553: How the rhizosphere was extracted?

Line 558: Reference

Line 562: More details on library preparation need to be provided, for example, were the reactions made in triplicate, pooling strategy, etc.?

Line 565: this platform has been surpassed by QIIME2.

Line 570: Databases used for taxonomy assignment are not mentioned

Line 574: In the current experiment. Plots or pots. When was this experiment carried out i.e., how long was time pass between the soil harvesting and the pot experiment?

Line 579: wow that is nasty. I wonder if the mercury persists in the seeds after successive rinses.

Line 587: I would say this is a biological replicate. This is confusing as the caption of Figure 2 is mentioned n=3

Line 591: fresh weight

Line 594: Are they pictures of these assays? it would be nice as a supplementary figure

Line 603: because you were interested in bacterial communities only?

Line 605: is it volatile suppression?

Line 610: total volume?

Line 621: NA medium?

Line 635: the septum?

Line 641: i.d.

Line 650: Reference

Line 663: Reference

Line 666: Sanger sequenced

Lines 669-671: unclear, were the isolate taxonomy compared with the 16S information of the entire rhizosphere community?

Line 676: all strains survive in a rhizosphere environment. The bacterial strains are depleted in different management regimens, not in a rhizosphere environment per se

Line 681: Reference format

Line 685: with just NB medium as mock control?

Lines 712: Figure?. Please, indicate how these assays were performed

Line 716: Metatranscriptome. has the putative host contamination been removed? How the per-sample abundance of each gene was calculated? How the gene abundance profiles were converted into functional profiles? How the impact of differential growth of bacterial strains has been corrected

Line 749: Please mention the Syncom composition

Line 751: strain typo. Mention the Syncom composition

Line 774: Indoor?

Line 814: References

Captions:

Line 1010: what are lines and what bars?

Line 1028: impossible to read

Figures:

Fig 1: may feature as a supplementary figure.

Fig. 2: A) The picture in fig 2A has a no control healthy roots to compare. B) What is represented in bars and what in lines? C) panel contains two figures, and the pathogen disease representation is unclear. Typo 'strains'. D) is difficult to know how the figure was produced as commented somewhere above these lines.

Fig. 3:

Fig. 4: A) Exudates and biofilm are two panels. In exudates, the y-axis 'Quantity change' as a title is unclear and impairs the interpretation of the results. Also, the differentially abundant taxa are impacted by the plant exudates, it is unclear the rationale behind this experiment as is explained

above. C) difficult to read.

rotation More efficient would be if strains enriched and depleted are differentiated in the figure.

Fig. 5: F), G) multiple figures in the same panel

Extended Data:

Extended Data Fig. 1: it seems unclear if the number above bars are p-values e.g. Line 4: ' the bars indicate statistically significant differences, perhaps more precise to indicate bars indicate significance. Individual replicates may be represented by points.

Extended Data Fig. 3: this is a valuable piece of information

Extended Data Fig. 4: Unclear the units of the y-axis, perhaps the number of bacteria isolates at Genus level. B panel is located after C panel. B

Extended Data Fig. 6: Weight is fresh or dry? What is the difference between root length and root height? There is no representation of individual points.

Extended Data Fig. 7: the authors are referring to the same with plant height and shoot height (see Extended Data Fig.6). Stains? Unclear what the control is, not specified in the caption. Why the number of replicates is so variable i.e. from 4 to 35? No differences with control treatment? In the plot shoot height is expressed in grams, perhaps is this a type and the author's mean weight, is that dry or fresh? What is the difference between shoot length and shoot height? What is the difference between root length and root height?

Extended Data Table 3: unclear the inhibitory characteristics of *F. oxysporum*

Reviewer #2 (Remarks to the Author):

Zhou et al aim to identify which microbial taxa drive suppression of root rot disease caused by fungal pathogens in peanut plants. They do this using a range of in vitro and in field experiments, where they compare different growth stages of peanut grown in rotation and monocropping. The authors report an increased susceptibility to the disease under a monocropping regime compared to growth in soil with a rotation regime. Next, they try to dig into the microbial cause for this phenomenon by studying microbial communities using culture-dependent and -independent techniques and by examining microbial volatile compounds.

This type of studies can increase our knowledge on the mode of action of alternative agricultural practices that can be used to protect crops from pathogens in a sustainable way.

Despite its interest and relevance, the manuscript requires significant text editing that will benefit how the experiments and results are communicated with the reader. Furthermore, some statements need to be toned down or supported by new experiments. Please see below:

Major comments

1. The abstract and introduction should be more focused and include information critical for the current study. For instance, it would be useful to mention peanut root rot, its known causal agents or known ability of certain microbes to protect against it. Another example is in Lines 81-84, where it should be evident if the practice the authors refer to is rotation. As it is now, it is hard to connect parts of the introduction with the actual research performed.

2. Several phrases need re-wording to be clearer such as Lines 42-43 and Lines 84-89. Also together with text edits, some extra experimental evidence is needed to strengthen some messages:

a. Lines 164-167: In this sentence it is not clear what the connection is between the identity of the VOCs and that the pathogenic fungi is a prerequisite, since it is supposed to be present in both types of soil.

b. Lines 175-176: Please elaborate on how these two things are connected

c. Lines 197-198: The authors should not conclude about the sensitivity of the microbes to peanut root metabolites since it was not tested.

d. Lines 240-248: Please clarify that this growth does not refer to colonization on root, but to quantification for growth on the sterile water and the media, which is not the same as rhizosphere colonization.

e. Line 249: *Pantoea* shows an increase as well. Since what the authors show is fold change, all bacteria display increase in biofilm formation in response to exudates compared to control

treatment, not only *Stenotrophomonas*.

f. Line 292: The authors should increase the controls used in this experiment to be able to conclude that this phenomenon is explained by this set of depleted strains alone. For instance, they could utilize also random strains to construct a SynCom or heat kill the mixture of bacteria used in the 7-member SynCom.

g. Line 160: really strong statement that VOCs can affect infection, if some of the differential VOCs between rotation and monocropping are not tested against the pathogen, in a concentration dependent-manner.

h. Line 255: To increase the impact of this section, the authors should divide in taxa between enriched in monocropping and rotation and then highlight what properties they possess and if they could be relevant for disease suppression.

i. Line 311: The authors could consider the relevance of evaluating the interactions between these 7 microbes? are they affecting growth of each other? or biofilm? please see experiments in <https://www.nature.com/articles/s41396-018-0093-1> for ways to assess SynCom behavior vs individual strains. Also composition of different SynComs tested should be shown in a Table.

3. In several parts of the text there is relevant information missing or mislocated:

a. Line 799: Please indicate the figure where this is described.

b. Lines 120-124: This information is not shown in figure 2C because that is about the cultivable fungi.

c. Line 125-128: Please indicate where this information is shown.

d. Line 142: Please indicate if you refer to bulk or rhizosphere soil.

e. Lines 143-145: Did the authors assess disease incidence in the other growth stages? That would help to know whether disease development was consistent with what was observed in the field. As it is right now, it reads incomplete, since we don't see if the difference at this stage can determine how disease develops.

f. Line 673: Indicate in the text how this information is also laid out in Fig 3C. Also, please include in the text information about the lowest and highest sequence similarity that you used to consider two isolates the same.

g. Line 701: The authors should indicate if the Hoagland remained sterile and whether plants were stressed after this long incubation without changing to fresh medium.

h. Line 150 and Figure 2D: Please clarify what the data refers to because in the figure it says it's part of the pot experiment but in the text it says it's in vitro.

i. Lines 180-181: Please indicate to which growth stages you are referring to and include the results for each of the growth stages.

j. Line 315: Please clarify if this conidia production is measured in vitro or in pots. The current titles in Figure 5 are misleading.

4. The section about culturable microbiome and rhizosphere microbiome is not clearly written
Line 201-221: In this section what was sequenced with which technology needs to be clarified because it's hard to follow what is the culturable microbiome and what not. The authors should more clearly explain the different experiments where Illumina Sequencing was used because it is confusing at the moment. If these strains were isolates (200 isolates according to line 659) why are they talking about more than 6000 OTUs? How many OTUs would be deriving from one isolate? If this is the case, it should be displayed. However, in Line 662 they mention full length 16s rRNA primers, much longer than what Illumina MiSeq can sequence, so please clarify which technology was used specifically for which section. Line 205 Please indicate the growth stage that this belongs to. Line 207 The number of bacteria shown in Fig 4 is much lower, how is this related to the 6832 OTUs mentioned at the beginning of the paragraph?

Minor comments

Introduction

1. Line 40 "activating nutrient transformation" is unclear, it would be more appropriate to say they can aid plant nutrition.

2. Line 48 With service function, do you mean ecosystem services?

3. Line 53 What do you mean that the functions have been isolated from the rhizosphere microbiome?

4. Line 58-62 These two phrases sound contradictory. To what do you refer with structural stability? And how is it that functional performance is expanded but "its difficult to achieve the

function of the entire microbiome”?

5. Line 60 Why is reference 2, two times?

6. Line 79 references <https://www.science.org/doi/full/10.1126/science.1203980>
<https://www.science.org/doi/full/10.1126/science.aaw9285> are appropriate for this statement.

7. Line 84-87 This phrase is unclear, if the taxa is stable, it would be expected to be less “sensitive” to environmental cues, sensitivity would lead to changes in its abundance.

8. Lines 90-91 This is not a hypothesis, this is known for coumarins, benzoxazinoids, glucosinolates, triterpenes. Please see <https://www.pnas.org/doi/10.1073/pnas.1722335115> ,
<https://www.nature.com/articles/s41467-018-05122-7> ,
<https://www.nature.com/articles/ismej200968>, etc.

Methodology

1. Line 499-500: Please indicate the five-class scale in the text and fix the spelling of *Fusarium* in the citation.

2. Line 508: Here the authors should already specify the pathogen they refer to (*Fusarium* (?)).

3. Lines 515, 681 and 760: Please specify the year of these publications, and add the reference number as it is in the rest of the manuscript.

4. Lines 526 to 529 and where this applies: please remove the space between the number and the Celsius symbol in temperatures.

5. Line 547: Please separate sample processing and DNA extraction from the bioinformatic analysis.

6. Line 565 and other parts of methods: The authors should cite the references from the bioinformatic tools used e.g. QIIME.

7. Line 635: It is not clear what the septum is. Is there maybe something missing in between these two phrases?

Results

1. Line 150: Please refer to figure's numbers and subfigures in the text in a manner consistent with the actual Figures e.g., “2D” instead of “2d”

2. Line 151: Please make this sentence more clear to people lacking the background, for instance saying *Fusarium* inhibition bioassays.

3. Line 172: Not clear from Figure 1 that sampling for microbiome analysis took place in 2018.

4. Lines 261-264: Not clear what the contribution of this analysis is to the story.

Figures

Figure 1: Please indicate why the word “maturing” is surrounded by a red square only in the rotation crop image (surrounded by a blue square).

Figure 2C and 5E: “Isolated strains” and “strains”.

Extended Data

Figure 1: Please indicate if this is related to bacterial or fungal communities in the legend of the figure.

Table 3: Fix “Siderophore”.

Reviewer #3 (Remarks to the Author):

In this paper, the authors investigated the difference between peanut plants grown in rotation and monocropping. They performed a field experiment and monitored disease. They did not find any difference in disease in the field experiment. However, they still decided to investigate many different parameters to try to understand what is happening in peanut grown in rotation and monocropping. They found that the rhizosphere bacterial communities from monocropping and rotation plots are different. They collected bacteria from the rhizosphere and characterised their collection

in terms of phylogeny, biofilm production and response to exudates.

The authors performed an enormous amount of experiments and used many different techniques. They used Illumina sequencing to characterise the rhizosphere of several experiments, they isolated microbes, they analyzed VOC, they analyzed exudates and they even used metatranscriptomics! However, I have strong concerns about this manuscript. First of all, none of the experiments were repeated. While I understand this is difficult for some experiments like the field experiment or the isolation of bacteria, it is quite feasible for the pot experiments. Second, the number of replicate per treatment is 3 which is on the low side for this kind of study. Third, some of the side experiments (VOCs for example) are not described with enough detail to convince the reader that this experiment can actually answer the tested hypothesis.

I find the introduction very difficult to read. Partly because of the English, which is definitely not good enough, but also because the references are not always well-chosen and the structure is not clear.

Lots of English mistakes, too many to list here. Please have the manuscript checked by native speaker.

When reading the results section, I was missing the connection between different subsections. Why was this experiment performed? To test which hypothesis? How was it performed?

Some experiment are not described correctly. For example, the abundance of *Fusarium* is quantified by sequencing (lines 129-135) but the results reported are in copies/g dry soil, which appears to be qPCR results.

The information about which community was sequencing with Illumina is missing. Is it the bacterial community? the fungal community? Also, the motivation for focusing on the bacterial community should be stated.

The authors performed also some pot experiments. According to Figure 1, there were actually 10 pots, but in Figure 2D the number of replicates was 3. Especially for a pot experiment, it is possible to have more replicates.

I did not read the discussion as I feel this paper is not yet matured enough.

Detailed comments

RESULTS (only section commentated with line edits)

line 109: rotation regime => with what crop? This is an important information which should be stated in the results section as well as the figure legend.

line 113: 2.3-2.6 higher : please state the stage where this was found. Is this a confidence interval? a range? please be more precise.

line 115: "we isolated 38 fungi": how many strains did you obtain for each species? add the numbers in the barplot.

line 119: start a new paragraph with the results of the illumina sequencing as this is a new approach. Describe what you did in a few words. Please also report some results of the quality of the sequencing (as supplemental: how many sampled were sequenced, how many sequences per sample were obtained) as well as rarefaction plots.

line 122: "F. oxysporum associated with OTU177 " what do you mean with these words? I assume that you found sequences of OTU177 (taxonomically assigned *Fusarium oxysporum*) to be more

abundant in the diseasead-pea rot, please be more precise, the word "associated" means something else.

line 124: "34.1 and 2712.7 times more abundant" how are these numbers calculated?

Figure 2 Pannel C: the right half (OTU177 and OTU90) does not fit the left half which is labeled with a C and which shows the cultivable Fungi. It seems this panel with OTU177 and OTU90 needs its own letter. Also, what does the disease level 0, 2, and 4 correspond to?

Line 125: again, start a new paragraph with the results of the pot experiment. Describe in a few words how the pot experiment was performed.

The lines 125-128 do not correspond to any figure as far as I can tell. Also, the experiment is not really explained. What was the goal?

line 130: "specific primers" targeting which organisms?

line 130 talks about sequencing but the figure 2b seems to present results of qPCR

line 136: why present result of ANOVA and then student's t test on line 138? In this case, since this treatment only has two levels, a t-test makes sense.

line 141-146: how was this pot experiment performed? what does it mean to do a rotation pot experiment?

Extended Data Table 2: what do the numbers in the table represent? Please provide T and P values instead of letters.

Line 151 : "Directed and inverted culture experiments": what does this mean

line 157: and Figure 2d: in the legend, it is written that the VOC are produced by cultural bacteria, which bacteria, how were they isolated, how many isolates, which medium?

Figure 3B: the title of that panel is "cultivable microbiome" which is misleading as the this Figure presents results of Illumina sequencing.

Line 172: Illumina sequencing: of the bacterial or of the fungal community? please state in a few words the approach that was chosen.

line 173: "a downward trend" referring to what? this sentence should refer to a specific figure or panel. Also write more clearly "alpha diversity" as the concept "diversity" includes also beta diversity (and even gamma!).

line 178-180: ordination cannot reveal significant differences. If you would like to test for differences, please use a statistical test, e.g. PERMANOVA as in the following sentences.

line 186-188: "enriched in the mono cropped plant rhizosphere" compared to what? please state the other side of comparison.

Extended Data Fig. 1

what is the number above the bar? is it a P-value?

please label the y-axis of each plot

lines 188-191: similar "recruited to the rhizosphere compared to what"

Extended Data Fig. 2 legend should also explain in more detail what is the experimental factor tested here. STAMP is a program, please refer to the statistical analysis performed by the

program. Are the P-values corrected for multiple testing? and if yes, with what method?

We are actually not so much interested in which bacteria are more abundant in rotation vs bulk soil and which bacteria are more abundant in monoculture vs bulk soil but more interested in which bacteria are more abundant in monoculture vs rotation.

line 199 Culturable

line 201 cultivable

please chose one word and stick to it. Personally, I prefer cultivable.

line 201. start this new section with one sentence which describes why you decided to focus on cultivable bacteria, and how you performed this isolation approach.

also, how did you sequence with Illumina the cultivable microbiome?

line 202-203: it is strange that for this library you offer the range of sequences and the number of OTUs, information which you did not provide for the other libraries. Please be consistent, provide that information for all the sequencing approaches (for example in a supplemental table).

Extended Data Fig. 3 don not use box-plot with only three points. please provide labels for y-axis.

I am very surprised that the P-values are all > 0.05 especially for Shannon.

line 212: "we then determined OTUs of culturable strains depleted" what does this mean?

line 222 : what does "purified" mean?

lines 241-248: i am very confused about this experiment. The author mention exudates, but they report abundance of different bacteria. What is the hypothesis tested with this experiment? How was it performed? Similarly, how was the biofilm experiment performed?

line 260-261;: Co-occurrence network analysis: which of the dataset was used to construct this network? It is not interesting to just list that there were negative and positive correlations. this is obvious. I would just drop this figure entirely.

line 265: "changes inculturable microbiome function"

again, briefly state, what is the goal of this experiment? how was it performed?

lines 292-295: Good! this is exactly what I mean. Each new section should start with the question or hypothesis and then explain briefly how the experiment was performed.

Response to reviewers' comments

Reviewer #1 (Remarks to the Author):

NCOMMS-22-44706

This work aimed to understand the disease biology of the different incidences of the 'root rot' causing agent *F. oxysporum* in peanut under different management regimes, i.e. monocropping and rotation with different crops. The more prominent results are that:

1. The rotation regime reduced the pathogens incidence and the associated microbial communities were studied to understand the origin of this effect
2. The metatranscriptome showed functions derived from these management regimes
3. Depleted strains in the monocrop regime reduced the *F. oxysporum* incidence

The manuscript's main findings are 1) that a rotation field management, rather than monocropping, reduces the pathogen incidence (Figure 2B). This finding has been extensively proved in the literature, but less understood is the mechanism of disease suppression by rotation. The disease suppression assay *in vitro* is ambiguously explained and presented, as commented below. 2) different Symcoms made up of bacteria taxonomically related to the bacteria depleted in the monocrop rhizosphere reduced the pathogen incidence in a number of taxa dependent manner i.e. the more taxa the better pathogen incidence control. I would emphasis this finding as represents the more interesting one. There is a lot of experimental effort, however, this is not presented, analysed and explained in the best way as described below.

Answer: We are appreciated your compliments on our study and make full sense of your comments and concerns on polishing our manuscript. During preparing the revised manuscript for fully addressing your comments, we have conducted more additional experiments, such as the effects of root exudates from diseased and health peanuts on growth of enriched and depleted microbes in monocropping rhizosphere, and the help of different synthetic communities composed of depleted strains to restore pathogen suppression of monocropping rhizosphere community. The results from these additional experiments have been appended in the revised manuscript. Referring to your point on the unclarity of disease suppression assay *in vitro*, we have revised the corresponding parts including in both result and method sections for more clearly. We hope that all concerns raised by the reviewer are now well addressed.

General comments:

The introduction is too general and needs significant improvement, as does not include key concepts as the introduction to the system. For example, the host plant i.e. peanuts, the monoculture vs intercropping, the disease under study 'root rot' is not even explained, this is a general term that may refer to different diseases. The reader arrives at the result section ignoring essential information about the research.

Answer: We are sorry for not specifically explaining our present study in the introduction section of original manuscript. Based on your suggestions, we have re-organized the introduction and added essential information on why focusing peanuts of the monoculture vs rotation, and the characteristic of 'root rot' in peanut in the revised manuscript (lines 58-68).

In the results section the field experiments are split into two figures, consider dedicating a figure for field results and other for the pot/inhibition experiment as it is complicated to follow a rationale for the experiments. In some parts, the authors explained different taxonomic levels family, genus without a clear justification.

Answer: Thanks for your valuable suggestions. The results of the field and pot experiments have been arranged two separate figures (Figures 2 and 4 in the revised manuscript). In addition, we analyzed the differences in rhizosphere bacterial community composition between monocropping and rotation seedlings at the OTU level.

I am concerned about Figure 2D, there are no differences between different disease incidence, meaning that the disease phenotype has somehow disappeared by bringing the experiment to pots and the suppression assay is ambiguously explained in the main text and the caption.

Answer: In order to make paper easier to understand, we removed Figure 2D from the revised manuscript. We are sorry for not clearly explaining the suppression assay. Referring to the reviewer's comment, we have added more detailed information for the pathogen suppression assay in the main text and the captions on the corresponding results were revised as well (lines 200-205).

Qiime is less common, however valid, these days compared to DADA2 pipeline; while the former clusters sequences at (typically) 97% identity into Operational Taxonomical Units (OTUs), the latter attempts to reconstruct the exact biological sequences present in the sample, so-called Amplicon Sequence Variants (ASVs). The reference databases for taxonomic information were not included.

Answer: Regarding the difference between OTU and ASV, we agree with you. Clustering as ASV based on a sequencing error may be dismissed if some species with very low abundance are present in one sample. For our present study, species with low abundance were enabled to access rhizosphere community assembly in rotation peanut rather than that of monocropped peanut, which were an important hypothesis for determining plant status to resist pathogen invasion, thus clustering as OTU were performed to assemble the potential depleted microbes in monocropped peanut rhizosphere as compared with rotation peanut. In addition, referring to the reviewer's suggestion, reference databases, i.e., "... using UNITE database (v7.0) for fungi" and "... using SILVA reference database (v12.8) for bacteria", have been added in the revised manuscript. (lines 452 and 510).

Furthermore, the differential abundance analysis was performed with a t-test, the

microbial community differential abundance analysis of data is multivariate and of compositional nature (<https://microbiomejournal.biomedcentral.com/articles/10.1186/s40168-017-0237-y>, <https://www.frontiersin.org/articles/10.3389/fmicb.2017.02224/full>, <https://academic.oup.com/gigascience/article/8/9/giz107/5572529>).

Answer: We agree with you that community differential abundance is multivariate and compositional. Differences in bacterial OTUs abundance were examined using DESeq2. *P* values were corrected using the method of FDR.

In the results section cultivable microbiome, there is a difficult to read mixture between the whole bacterial community (non-cultivable) and the cultivable bacteria microbiota as well as in figure 2.

Answer: We are sorry for unclear distinction of the whole bacterial community (non-cultivable) in field experiment from the cultivable microbiome of pot experiment. In the field experiment, the whole bacterial community (non-cultivable) in rhizosphere of monocropping and rotation peanuts were first analyzed by Illumina sequencing. For the pot experiment, we tested the pathogen suppression of rhizosphere cultivable microbiota from peanut seedlings, and cultivable microbiomes of monocropping and rotation peanuts were parallelly analyzed by Illumina sequencing. This would provide substantial information of rhizosphere cultivable microbiome between monocropping and rotation peanuts, contributing to recognize the potential depleted strains in monocropped peanut rhizosphere.

Referring to the reviewer's comment, the results for cultivable microbiome have been removed to supplementary files, and we added relevant information for clear identification on analyzing bacterial community (non-cultivable) of peanut rhizosphere in field experiment (lines 176-196), and Figure 2 has been arranged to focus on whole bacterial community (non-cultivable) of field experiment in the revised manuscript (lines 176-196).

The rationale behind the exudate assay (Figure 4A) is unclear, perhaps using exudates from infected plants (with exudates of uninfected plants as control) and testing the effect on differentially abundant microbes, instead just only healthy plant exudates, would have been better justified.

Answer: We appreciated your constructive comments. Referring to your suggestion, we have conducted additional experiments to collect root exudates of peanuts that were planted in monocropping and rotation soils, and tested the response of depleted strains to root exudates of both infected and healthy plants. This was an important reason for taking such long time to prepare the revised manuscript. The corresponding results and experiment method have been appended in the revised manuscript. (lines 295-308).

There is a poor description of the metatranscriptomic assay, particularly how this was conducted and analyzed. These types of analysis are not well established in the microbiome field, for example, how the per-sample abundance of each gene was

calculated? How the gene abundance profiles were converted into functional profiles?
How the impact of differential growth of bacterial strains has been corrected?

Answer: To date, there are technical challenges to directly perform metatranscriptomic sequencing on soil environment microbiome *in situ*, as it is difficult to extract high quality of mRNA from soils to meet the standard of metatranscriptomic sequencing. Thus, for extracting high quality of mRNA from microbiome combined with experimental feasibility, we collected rhizosphere cultivable microbiome growing on the plates to extract total RNA. The detailed methods have been described in Online Methods section (lines 566-578).

For the bioinformation analysis on metatranscriptomic microbiome that the reviewer concerned, detail information, e.g., “Gene expression levels were determined by RNA sequencing (RNA-seq) as transcripts Per Kilobase of exon model per Million mapped reads (TPM).”, “All genes in the catalogue were translated to amino acid sequences and aligned with data in the Kyoto Encyclopedia of Genes and Genomes (KEGG) database.” have been added in the revised manuscript (lines 589-594).

The difference in cultivable microbiomes between monocropping and rotation should be attributing to the interaction between bacterial strains and their position in the community. Therefore, the impact of differential growth of bacterial strains would be generated as consequence of the whole cultivable microbiomes that were why we focused on performing metatranscriptome sequencing of cultivable rhizosphere microbiome.

The discussion section is difficult to follow, I am afraid I could not properly review this section and just a few points were commented on. The discussion is in many instances not supported by references and shows no evidence for the stated arguments (see below ‘Discussion section’).

Answer: Thanks for your valuable suggestions. To enhance the logic and readability of the manuscript, we merged the results and discussion

The general language and description of terms will benefit from substantial improvement. The different typos are perceived as a poor edition, of course, it is not a final version manuscript, but does not give the best impression. The structure and how the experiments and figures are lined up is confusing and does not follow a logical progression e.g., the field experiment is separated in Fig 2 and Fig 3; in Figure 2 A and B are field experiments whereas C and D are cultivable microbiome. The scripts and the input files used for the analyses should be uploaded to a publicly available repository, were not found as source file. Please, see in detail the comments below:

Answer: We are appreciated your comments on polishing our manuscript and make full sense of your concerns on our study. The language of the manuscript has been revised by a native speaker editor (Joanna Mackie, email: joanna.potrykus.mackie@gmail.com), and we further invited a microbiologist (Delgado-Baquerizo, who have published many papers in *Nature* and *Nature-sister*

journals) with close research on the present study to polish our manuscript. We are very guilty for some typos in the original manuscript, and we have carefully checked the terms throughout the whole manuscript for avoiding other mistakes. For the arrangement of the figures, we have structured them referring to your comments to follow a logical progression. The original files of the Illumina sequencing have been uploaded to the NCBI (<https://www.ncbi.nlm.nih.gov/bioproject/PRJNA860278>), but some of them will be released at a set time.

Title: The title does not mention the management or the system and the key taxa mentioned are vaguely identified in the manuscript.

Answer: Referring to the reviewer's comments, the title has been revised to "Management and microbiome inoculation bring back the capacity of soils to support disease suppression".

Abstract:

Line 26: typo health

Answer: We are very sorry for the error. This has been corrected.

Line 34: 'pest control', perhaps it is pathogen control instead as this research is based on *F. oxysporum*

Answer: Thanks for your suggestion. "pest control" has revised to "pathogen control". (line 42)

Introduction:

Line 39: what plant phenotypes do the authors refer to?

Answer: Referring to the reviewer's comment, "plant phenotypes" has been revised to "plant growth". (line 42)

Line 41: rhizosphere definition

Answer: The information, "The rhizosphere refers to the soil directly associated with the roots. This complex environment, enriched with carbon and nutrients, is home to a diverse microbial community that plays an essential role in promoting plant growth and health." has been added to define rhizosphere. (lines 47-49)

Line 46: definition of compensatory colonization

Answer: In order to enhance the readability of the article, this word has been omitted from the revised manuscript.

Lines 48-49: what does 'the engineering of service function' means?

Answer: The sentence has been revised to "Mechanistic understanding of the variation and assembly characteristics of rhizosphere microbiome would provide innovative strategies for improving plant health and productivity". (lines 47-49)

Line 51: Reference

Answer: Done.

Line 55: Reference, may be nice to provide some examples

Answer: In order to enhance the readability of the article, this sentence has been omitted from the revised manuscript.

Line 65: References

Answer: Done.

Line 74: References. In the same line, I would justify better my research question, there is plenty of research conducted for unveiling ‘fingerprinting the roles and status of microbial populations in a community assembly, as well as their corresponding modes of action’ there is more scope here to defend the novelty of your research. For example lines 101-104.

Answer: This has been revised to “Yet, how management by regulating the impact of rhizosphere microbial communities on peanut root rot diseases remains largely unknown.”. (lines 68-69)

Line 79: Please mention the specific research carried out, different plant species, or more relevant references, do the authors mean plant varieties or plant species?

Answer: In order to enhance the readability of the manuscript, this sentence has been omitted from the revised manuscript.

Lines 80-81: please specify what is the meaning ‘governed by selection effects of plant root metabolism’. Not only does the plant’s metabolism governs the microbiota assembly in the rhizosphere, see literature regarding immune system or root system architecture among these factors.

Answer: Referring to the reviewers’ comment, the sentence has revised to “Recent studies have showed that rhizosphere microbiome assembly is significantly influenced by the selection effects of plant root metabolisms. As a result, different outcomes of plant health can be fostered depending on which microbial populations are able to take advantage of the root metabolites”. (lines 74-77)

Line 81: ‘host morbidity’ may refer to a human condition.

Answer: In order to enhance the readability of the manuscript, this sentence has been omitted from the revised manuscript.

Lines 81-84: the text does not mention the agricultural practice described.

Answer: We added “Crop management is known to have a critical role in regulating rhizosphere microbiomes. In China, peanut represents one of the most profitable crops. However, the yield and quality of this product has been reported to be greatly compromised by soil-borne diseases, especially under intensive monocropping managements. In subtropical China, 10-40% of crop yield is lost due to increasing disease pressure, with peanut root rot caused by fungal pathogens being a major

constraint. Unfortunately, few effective control measures are available, and the use of chemical pesticides is limited by environmental concerns. Interestingly, we also know that crop rotations including different crop varieties can mitigate the negative impacts of pathogens on crop production, becoming a non-expensive alternative for disease control...". (lines 58-68)

Line 84-85: unclear the meaning of 'microbial taxa with sensitive to root metabolisms'

Answer: The sentence has been revised to "As a result, different outcomes of plant health can be fostered depending on which microbial populations are able to take advantage of the root metabolites. Thus, agricultural practices could potentially exploit the differences in root metabolisms of various crops to disrupt the directional selection for maintaining the stability of agricultural ecosystem". (lines 76-80)

Line 96: Which microbial consortia are affected? What means 'unintended depletion'?

Answer: The sentence has been revised to "Our findings revealed that under peanut monocropping, there was a selective enrichment of bacteria that exhibited positive responses to peanut root exudates, but also significant losses of other bacteria showing weaker responses to root exudates." (lines 87-90)

Line 100: please specify which discovered mechanisms underlying plant health.

Answer: The sentence has been revised to "Our findings revealed that under peanut monocropping, there was a selective enrichment of bacteria that exhibited positive responses to peanut root exudates, but also significant losses of other bacteria showing weaker responses to root exudates. Losses in these bacteria associated with the rhizosphere microbiome resulted in reductions in the capacity of rhizosphere to suppress pathogen invasion". (lines 105-106)

Results:

Line 109: peanut scientific name

Answer: Latin name of peanut has been added (line 115).

Line 110: provide the data or figure of low index

Answer: The data of low index has been provided in line 128.

Line 111: With the advance of plant growth, the growth stage or time after planting, needs to be more concise with descriptions. Add the statistical test.

Answer: Referring to the reviewer's comments, the sentence has been revised to "However, from flowering stage, the DI in peanuts under monocropping regime dramatically increased, and was 2.6 times higher than that of peanuts sown under the rotation regime at pod-bearing stage". Statistical test was also added. (line 130)

Line 113: When 'the index in monocrop peanut dramatically increased'?

Answer: “from flowering stage” has been added. (line 130)

Line 114: Specify the developmental stage you are describing here. Add the statistical test.

Answer: “at pod-bearing stage” and the corresponding statistical test have been added. (line 132-133)

Line 118: how confident by BLAST results the authors are of the fungal species-level identification of fungi, particularly of the *Fusarium* group? Figure 2C

Answer: The similarity of the isolated strains with BLAST identification was more than 99%, and the details for the BLAST results have been added as Extended Data Fig. 1. Thus, these fungi can be classified at species-level identification. In addition, NCBI is an authoritative database of microbial sequencing, and some studies also employed BLAST method for fungal identification (Zhou, X. et al. 2022).

Zhou, X. et al. (2022) Cross-kingdom synthetic microbiota supports tomato suppression of *Fusarium* wilt disease. *Nat. Commun.* 13, 7890.

Line 119: culture-independent method

Answer: Thanks for your suggestion. We deleted it.

Line 124: I cannot understand where these numbers come from since in 2C figure the disease level range from -1.5 to 1.5. Are 0, 2, and 4 in the figure time points?

Answer: For the description in Line 124, these numbers indicated the difference in relative abundance of OTU177 and OTU90 between diseased and healthy peanuts, as this was the results of Illumina sequencing on peanut roots. For Figure 2C, “-1.5 to 1.5” represents the relative abundance of normalized OTU.” These descriptions have been added in lines 981-982 of the revised manuscript for more clarity.

Line 125: please rephrase as is difficult to interpret. Also, different experiments, isolates, the whole community sequence, and the re-inoculation should be separated and properly introduced in different paragraphs.

Answer: Referring to the reviewer’s comments, this part has been rephrased for clearly interpreting the relevant results. We would like to further explain that this paragraph mainly focused on the work of pathogen identification, which included different experiment of isolates, the whole community sequence, and the re-inoculation. (lines 136-156)

Lines 122-128: are these fungal pathogens putatively the ‘root rot’? Take into consideration that *F. oxysporum* is a cryptic species i.e., the same species may contain individuals of different strains with different ecological roles. For example, see:

Larkin RP, Hopkins DL, Martin FN (1996) Suppression of *Fusarium* wilt of watermelon by nonpathogenic *Fusarium oxysporum* and other microorganisms recovered from a disease suppressive soil. *Phytopathology* 86:812–819

Answer: We agree with you that *Fusarium oxysporum* containing pathogenic and non-pathogenic strains. In order to determine whether *F. oxysporum* isolated from peanut roots had pathogenicity on peanut, we conducted a recolonization experiment. Our results showed that the individuals of *F. oxysporum* that were isolated from peanut root rots can infect peanut roots, and with higher pathogenicity than the isolates of *F. solanum*. This was summarized in Fig. 2E.

Line 133: in the next developmental stages

Answer: We have made the modification according to your suggestion.

Line 134: there is a jump in the text that is confusing, Figure 2B was explained in line 114, and the rest of the description is in line 134

Answer: Thanks for your suggestion. For keeping continuity, we have separated the results of disease index and *F. oxysporum* abundance in two figures (Fig. 2C, 2F), and their description was shown in order. (lines 119 and 143)

Line 135: evident or significant? If I understood figure 2B correctly, the flowering stage is also significant. The figure information is incomplete in the caption, e.g., what represent the lines and what represent the bars.

Answer: This sentence aimed to indicate the variation of *F. oxysporum* abundance in peanuts sown in rotation across different growth stages, rather than comparison of monocropping and rotation. In order to enhance the readability, this sentence has been omitted from the revised manuscript. In addition, the figure has been formatted as two separated figures (Fig. 2C, 2F).

Line 136: What means (3,8)? What indicates in the rotation regime?

Answer: ANOVA analysis was performed for *F. oxysporum* abundance in the rhizosphere of rotation at different growth stages, with 3 and 8 representing inter-group and intra-group degrees of freedom, respectively. This result indicated rhizosphere community in the rotation regime had higher suppression on pathogen infection.

Lines 139-141: this is not a valid interpretation as no rhizosphere data support it

Answer: Thanks for your suggestion. The sentence has been revised to "Therefore, pathogen accumulation in the rhizosphere after seedling stage indicated that the failure of the rhizosphere to resist soil pathogen invasion takes place early in the plant growth, since the proliferation of pathogens that were identified likely as *Fusarium oxysporum* in the present study takes time on pathogenicity enhancement". (lines 167-172)

Line 141: The ideal stage for intervention, well interpreted.

Answer: Thanks for your encouragements.

Line 146: mention what differences

Answer: In order to enhance the logic and readability of this manuscript, we have deleted it.

Line 151: briefly explain what these experiments consist of as it is essential to understand the section.

Answer: For more clarity, this was revised to “To examine the role of rhizosphere community in the occurrence of peanut root rot, we further collected soils from field plots of both monocropping and rotation to grow peanut at a greenhouse experiment (Fig. 1C). The rhizosphere microbiome of peanuts at seedlings was harvested to determine the ability of *F. oxysporum* inhibition. Antifungal activity against *F. oxysporum* were tested in in VOC-mediated and directed microcosm antagonism assays to evaluate antifungal activity.”. (lines 200-205)

Lines 151-156: I could not understand this experiment. There is a little description. The inhibitory effect seems not to impact the disease level and it is not clear what microbiome was used for the inhibition, controls, etc.

Answer: Antagonism effect of rhizosphere microorganisms is a critical protective strategy against pathogen invasion, thus VOC-mediated and directed microcosm antagonism assays were employed to determine whether microbial community assembly in plant rhizosphere suppressed pathogen invasion. Directed microcosm antagonism assays represents the ability of rhizosphere community on suppressing pathogen growth by producing soluble metabolites. VOC-mediated assays represents the indirect effect of rhizosphere community on suppressing pathogen growth by producing volatile organic compounds. These two antagonism assays have been employed in several previous studies:

Li X.G. et al. (2020) Volatile-mediated antagonism of soil bacterial communities against fungi. *Environmental Microbiology* 22(3), 1025–1035.

Carrión V.J. et al. (2019) Pathogen-induced activation of disease-suppressive functions in the endophytic root microbiome. *Science* 366(6465): 606-612

Carrión V. J. et al. (2018) Involvement of Burkholderiaceae and sulfurous volatiles in disease-suppressive soils. *ISME J* 12: 2307-2321.

Line 158: Reference

Answer: Done.

Line 161: In the field or the glasshouse experiment

Answer: “... at a greenhouse experiment.” has been added. (line 202)

Line 163: not found? Are they control of non-diseased plants?

Answer: Yes, these compounds were not found in the rhizosphere microbiome of monocropped peanut. To determine if they were responsible for controlling non-diseased peanuts, we conducted additional experiments using their standard substances purchased from company (Sigma-Aldrich LLC.). These results showed that these compounds had high suppression on pathogen development even at low

concentrations, which are shown in Extended Data Fig. 4.

Line 166: unclear sentence, likely incomplete?

Answer: We are sorry for the mistake. This has been omitted from the revised manuscript.

Line 170: is the pathogen that inhibits the rhizosphere?

Answer: For more clarity, the sentence has been revised to “We then investigated the impacts of crop managements on the bacterial microbiome of peanut rhizosphere at different plant growth stages and bulk soil in our 2018 field experiment.”. (lines 177-179)

Line 173: alpha-diversity? Is significant?

Answer: In order to enhance the logic and readability of the manuscript, this has been omitted from the revised manuscript.

Line 175: Figure?

Answer: The results of the Illumina sequencing in the field experiment are shown in Figure 3.

Line 177: the main question in these results is not mentioned in the main text. Are differences in alpha diversity different between managing regimes?

Answer: In order to enhance the logic and readability of the manuscript, this has been omitted from the revised manuscript.

Line 177: these stats do not feature in the figure. Are the numbers above the bars in extended Fig 1 p-values?

Answer: In order to enhance the logic and readability of the manuscript, this has been omitted from the revised manuscript.

Line 180: Add the statistical test.

Answer: Done.

Line 182: figure? perhaps Figure 2A, but these specific stats do not feature there

Answer: Yes, the specific stats have been added in Fig. 3A of the revised manuscript.

Line 183: Figure?

Answer: Fig. 3A, this has been added. (line 182)

Line 185: only the seedling stage analysis was found, is there a reason for this?

Answer: One sentence was added: “The difference of rhizosphere microbiome occurred from seedling stage, consisting with previous research showing that difference in community composition can predict disease earlier than pathogen density. Therefore, we compared the variation of rhizosphere bacterial community

between monocropping and rotation at seedling stage.”. (lines 182-186)

Line 186: belonging to the families...

Answer: Done.

Line 188: the differential abundance analysis was performed with a t-test, the microbial community differential abundance analysis must be because it is multivariate and it is compositional

(<https://microbiomejournal.biomedcentral.com/articles/10.1186/s40168-017-0237-y>, <https://www.frontiersin.org/articles/10.3389/fmicb.2017.02224/full>, <https://academic.oup.com/gigascience/article/8/9/giz107/5572529>).

Answer: We agree with you that community differential abundance is multivariate and compositional. Differences in bacterial OTUs abundance were examined using DESeq2. *P* values were corrected using the method of FDR. The corresponding results were revised as well.(lines 177-196)

Line 194: see the previous comment (Line 188) on differential abundances analyses. Figure? What does this indicate as this is the base of subsequent experiments? This should feature as the main panel

Answer: We analyzed the differences in rhizosphere bacterial community composition between monocropping and rotation, and the corresponding results have been removed to the main panel (Fig. 3), referring to the reviewer's comment.

Line 197-198: unclear sentence, please reformulate

Answer: We analyzed the differences in rhizosphere bacterial community composition between monocropping and rotation, and the corresponding results were revised as well. (lines 177-196)

Line 202-204: Are these the number of cultured bacteria? Were 6832 OTUs cultured or is this info for a sort of database, please clarify

Answer: Yes, this is the result of Illumina sequencing on cultivable microbiome. Now we have supplemented these methods (lines 691-704) and modified the number of OTUs after deleting the OTUs with sequence of 0 in all samples.

Line 206: typo significant

Answer: This has been omitted from the revised manuscript.

Line 212: please refrain from this affirmation as this had not been tested, and causal relationships have not been proved. Which growing stage correspond to these data?

Answer: Referring to the reviewer's comment, the sentence has been revised to "Considering the difference in inhibition ability of the rhizosphere microbiome against *F. oxysporum* on agar plates between monocropping and rotation, we first compared the characteristics of the whole cultivable microbiome by Illumina sequencing.”.(lines 253-256) These data were derived from cultivable microbiome at

seedling stage in pot experiments (Details in method section, lines 628-636).

Line 213: were these taxa purified? Were these taxa clustered at 97% of similarity to name them OTUs? Otherwise, change the terminology.

Answer: No, this is the result of cultivable microbiomes, and sequences are clustered based on 97% similarity. This sentence has been deleted in order to enhance the readability of the manuscript.

Line 214: avoid terms as mainly

Answer: We modified “mainly” to “most”. (line 261)

Line 217: avoid terms as mainly

Answer: We modified “mainly” to “most”. (line 264)

Line 222: were not pure cultures the previous paragraph taxa?

Answer: Yes, the previous paragraph describes Illumina sequencing results for cultivable microbiome. This paragraph describes the results of the isolation and purification of bacteria. For more clarity, one sentence was added: “In order to obtain depleted and enriched strains in monocropping rhizosphere, ...”. (line 269)

Line 224: which biochemical characteristics?

Answer: Sorry, this had some redundancy, and has been omitted.

Line 232: Bacillus is a well-known biocontrol agent, perhaps is worthy to include

Answer: Thank you for your enlightening suggestion. Yes, Bacillus is a well-known biocontrol agent, but in our present study there were abundant Bacillus growing on agar plats of both monocropping and rotation (Extended Data Fig. 5), thus indicating that Bacillus may not be potential agents responsible for the difference in pathogen suppression of rhizosphere microbiome between monocropping and rotation regime.

Line 239: nice approach

Answer: Thanks for your encouragements.

Line 242: peanut root exudates from monocrop peanuts? Is there a control with exudates from rotated peanuts?

Answer: Yes, root exudates were collected from monocropping peanuts. Referring to your suggestion, we have conducted additional experiments to collect root exudates of peanuts that were planted in monocropping and rotation soils, and tested the response of depleted strains to root exudates of both infected and healthy plants. This was an important reason for taking such long time to prepare the revised manuscript. The corresponding results and experiment method have been appended in the revised manuscript. (lines 294-307).

Line 243: there is essential information missing as was performed in vitro, in pots?

Answer: This has been revised to “To understand whether the depleted of low abundance of taxa is associated with responsiveness to plant root exudates, we collected root exudates of peanuts grown on monocropping and rotation soil, and evaluated the effects of root exudates on bacterial growth *in vitro*.”. (lines 295-298)

Line 244: Sporosarcina, Lysinibacillus, Pseudomonas, and Fictibacillus are not significantly different according to Figure 4A

Answer: We have conducted additional experiments to test the response of depleted strains to root exudates of both infected and healthy plants and described according to the new results. (lines 298-302)

Line 248: What does all this indicate?

Answer: We have added the information: “Accordingly, by failing response to root exudates of monocropped peanut, depleted taxa are thereby unable to effectively participate in community assembly in the monocropping rhizosphere.”. (lines 302-305)

Line 254: Putative growth promotion properties, as growth promotion per se was not tested

Answer: This has been revised to “potential growth-promoting properties “. (line 308)

Line 260: is this analysis performed with the field or pots? This might be miss-placed in this section

Answer: “in vitro” has been added. (line 309)

Line 262: vague interpretation, this information does not add value to the manuscript.

Answer: This information has been deleted.

Line 269: how this experiment was carried out?

Answer: “We extracted mRNA of cultivable microbiome from monocropping and rotation rhizosphere after the VOCs-mediated antagonism assay, and transcribed it into cDNA for sequencing.” has been added. (lines 232-234)

Line 272: in rotation or monoculture?

Answer: This has been revised to “The metabolic pathways that were primarily responsible for the differences in KO functional categories in the monocropping and rotation included ABC transporters and Two-component system”. (lines 237-239)

Line 274: what does this indicate?

Answer: This information has been deleted.

Line 292: how this experiment was carried out, in plates?

Answer: Yes, “co-cultured on a plate with *F. oxysporum*” has been added. (line 340)

Line 295: which strains?

Answer: This is suspension of rhizosphere microbiome from monocropped peanuts with synthetic communities composed of depleted strains. Now, it has been omitted from the revised manuscript.

Line 305: I would have included an additional control supplying the RhiCom with a random conformed Syncom

Answer: Referring to your suggestion, we have conducted additional experiments to test the effects of supplying the RhiCom with a random conformed Syncom on *F. oxysporum* pathogenicity. This was an important reason for taking such long time to prepare the revised manuscript. The corresponding results and experiment method have been appended in the revised manuscript. (lines 341-346)

Line 310: poor description

Answer: This has been revised to “Further, we tested the compensatory effect of the depleted strains in experiments with sterile peanut seedlings”. (line 352)

Line 319: was the soil inoculated and after planted?

Answer: “Peanut seeds were inoculated with SynComs composed of 7, 4, or 2 strains by soaking..” has been added. (lines 364)

Line 333: References

Answer: Done.

Line 334: unclear sentence

Answer: We merged the results and discussion, therefore this sentence has been omitted from the revised manuscript.

Line 337: molecular?

Answer: We merged the results and discussion, therefore this word has been omitted from the revised manuscript.

Line 338: manipulate or impact

Answer: We merged the results and discussion, therefore this sentence has been omitted from the revised manuscript.

Line 340: unintended effect? The reviewer could not understand the meaning of this sentence

Answer: This has been revised to “Our work further anticipates that, compared with rotation managements, long-term monocropping strategies weaken the capacity of soils to prevent the entrance of pathogens in the soil, and the promote the development of soil-borne pathogen diseases.”. (lines 100-103)

Line 334: References

Answer: Done.

Line 347: Where is the mechanistic model for the inhibitory effect? There is no mechanism of community assembly defined. Can this mechanism be explained? how we can use the information provided for other crops?

Answer: This has been revised to “Thus, our study highlights the role of management to fight against microbial disease, and further provide evidence on how we can actively restore rhizosphere to promote soil disease suppression.”. (line 107-110)

Line 355: at the seedling stage there were no significant differences in disease index (Fig 2 B)

Answer: At seedling stage, there was no significant difference in disease index between monocropping and rotation. From the flowering stage, the disease index of monocropping was significantly higher than rotation. Therefore, we speculated that changes in the rhizosphere of monocropping peanut seedlings may have caused the difference in the disease index after seedling stage.

Line 361: this implies that a 'not mature' microbiome (seedling stage) is not able to prevent infection, whereas a 'mature' assembly in successive developmental stages in monocrop or rotation triggers a different disease outcome

Answer: This has been revised to “Taken together, our results suggested that the effective colonization of *F. oxysporum* in the peanut rhizosphere has critical consequences for plant health in crops subjected to monocropping.”.(lines 171-174)

Line 362: likely? could not understand this sentence

Answer: To make it easy to understand, this word has been omitted from the revised manuscript.

Line 364: inhibit the pathogen

Answer: “resist” has been revised to “inhibit”.(line 204)

Line 371: this experiment was not conducted in roots

Answer: “Antifungal activity against *F. oxysporum* were tested in VOC-mediated and directed microcosm antagonism assays to evaluate antifungal activity.” has been added. (lines204-205)

Line 371: are these compounds relevant to your investigation?

Answer: This has been revised to “For instance, sulfur compounds and long-chain ketones produced by many rhizobacteria effectively suppress fungal pathogens”. (lines 215-216)

Line 379: exudates, for consistency. Does this experiment say something about

management regimes or disease incidence?

Answer: “secretions” has been revised to “exudates”. “Plant root exudates are one of the sources that regulate rhizosphere microbial assembly. To understand whether the depleted of low abundance of taxa is associated with responsiveness to plant root exudates, we collected root exudates of peanuts grown on monocropping and rotation soil, and evaluated the effects of root exudates on bacterial growth *in vitro*” has been added. (lines 294-298)

Line 380: environmental filtering?

Answer: In order to enhance the readability of this manuscript, this sentence has been omitted from the revised manuscript.

Line 382: which diversity parameters?

Answer: This has been revised to “Nonmetric multidimensional scaling (NMDS) based on the Bray–Curtis dissimilarity matrix revealed significant differences in bacterial community of monocropping and rotation at all development stages”.(lines 180-182)

Line 390: where this was shown?

Answer: In order to enhance the readability of this manuscript, this sentence has been omitted from the revised manuscript.

Line 405: depleted in the monocropping regime?

Answer: Yes. These OTUs exist in the rotation rhizosphere and the monocropping rhizosphere is 0, so we define them as depletion OTUs in monocropping.(lines 782-785)

Line 417: colonization ability to interaction relationship?

Answer: This has been revised to “To understand whether the depleted of low abundance of taxa is associated with responsiveness to plant root exudates, we collected root exudates of peanuts grown on monocropping and rotation soil, and evaluated the effects of root exudates on bacterial growth *in vitro*.” (lines 295-298)

Line 422: however were present under rotation, therefore able to colonize the rhizosphere environment

Answer: This has been revised to “Accordingly, by failing response to root exudates of monocropped peanut, depleted taxa are thereby unable to effectively participate in community assembly in the monocropping rhizosphere.”. (lines 302-305)

Line 428: was this shown in the metatranscriptomics?

Answer: In order to enhance the readability of this manuscript, this sentence has been omitted from the revised manuscript.

Line 434: vigorous metabolism?

Answer: This has been revised to “Accordingly, by failing response to root exudates of monocropped peanut, depleted taxa are thereby unable to effectively participate in community assembly in the monocropping rhizosphere.” (lines 302-305)

Line 441: minor microbes?

Answer: This has been revised to “low abundance microbes”. (lines 349)

Line 454: in the field?

Answer: In order to enhance the readability of this manuscript, this sentence has been omitted from the revised manuscript.

Line 466: cannot understand

Answer: This has been revised to “Conversely, specific microbial taxa that drive resident microbial community assembly can amplify rhizosphere community functions.”. (lines 375-376)

Line 475: what means fundamental ecological patterns that govern microbes assembly?

Answer: This has been revised to “In conclusion, we here uncovered differences in the assembly of rhizosphere microbial communities caused by agricultural management practices and the resulting disease consequences, and provided a new potential strategy for an efficient and sustainable biological disease control.”. (lines 383-386)

Line 477: pest? The manuscript is about a pathogen

Answer: This has been revised to “provide a new potential strategy for an efficient and sustainable biological disease control”. (lines 386)

Methods:

Line 487: which crops?

Answer: This has been revised to “rotation, peanut was grown first (2012), and then maize (*Zea mays* L.), potato (*Solanum tuberosum*), and soybean (*Glycine max*) were ordinarily planted in every other peanut planting year.”. (lines 397-399)

Line 493: Please, briefly mention the management practices here

Answer: “Commonly used management practices, including tillage, fertilizer application, and weed control, were applied manually” has been added. (lines 401-403)

Lines 502: meaning that the bulk soil and the rhizospheres were collected at different times in the season?

Answer: Rhizosphere samples were just collected at different times across the season rather than for bulk soil. The sentence has been revised for clarity. (lines 416-424)

Line 506: bulk soil and rhizosphere?

Answer: More experiment details were added to the revised manuscript. (lines 416-424)

Line 508: primers used and qPCR program. Refer where to find this information

Answer: We deleted this sentence. Primers and qPCR procedures as well as references are mentioned in the section "Determination of pathogen abundance in peanut rhizosphere". (lines 488-496)

Line 509: It is not clear if the root does have not soil particles attached with a gentle wash of water that will interfere with the root fresh weight (RFW). Root dry weight is a more accurate measurement

Answer: Thank you for your suggestion. Unfortunately, we did not measure the root dry weight (RDW). However, in order to get the RFW as accurately as possible, we have washed the root with sterile water 6-8 times before measuring the fresh weight, and sucked up the remaining water with absorbent paper.

Line 509: Rhizobium?

Answer: Yes, root nodule, a specific tumor growing on peanut root, was counted.

Line 522: are these primers published? please cite

Answer: Done.

Line 530: The fungal sequences have a dedicated database UNITE, both databases may be used for fungal identification. E-values and hit consistency in BLAST outputs need to be mentioned for taxonomical identification of Fungi, as NCBI contains miss-annotated sequences.

Answer: We constructed a phylogenetic tree of fungi and provided E-values and hit consistency in BLAST outputs. The results are presented in Extended Data Fig. 1.

Lines 531-532: the phylogenetic tree of the cultivable fungal communities? Where it is located?

Answer: We added phylogenetic tree in Extended Data Fig. 1.

Line 532: pathogen presence, pathogenicity?

Answer: This has been revised to "Determination of pathogenicity of potential pathogen". (lines 476)

Line 533: all fungi were inoculated, in monocultures, in communities? please indicate the concentration of the inoculants, where this experiment was conducted, field/glasshouse? how the rhizosphere was inoculated?

Answer: "The precipitated spores were re-suspended with sterile water, and the concentration of spore suspension was adjusted to 10^9 CFU/mL. To confirm

pathogenicity of potential pathogens, peanuts were planted in glasshouse (30°C, 70% relative humidity, light intensity 500 $\mu\text{M m}^{-2} \text{s}^{-1}$). 10 mL spore suspension of *F. oxysporum* or *F. solanum* were poured to 14 day-old peanuts rhizosphere, and the incidence of disease was recorded 30 d after inoculation.” has been added. (lines 481-486)

Line 541: are these primers published? please cite

Answer: Done.

Line 547: This is rather sample preparation

Answer: New section “Sample collection and processing” has been added. (lines 415)

Line 553: How the rhizosphere was extracted?

Answer: “Excess soil on the roots was discarded by gently shaking the plants, and the remaining soil particles attached to the root surface were collected as rhizosphere soil.” has been added. (lines 419-421)

Line 558: Reference

Answer: Done.

Line 562: More details on library preparation need to be provided, for example, were the reactions made in triplicate, pooling strategy, etc.?

Answer: Yes, “All samples were pooled in equimolar concentrations and then sequenced with a paired-end protocol at Majorbio Bio-Pharm Technology Co. Ltd. (Shanghai, China) using the Illumina MiSeq platform, according to the manufacturer’s instructions” has been added. (lines 441-444)

Line 565: this platform has been surpassed by QIIME2.

Answer: Yes, QIIME platform has been surpassed by QIIME2. Regarding the difference between OTU and ASV, we agree with you. However, clustering as ASV may be dismissed as a sequencing error if some species with very low abundance are present in the sample. Low abundance species are important in our study, therefore we chose the cluster as OTU.

Line 570: Databases used for taxonomy assignment are not mentioned

Answer: “Taxonomic assignment was performed using SILVA reference database (v12_8) for bacteria.” has been added. (lines 510)

Line 574: In the current experiment. Plots or pots. When was this experiment carried out i.e., how long was time pass between the soil harvesting and the pot experiment?

Answer: This has been revised to “soil samples were collected for pot cultivation experiments after 2018 peanut planting season, which prevented disorganizing the field plot experiment”. (lines 515-517)

Line 579: wow that is nasty. I wonder if the mercury persists in the seeds after successive rinses.

Answer: Mercury chloride as a common form of disinfection, was used to disinfect crop seeds. Continuous rinsing can ensure that no mercuric chloride remains in the seeds. The method is adopted widely to sterilize seeds (Gammoudi.2021; Ahmad et al. 2019; Si et al. 2014).

Gammoudi, N. K. et al. (2022) Establishment of optimized in vitro disinfection protocol of *Pistacia vera* L. explants mediated a computational approach: multilayer perceptron-multi-objective genetic algorithm. BMC Plant Biol. 22.

Ahmad, H. V. et al. (2020) Enhanced biosynthesis synthesis of copper oxide nanoparticles (CuO-NPs) for their antifungal activity toxicity against major phyto-pathogens of apple orchards. Pharmaceutical research. 37(12).

Si, Y., Haxim, Y. & Wang, L. Optimum sterilization method for in vitro cultivation of dimorphic seeds of the succulent halophyte *Suaeda aralocaspica*. Horticulturae. 8(2022).

Line 587: I would say this is a biological replicate. This is confusing as the caption of Figure 2 is mentioned n=3

Answer: We changed the 3 to 30 (lines 520-522).

Line 591: fresh weight

Answer: Done.

Line 594: Are they pictures of these assays? it would be nice as a supplementary figure

Answer: Yes, they are the pictures of these assays. Thanks for your encouragements.

Line 603: because you were interested in bacterial communities only?

Answer: Yes. Bacterial communities in plant rhizosphere were reported to determine the suppression on fungal pathogen infection, and thus rhizosphere bacterial community at different plant growth stages from field experiments was first focused on Illumina sequencing.

Line 605: is it volatile suppression?

Answer: Yes, this has been revised to “For the VOC-mediated antifungal activity”. (lines 541)

Line 610: total volume?

Answer: Yes, it was the total volume.

Line 621: NA medium?

Answer: This has been revised to “TSA”.(line 557)

Line 635: the septum?

Answer: “septum” has been revised to “dish lid”. (line 598)

Line 641: i.d.

Answer: This has been revised to “internal diameter”. (line 604)

Line 650: Reference

Answer: Done.

Line 663: Reference

Answer: Done.

Line 666: Sanger sequenced

Answer: We have corrected the word.

Lines 669-671: unclear, were the isolate taxonomy compared with the 16S information of the entire rhizosphere community?

Answer: This has been revised to “To confirm the existence of these bacterial isolates in the peanut rhizosphere, similarity analysis was performed based on the 16S rRNA sequence of the bacterial isolates and gene sequences of OTUs (same genus with isolates).”. (lines 651-654)

Line 676: all strains survive in a rhizosphere environment. The bacterial strains are depleted in different management regimens, not in a rhizosphere environment per se

Answer: Referring the reviewer’s comment, this has been revised to “To investigate why certain strains isolated from the rotation peanut rhizosphere were depleted from the monocropped peanut rhizosphere, their response to root exudates of monocropping and rotation peanuts were determined.”.(lines 658-660)

Line 681: Reference format

Answer: Done.

Line 685: with just NB medium as mock control?

Answer: The response of monocropping depleted and enriched strains to root exudates of monocropping and rotation peanut was studied.(lines 658-674)

Lines 712: Figure?. Please, indicate how these assays were performed

Answer: The results were exhibited in Extended Data Table 1.

Line 716: Metatranscriptome. has the putative host contamination been removed? How the per-sample abundance of each gene was calculated? How the gene abundance profiles were converted into functional profiles? How the impact of differential growth of bacterial strains has been corrected

Answer: “Gene expression levels were determined by RNA sequencing (RNA-seq) as

transcripts Per Kilobase of exon model per Million mapped reads (TPM). All genes in the catalogue were translated to amino acid sequences and aligned with data in the Kyoto Encyclopedia of Genes and Genomes (KEGG) database” has been added. (lines 588-593)

Line 749: Please mention the Syncom composition

Answer: Syncom composition has been added in Extended Data Table 2.

Line 751: strain typo. Mention the Syncom composition

Answer: Thanks. The word has been corrected, and Syncom composition has been added in Extended Data Table 2.

Line 774: Indoor?

Answer: “Suppression of *F. oxysporum* by SynCom combined with monocropping rhizosphere community *in vitro*” has been revised. (lines 730)

Line 814: References

Answer: Done.

Captions:

Line 1010: what are lines and what bars?

Answer: The results of disease index and *Fusarium oxysporum* abundance are shown in Fig. 2C and Figure. 2F respectively.

Line 1028: impossible to read

Answer: This has been revised to “The relative abundance of depleted or enriched OTUs in cultivated microbial communities of monocropping and rotation. The nine depleted or enriched OTUs with > 97% similarity to isolates are shown.” in Fig. 4E.(lines 1004-1006)

Figures:

Fig 1: may feature as a supplementary figure.

Answer: Considering the large number of experiments involved in our study, the flow chart of the experimental design has been retained in the revised manuscript in order to make it easier to understand.

Fig. 2: A) The picture in fig 2A has a no control healthy roots to compare. B) What is represented in bars and what in lines? C) panel contains two figures, and the pathogen disease representation is unclear. Typo ‘strains’. D) is difficult to know how the figure was produced as commented somewhere above these lines.

Answer: A) control healthy roots has been added in Fig. 2A.

B) The results of disease index and *Fusarium oxysporum* abundance are shown in Fig. 2C and Figure. 2F respectively.

C) Fungal isolates and OTUs enriched in diseased roots are shown in Extended Data

Fig. 1 and Fig. 2D, respectively.

D) In order to enhance the logic and readability of this paper, the disease index of the pot experiment has been omitted from the revised manuscript.

Fig. 4: A) Exudates and biofilm are two panels. In exudates, the y-axis 'Quantity change' as a title is unclear and impairs the interpretation of the results. Also, the differentially abundant taxa are impacted by the plant exudates, it is unclear the rationale behind this experiment as is explained above. C) difficult to read.

rotation More efficient would be if strains enriched and depleted are differentiated in the figure.

Answer: A) The results of bacterial growth are shown in Fig. 4F. y-axis has been revised to "OD_{600nm}".

C) This has been revised to "Enrichment analysis of differential genes of culturable microorganisms in rhizosphere of monocropping and rotation." in Fig.4C.

Fig. 5: F), G) multiple figures in the same panel

Answer: Fig. 5F and 5G have been divided to Fig. 5D and 5H.

Extended Data:

Extended Data Fig. 1: it seems unclear if the number above bars are p-values e.g. Line 4: the bars indicate statistically significant differences, perhaps more precise to indicate bars indicate significance. Individual replicates may be represented by points.

Answer: In order to make the paper more logical, these results have been omitted from the revised manuscript.

Extended Data Fig. 3: this is a valuable piece of information

Answer: This figure has been moved to Fig. 3.

Extended Data Fig. 4: Unclear the units of the y-axis, perhaps the number of bacteria isolates at Genus level. B panel is located after C panel. B

Answer: The units of the y-axis is the number of bacteria isolates at Genus level. Now, this information is presented in Extended Data Fig. 6.

Extended Data Fig. 6: Weight is fresh or dry? What is the difference between root length and root height? There is no representation of individual points.

Answer: This was fresh weight. We unified root length and root height and added the representation of individual points.

Extended Data Fig. 7: the authors are referring to the same with plant height and shoot height (see Extended Data Fig.6). Stains? Unclear what the control is, not specified in the caption. Why the number of replicates is so variable i.e. from 4 to 35? No differences with control treatment? In the plot shoot height is expressed in grams, perhaps is this a type and the author's mean weight, is that dry or fresh? What is the

difference between shoot length and shoot height? What is the difference between root length and root height?

Answer: “shoot height, root length, shoot weight, and root weight” has been revised. We added “Control: without inoculating the strain.”.

Extended Data Table 3: unclear the inhibitory characteristics of *F. oxysporum*

Answer: “Except *Pantoea* and *Lysinibacillus*, the other depleted strains inhibited the growth of *F. oxysporum*” has been revised.(lines 313-314)

Reviewer #2 (Remarks to the Author):

Zhou et al aim to identify which microbial taxa drive suppression of root rot disease caused by fungal pathogens in peanut plants. They do this using a range of in vitro and in field experiments, where they compare different growth stages of peanut grown in rotation and monocropping. The authors report an increased susceptibility to the disease under a monocropping regime compared to growth in soil with a rotation regime. Next, they try to dig into the microbial cause for this phenomenon by studying microbial communities using culture-dependent and -independent techniques and by examining microbial volatile compounds.

This type of studies can increase our knowledge on the mode of action of alternative agricultural practices that can be used to protect crops from pathogens in a sustainable way.

Despite its interest and relevance, the manuscript requires significant text editing that will benefit how the experiments and results are communicated with the reader. Furthermore, some statements need to be toned down or supported by new experiments. Please see below:

Answer: We are appreciated your compliments on our study and make full sense of your comments and concerns on polishing our manuscript. During preparing the revised manuscript for fully addressing your comments, we have conducted more additional experiments, such as the effects of root exudates from diseased and health peanuts on monocropping enriched and depleted microbes, and the help of different synthetic communities composed of depleted strains to restore suppression of monocropping rhizosphere community on pathogen. The results from these additional experiments have been appended in the revised manuscript. We hope that all concerns raised by the reviewer are now well addressed. For the structure of manuscript especially in the result section, we have rearranged the text descriptions based on reviewer’s suggestion for better readability. Furthermore, some statements in the manuscript have been toned down to clearly deliver the relevant information.

Major comments

1. The abstract and introduction should be more focused and include information critical for the current study. For instance, it would be useful to mention peanut root rot, its known causal agents or known ability of certain microbes to protect against it. Another example is in Lines 81-84, where it should be evident if the practice the authors refer to is rotation. As it is now, it is hard to connect parts of the introduction

with the actual research performed.

Answer: We are sorry for not specifically explaining our present study in the introduction section of original manuscript. Based on your suggestions, we have re-organized the introduction and added essential information on why focusing peanuts of the monoculture vs rotation, and the characteristic of 'root rot' in peanut in the revised manuscript (lines 58-69).

2. Several phrases need re-wording to be clearer such as Lines 42-43 and Lines 84-89. Also together with text edits, some extra experimental evidence is needed to strengthen some messages:

a. Lines 164-167: In this sentence it is not clear what the connection is between the identity of the VOCs and that the pathogenic fungi is a prerequisite, since it is supposed to be present in both types of soil.

Answer: This sentence has been deleted.

b. Lines 175-176: Please elaborate on how these two things are connected

Answer: Thanks for your comment. We deleted this sentence.

c. Lines 197-198: The authors should not conclude about the sensitivity of the microbes to peanut root metabolites since it was not tested.

Answer: Referring to the reviewer's suggestion, we compared the differences in rhizosphere microbial composition of monocropping and rotation seedlings, and added "The difference of rhizosphere between monocropping and rotation of low abundance taxa indicated that they played an important role in peanut root rot." (lines 192-194)

d. Lines 240-248: Please clarify that this growth does not refer to colonization on root, but to quantification for growth on the sterile water and the media, which is not the same as rhizosphere colonization.

Answer: Referring to the reviewer's suggestion, the title of this section has been revised to "Characteristics of depleted strains responding to peanut root exudates".(line 293)

e. Line 249: Pantoea shows an increase as well. Since what the authors show is fold change, all bacteria display increase in biofilm formation in response to exudates compared to control treatment, not only Stenotrophomonas.

Answer: We appreciate your professional comment. Now the effects of root exudates from monocropped and rotation peanuts on depleted/enriched bacterial growth have been investigated. "In this regard, all depleted bacterial strains had less sensitive response to root exudates of monocropped peanut as compared to that from rotation peanut" has been added. (lines 298-300)

f. Line 292: The authors should increase the controls used in this experiment to be

able to conclude that this phenomenon is explained by this set of depleted strains alone. For instance, they could utilize also random strains to construct a SynCom or heat kill the mixture of bacteria used in the 7-member SynCom.

Answer: We appreciated your constructive comments. Referring to your suggestion, we have conducted additional experiments to construct SynComs composed of random depleted strains and the mixture of inactivated strains as well, and tested their suppression on pathogen growth. This was an important reason for taking such long time to prepare the revised manuscript. The corresponding results and experiment method have been appended in the revised manuscript. (lines 341-348).

g. Line 160: really strong statement that VOCs can affect infection, if some of the differential VOCs between rotation and monocropping are not tested against the pathogen, in a concentration dependent-manner.

Answer: Referring to your suggestion, we have conducted additional experiments to test inhibitory effect of differential VOCs on pathogen based on concentration dependent-manner. The corresponding results and experiment method have been appended in the revised manuscript. (lines 224-228).

h. Line 255: To increase the impact of this section, the authors should divide in taxa between enriched in monocropping and rotation and then highlight what properties they possess and if they could be relevant for disease suppression.

Answer: This sentence has been revised to “Except *Pantoea* and *Lysinibacillus*, all the other depleted strains inhibited the growth of *F. oxysporum*”.(lines 313-314)

i. Line 311: The authors could consider the relevance of evaluating the interactions between these 7 microbes? are they affecting growth of each other? or biofilm? please see experiments in <https://www.nature.com/articles/s41396-018-0093-1> for ways to assess SynCom behavior vs individual strains. Also composition of different SynComs tested should be shown in a Table.

Answer: We appreciated the reviewer for the constructive comments. Referring to your suggestion, we have conducted additional experiments to test the relevance of evaluating the interactions between these 7 microbes. The corresponding results and experiment method have been appended in the revised manuscript. (lines 317-320).

3. In several parts of the text there is relevant information missing or mislocated:

a. Line 799: Please indicate the figure where this is described.

Answer: “(Fig. 5G)” has been added.(line 771)

b. Lines 120-124: This information is not shown in figure 2C because that is about the cultivable fungi.

Answer: This information has been revised to “Fig. 2D”.

c. Line 125-128: Please indicate where this information is shown.

Answer: “Figure 2E” has been added.

d. Line 142: Please indicate if you refer to bulk or rhizosphere soil.

Answer: We deleted it.

e. Lines 143-145: Did the authors assess disease incidence in the other growth stages? That would help to know whether disease development was consistent with what was observed in the field. As it is right now, it reads incomplete, since we don't see if the difference at this stage can determine how disease develops.

Answer: In order to make the logic clearer, we have deleted this section.

f. Line 673: Indicate in the text how this information is also laid out in Fig 3C. Also, please include in the text information about the lowest and highest sequence similarity that you used to consider two isolates the same.

Answer: This has been revised to "To confirm the existence of these bacterial isolates in the peanut rhizosphere, phylogenetic analysis was performed based on the 16S rRNA sequence of the bacterial isolates and gene sequences of OTUs (same genus with isolates). Isolates with the V4 region matching OTUs with more than 97% identity were considered to be the same strain". (lines 651-655)

g. Line 701: The authors should indicate if the Hoagland remained sterile and whether plants were stressed after this long incubation without changing to fresh medium.

Answer: To avoid nutrient solution stress and contamination, the nutrient solution was changed every 7 days and applied to TSA agar medium to check for the presence of bacteria.

h. Line 150 and Figure 2D: Please clarify what the data refers to because in the figure it says it's part of the pot experiment but in the text it says it's in vitro.

Answer: Referring to the reviewer's comment, the caption has been revised to "Suppression of peanut rhizosphere community on fungal pathogen development".(line 530)

i. Lines 180-181: Please indicate to which growth stages you are referring to and include the results for each of the growth stages.

Answer: This has been revised to "Nonmetric multidimensional scaling (NMDS) based on the Bray-Curtis dissimilarity matrix revealed significant differences in bacterial community of monocropping and rotation at all development stages".(lines 180-182)

j. Line 315: Please clarify if this conidia production is measured in vitro or in pots. The current titles in Figure 5 are misleading.

Answer: This result was deleted to make the paper more readable.

4. The section about cultivable microbiome and rhizosphere microbiome is not

clearly written

Line 201-221: In this section what was sequenced with which technology needs to be clarified because it's hard to follow what is the cultivable microbiome and what not. The authors should more clearly explain the different experiments where Illumina Sequencing was used because it is confusing at the moment. If these strains were isolates (200 isolates according to line 659) why are they talking about more than 6000 OTUs? How many OTUs would be deriving from one isolate? If this is the case, it should be displayed. However, in Line 662 they mention full length 16s rRNA primers, much longer than what Illumina MiSeq can sequence, so please clarify which technology was used specifically for which section. Line 205 Please indicate the growth stage that this belongs to. Line 207 The number of bacteria shown in Fig 4 is much lower, how is this related to the 6832 OTUs mentioned at the beginning of the paragraph?

Answer: We appreciate the reviewer for these professional questions. This is the result of the cultivable microbiome. For more clarity, these have been revised to "Considering the difference in the suppression ability of the rhizosphere microbiome against *F. oxysporum* on the agar plates between monocropping and rotation, we first compared the characteristics of cultivable microorganisms by Illumina sequencing. The number of sequences of the cultivable microbiome ranged from 14804 to 22748. The sequences were clustered into 714 (reads is 0 was deleted) OTUs at 97% similarity", and "In order to obtain the depleted and enriched strains in monocropping rhizosphere, we isolated 173 bacterial strains from agar plates of rhizosphere cultures for 16S rRNA sequencing.".(lines 253-258, 269-271)

Minor comments

Introduction

1. Line 40 "activating nutrient transformation" is unclear, it would be more appropriate to say they can aid plant nutrition.

Answer: Many thanks for your suggestion. "activating nutrient transformation" has been revised to "enriched with carbon and nutrients".(line 48)

2. Line 48 With service function, do you mean ecosystem services?

Answer: Referring to the reviewer' comment. This has been revised to "Advancing our mechanistic understanding on how variations in the assembly of rhizosphere microbiomes influence plant disease is of paramount important to provide innovative strategies for improving plant health and productivity.".(line 54-57)

3. Line 53 What do you mean that the functions have been isolated from the rhizosphere microbiome?

Answer: In order to enhance the readability of the manuscript, this word has been omitted from the revised manuscript.

4. Line 58-62 These two phrases sound contradictory. To what do you refer with structural stability? And how is it that functional performance is expanded but "its

difficult to achieve the function of the entire microbiome”?

Answer: In order to enhance the readability of the manuscript, this word has been omitted from the revised manuscript.

5. Line 60 Why is reference 2, two times?

Answer: We deleted one of them.

6. Line 79 references <https://www.science.org/doi/full/10.1126/science.1203980>
<https://www.science.org/doi/full/10.1126/science.aaw9285> are appropriate for this statement.

Answer: Thanks for your suggestion, we refer to these papers.

7. Line 84-87 This phrase is unclear, if the taxa is stable, it would be expected to be less “sensitive” to environmental cues, sensitivity would lead to changes in its abundance.

Answer: Thanks for your comment. This has been revised to “As a result, different outcomes of plant health can be fostered depending on which microbial populations are able to take advantage of the root metabolites. Thus, agricultural practices could potentially exploit the differences in root metabolisms of various crops to disrupt the directional selection for maintaining the stability of agricultural ecosystem.”.(lines 75-80)

8. Lines 90-91 This is not a hypothesis, this is known for coumarins, benzoxazinoids, glucosinolates, triterpenes. Please see <https://www.pnas.org/doi/10.1073/pnas.1722335115> , <https://www.nature.com/articles/s41467-018-05122-7>, <https://www.nature.com/articles/ismej200968>, etc.

Answer: Thank you for your suggestion. This has been revised to “Recent studies have showed that rhizosphere microbiome assembly is significantly influenced by the selection effects of plant root metabolisms”.(line 74)

Methodology

1. Line 499-500: Please indicate the five-class scale in the text and fix the spelling of *Fusarium* in the citation.

Answer: Referring to the reviewer’s comment, the sentence has been revised to “For each examination, 30 plants from each plot were carefully removed from the soil, and peanut root rot was evaluated using a five-class rating scale (0 = no lesions, 1 = small root lesions, 2 = central root lesions, 3 = large root lesions, 4 = dead plant)”.(lines 407-410)

2. Line 508: Here the authors should already specify the pathogen they refer to (*Fusarium* (?)).

Answer: The sentence has been moved to the section of “Determination of pathogen abundance in peanut rhizosphere”, and “*Fusarium*” was specified. (lines 488)

3. Lines 515, 681 and 760: Please specify the year of these publications, and add the reference number as it is in the rest of the manuscript.

Answer: We added “Schuck et al. (2014)” in lines 456, “Robledo et al. (2012)” were deleted, “Li et al. (2019)” in lines 748.

4. Lines 526 to 529 and where this applies: please remove the space between the number and the Celsius symbol in temperatures.

Answer: Done.

5. Line 547: Please separate sample processing and DNA extraction from the bioinformatic analysis.

Answer: The title, “Sample collection and processing” has been added to separate bioinformatic analysis. (line 415).

6. Line 565 and other parts of methods: The authors should cite the references from the bioinformatic tools used e.g. QIIME.

Answer: We cited references to QIIME(in line 445) and other bioinformation tools (in lines 450, 451, etc).

7. Line 635: It is not clear what the septum is. Is there maybe something missing in between these two phrases?

Answer: “septum” has been revised to “dish lid”.(line 598)

Results

1. Line 150: Please refer to figure's numbers and subfigures in the text in a manner consistent with the actual Figures e.g., “2D” instead of “2d”

Answer: Done.

2. Line 151: Please make this sentence more clear to people lacking the background, for instance saying Fusarium inhibition bioassays.

Answer: This has been revised to “Antifungal activity against *F. oxysporum* were tested in VOC-mediated and directed microcosm antagonism assays to evaluate antifungal activity.”.(lines 204-205)

3. Line 172: Not clear from Figure 1 that sampling for microbiome analysis took place in 2018.

Answer: Yes, “Sampling time” has been added in Fig. 1.

4. Lines 261-264: Not clear what the contribution of this analysis is to the story.

Answer: We deleted this analysis.

Figures

Figure 1: Please indicate why the word “maturing” is surrounded by a red square only in the rotation crop image (surrounded by a blue square).

Answer: This figure has been revised.

Figure 2C and 5E: “Isolated strains” and “strains”.

Answer: The spelling has been corrected.

Extended Data

Figure 1: Please indicate if this is related to bacterial or fungal communities in the legend of the figure.

Answer: In order to make the article more readable, we have deleted it.

Table 3: Fix “Siderophore”.

Answer: Done.

Reviewer #3 (Remarks to the Author):

In this paper, the authors investigated the difference between peanut plants grown in rotation and monocropping. They performed a field experiment and monitored disease. They did not find any difference in disease in the field experiment. However, they still decided to investigate many different parameters to try to understand what is happening in peanut grown in rotation and mono cropping. They found that the rhizosphere bacterial communities from monocropping and rotation plots are different. They collected bacteria from the rhizosphere and characterised their collection in terms of phylogeny, biofilm production and response to exudates.

The authors performed an enormous amount of experiments and used many different techniques. They used Illumina sequencing to characterise the rhizosphere of several experiments, they isolated microbes, they analyzed VOC, they analyzed exudates and they even used metatranscriptomics! However, I have strong concerns about this manuscript. First of all, none of the experiments were repeated. While I understand this is difficult for some experiments like the field experiment or the isolation of bacteria, it is quite feasible for the pot experiments. Second, the number of replicate per treatment is 3 which is on the low side for this kind of study. Third, some of the side experiments (VOCs for example) are not described with enough detail to convince the reader that this experiment can actually answer the tested hypothesis.

Answer: We are appreciated comments on polishing our manuscript and make full sense of your concerns on our study. Totally, our shortcomings in the writing of the manuscript have led to your possible misunderstanding.

(1) In field experiments, we found no significant difference in disease index between monocropping and rotation at seedling stage, but significant difference from flowering stage to pod-bearing stage. By real-time detection of pathogen abundance in peanut rhizosphere, the of pathogen in monocropped rhizosphere at the seedling stage ware accumulated, implying that the failure of the rhizosphere to resist pathogen invasion take place early in the plant growth. Therefore, we dedicated to conduct multiple experiments via microbiome pipelines to understand the underlying mechanisms of rhizosphere community assembly to determine

disease outcome in rotation and monocropping regimes.

(2) For the low side of replicate in our experiments concerned by the reviewer, the lack of sufficient and accurate presentation of the experimental design and results may have led to the reviewer's misunderstanding of the number of replicates. For example, there were actually 60 experimental units in pot experiments, with 30 pots (biological replicates) for each cropping regime (3 plots). At harvest, 10 pots from each plot were homogenized as a composite sample. Detailed information were referred in lines 518-522 in our original manuscript. Consequently, reviewer 1 put forward pertinent comments on this part: "Line 587: I would say this is a biological replicate. This is confusing as the caption of Figure 2 is mentioned n=3".

It was the fact that we arranged 3 plots (replicates) for each cropping regime in our long-term field experiment (date from April 2012). For evaluating plant disease, we continuously examined the results across the entire growing season with independent three times. This procedure provided total 540 plants for collecting data. In fact, four times of independent microbial sampling further proved the adequacy of our field data based on our study aims.

Again, referring to the reviewer's comments, we have re-performed some of the original experiments with the increased replicates to make the results convincing, such as the effect of root exudates on depleted and enriched bacterial growth.

(3) For avoiding the unclear understanding of some experiments, we have detailed the purpose of certain experiments in the result section to deliver the reader what's hypothesis we tried to answer. Further, we conducted additional experiments to make the logic of the study clearer, such as the effect of volatile organic compounds on the growth of pathogen and the interaction between depleted strains. We hope that all concerns raised by the reviewer are now well addressed.

I find the introduction very difficult to read. Partly because of the English, which is definitely not good enough, but also because the references are not always well-chosen and the structure is not clear.

Lots of English mistakes, too many to list here. Please have the manuscript checked by native speaker.

Answer: We are sorry for not specifically explaining our present study in the introduction section of original manuscript. Based on your suggestions, we have re-organized the introduction. The language of the manuscript has been revised by a native speaker editor (Joanna Mackie, email: joanna.potrykus.mackie@gmail.com), and we further invited a microbiologist (Delgado-Baquerizo, who have published many papers in *Nature* and *Nature-sister journals*) with close research on the present study to polish our manuscript. We are very guilty for some typos in the original manuscript, and we have carefully checked the terms throughout the whole manuscript for avoiding other mistakes. For the arrangement of the figures, we have structured them referring to your comments to follow a logical progression.

When reading the results section, I was missing the connection between different

subsections. Why was this experiment performed? To test which hypothesis? How was it performed?

Answer: For each subsection, we added the reasons and steps for the test to be performed. Please see the beginning of each subsection for specific details.

Some experiment are not described correctly. For example, the abundance of Fusarium is quantified by sequencing (lines 129-135) but the results reported are in copies/g dry soil, which appears to be qPCR results.

Answer: We are sorry for not correctly experiment description. It actually appears to be qPCR results. We modified to “We conducted further quantitative real-time PCR (qPCR) analyses to gain deeper insights into the abundance and dynamics of *F. oxysporum* in the peanut rhizosphere at different growth stages.”.(lines 157-159)

The information about which community was sequencing with Illumina is missing. Is it the bacterial community? the fungal community? Also, the motivation for focusing on the bacterial community should be stated.

Answer: Yes, it was the bacterial community. " DNA extraction and sequencing from field experiment" and “Sample preparation, DNA extraction and sequencing of cultivable microbiome” have been added in methods.(lines 498 and 628). Due to the well-known role of rhizosphere bacterial communities in soil-borne disease control, we focused our research on bacterial communities.

The authors performed also some pot experiments. According to Figure 1, there were actually 10 pots, but in Figure 2D the number of replicates was 3. Especially for a pot experiment, it is possible to have more replicates.

Answer: In our pot experiments, 10 biological replicate pots for each plot, resulting in 30 biological replicates for monocropping and rotation regime respectively (10 pots × 3 field plots × 2 crop treatments), which had been described in method section (lines 522). We are sorry for the inconsistent texts across the manuscript. Referring to the reviewer’s comment, we have corrected these mistakes.

I did not read the discussion as I feel this paper is not yet matured enough.

Answer: We are sorry for not specifically explaining our present study in the results and methods section of original manuscript. Referring to you and the above two reviewers’ comments, we have carefully revised the manuscript for more clarity and readable.

Detailed comments

RESULTS (only section commentated with line edits)

line 109: rotation regime => with what crop? This is an important information which should be stated in the results section as well as the figure legend.

Answer: Information: “For crop rotation, peanut was grown first (2012), and then maize (*Zea mays* L.), potato (*Solanum tuberosum*), and soybean (*Glycine max*) were ordinally planted in every other peanut planting year.” has been added in note of Fig.

1. (lines 965-967)

line 113: 2.3-2.6 higher: please state the stage where this was found. Is this a confidence interval? a range? please be more precise.

Answer: This has been revised to “However, from flowering stage, the DI in peanuts under monocropping regime dramatically increased, and was 2.6 times higher than that of peanuts sown under the rotation regime at pod-bearing stage”. (lines 130-132)

line 115: "we isolated 38 fungi": how many strains did you obtain for each species? add the numbers in the barplot.

Answer: This has been revised to “In order to obtain cultures associated with the potential pathogens, we isolated 38 fungi from the diseased peanut roots. Based on growth morphology, 20 isolates were selected for 18S rRNA sequencing and subsequently were identified as *F. oxysporum* (5), *Penicillium sp. 196F* (1), *F. solani* (3), *Talaromyces pinophilus* (2), *Talaromyces verruculosus* (4), and *Neocosmospora striata* (5).”. (lines 145-150)

line 119: start a new paragraph with the results of the illumina sequencing as this is a new approach. Describe what you did in a few words. Please also report some results of the quality of the sequencing (as supplemental: how many sampled were sequenced, how many sequences per sample were obtained) as well as rarefaction plots.

Answer: We have started a new paragraph with the results of the illumina sequencing and “To identify the pathogen associated with this root rot disease, we characterized the community composition of fungi in healthy and diseased peanut roots using Illumina sequencing (Fig. 1B). A total of 347,214 internal transcribed spacer 1 (ITS1) reads were obtained (range, 31995–44592 reads per sample), clustering into 183 fungal OTUs at $\geq 97\%$ sequence identity.”. (lines 136-140)

line 122: "F. oxysporum associated with OTU177 " what do you mean with these words? I assume that you found sequences of OTU177 (taxonomically assigned *Fusarium oxysporum*) to be more abundant in the diseased-pea rot, please be more precise, the word "associated" means something else.

Answer: This has been revised to “Results indicated OTU177 (taxonomically assigned *F. oxysporum*) and OTU90 (taxonomically assigned *F. solani*) were significantly enriched in the diseased peanut root.”. (lines 142-144)

line 124: "34.1 and 2712.7 times more abundant" how are these numbers calculated?

Answer: This has been revised to “Results indicated OTU177 (taxonomically assigned *F. oxysporum*) and OTU90 (taxonomically assigned *F. solani*) were significantly enriched in the diseased peanut root. The relative abundance of these two fungal species was 34.1 and 2712.7 times higher in diseased than in healthy peanut roots.”.

(lines 142-145)

Figure 2 Panel C: the right half (OTU177 and OTU90) does not fit the left half which is labeled with a C and which shows the cultivable Fungi. It seems this panel with OTU177 and OTU90 needs its own letter. Also, what does the disease level 0, 2, and 4 correspond to?

Answer: Significantly enriched OTUs were shown in Figure 1D. “-1.5 to 1.5 represents the relative abundance of normalized OTU.” has been added. (line 982)

Line 125: again, start a new paragraph with the results of the pot experiment. Describe in a few words how the pot experiment was performed.

Answer: In order to add logic and readability to this manuscript, we have deleted this sentence.

The lines 125-128 do not correspond to any figure as far as I can tell. Also, the experiment is not really explained. What was the goal?

Answer: “Therefore, we identified our isolated strain (*F. oxysporum*) as the most likely organism behind the root rot disease observed in the studied peanut fields” in line 132 and figure about “Test for pathogenicity of the potential pathogens *Fusarium spp.*” have been added in Fig.2E. (line 983)

line 130: "specific primers" targeting which organisms?

Answer: This has been revised to “We conducted further quantitative real-time PCR (qPCR) analyses to gain deeper insights into the abundance and dynamics of *F. oxysporum* in the peanut rhizosphere at different growth stages.”.(lines 157-159)

line 130 talks about sequencing but the figure 2b seems to present results of qPCR

Answer: Yes, we are guilty for this mistake. The sentence has been revised to “We conducted further quantitative real-time PCR (qPCR) analyses to gain deeper insights into the abundance and dynamics of *F. oxysporum* in the peanut rhizosphere at different growth stages”. (lines 157-159)

line 136: why present result of ANOVA and then student's t test on line 138? In this case, since this treatment only has two levels, a t-test makes sense.

Answer: Thank you for your suggestion. Statistical analysis has been revised to be consistent.

line 141-146: how was this pot experiment performed? what does it mean to do a rotation pot experiment?

Answer: To make the experiment purpose clearer, more details have been added: “To examine the role of rhizosphere community assembly in the occurrence of peanut root rot, we further collected soils from field plots of both monocropping and rotation to grow peanut at a greenhouse experiment (Fig. 1C). The rhizosphere

microbiome of peanuts at seedlings was harvested to determine the ability of *F. oxysporum* inhibition". (lines 200-204)

Extended Data Table 2: what do the numbers in the table represent? Please provide T and P values instead of letters.

Answer: These results are presented in Extended Data fig. 2, with asterisks indicating significant differences between treatments.

Line 151: "Directed and inverted culture experiments": what does this mean

Answer: This has been revised to "Antifungal activity against *F. oxysporum* were tested in VOC-mediated and directed microcosm antagonism assays to evaluate antifungal activity.". (lines 204-206)

line 157: and Figure 2d: in the legend, it is written that the VOC are produced by cultural bacteria, which bacteria, how were they isolated, how many isolates, which medium?

Answer: In order to determine the effects of rhizosphere bacteria on the growth of pathogen, we first obtained rhizosphere bacterial suspensions of monocropping and rotation. Rhizosphere bacterial suspensions were then applied to TSA medium and their antagonism with pathogen was determined. Therefore, VOCs was produced by rhizosphere cultivable microbiome as a whole, not only by individual cultural bacteria.

Figure 3B: the title of that panel is "cultivable microbiome" which is misleading as the Figure presents results of Illumina sequencing.

Answer: This is indeed the Illumina sequencing result of the cultivable microbiome. To make it easier to understand, we now move it in Extended Data Fig. 5.

Line 172: Illumina sequencing: of the bacterial or of the fungal community? please state in a few words the approach that was chosen.

Answer: This has been revised to "We then investigated the impacts of crop managements on the bacterial microbiome of peanut rhizosphere at different plant growth stages and bulk soil in our 2018 field experiment.". (lines 177-179)

line 173: "a downward trend" referring to what? this sentence should refer to a specific figure or panel. Also write more clearly "alpha diversity" as the concept "diversity" includes also beta diversity (and even gamma!).

Answer: In order to enhance the logic and readability of the manuscript, we have deleted this sentence.

line 178-180: ordination cannot reveal significant differences. If you would like to test for differences, please use a statistical test, e.g. PERMANOVA as in the following sentences.

Answer: “ANOISM, $P < 0.05$ ” has been added (line 182).

line 186-188: "enriched in the mono cropped plant rhizosphere" compared to what? please state the other side of comparison.

Answer: We analyzed the differences in rhizosphere bacterial community composition between monocropping and rotation seedlings.(lines 182-196)

Extended Data Fig. 1

what is the number above the bar? is it a P-value?

please label the y-axis of each plot

Answer: In order to add logic and readability to the manuscript, this figure has been deleted.

lines 188-191: similar "recruited to the rhizosphere compared to what"

Answer: We analyzed the differences in rhizosphere bacterial community composition between monocropping and rotation seedlings. (lines 182-196)

Extended Data Fig. 2 legend should also explain in more detail what is the experimental factor tested here. STAMP is a program, please refer to the statistical analysis performed by the program. Are the P-values corrected for multiple testing? and if yes, with what method?

Answer: We analyzed the differences in rhizosphere bacterial community composition between monocropping and rotation seedlings. Differences in bacterial OTUs abundance were examined using DESeq2. P values were corrected using the method of FDR. (lines 182-196)

We are actually not so much interested in which bacteria are more abundant in rotation vs bulk soil and which bacteria are more abundant in monoculture vs bulk soil but more interested in which bacteria are more abundant in monoculture vs rotation.

Answer: Thank you very much for your constructive suggestions. We analyzed the differences in rhizosphere bacterial community composition between monocropping and rotation seedlings. The corresponding results have been added in the revised manuscript. (lines 176-196)

line 199 Culturable

line 201 cultivable

please chose one word and stick to it. Personally, I prefer cultivable.

Answer: “culturable” has been revised to “cultivable” throughout the whole manuscript.

line 201. start this new section with one sentence which describes why you decided to focus on cultivable bacteria, and how you performed this isolation approach.

Answer: Corresponding information has been added: “Considering the difference in

the inhibition ability of the rhizosphere microbiome against *F. oxysporum* on the agar plates between monocropping and rotation, we first compared the characteristics of the whole cultivable microbiome by Illumina sequencing.”. (lines 254-257)

also, how did you sequence with Illumina the cultivable microbiome?

Answer: The relevant information is described in the method. (lines 628-636)

line 202-203: it is strange that for this library you offer the range of sequences and the number of OTUs, information which you did not provide for the other libraries. Please be consistent, provide that information for all the sequencing approaches (for example in a supplemental table).

Answer: All sequencing results have been added to the results.

Extended Data Fig. 3 don not use box-plot with only three points. please provide labels for y-axis.

Answer: Done.

I am very surprised that the P-values are all > 0.05 especially for Shannon.

Answer: Done.

line 212: "we then determined OTUs of cultivable strains depleted" what does this mean?

Answer: This has been revised to “Since the cultivable microbiome sequence came from all bacterial colonies growing on agar plates, we then determined OTUs depleted from monocropping or rotation rhizosphere cultures.”. (lines 258-260)

line 222: what does "purified" mean?

Answer: "purified" has been revised to “isolated”. (lines 270)

lines 241-248: i am very confused about this experiment. The author mention exudates, but they report abundance of different bacteria. What is the hypothesis tested with this experiment? How was it performed? Similarly, how was the biofilm experiment performed?

Answer: Root exudates regulate microbial community assembly in plant rhizosphere, thus we hypothesized that monocropping depleted or enriched bacteria respond differently to peanut root exudates. Therefore, monocropping depleted and enriched bacterial strains were inoculated with root exudates of peanut from monocropping and rotation respectively to investigate the effects of root exudates on bacterial growth. This has been revised to “To understand whether the depleted of low abundance of taxa is associated with responsiveness to plant root exudates, we collected root exudates of peanut grown on monocropping and rotation soil, and evaluated effects of root exudates on bacterial growth *in vitro*.”. (lines 295-298)

line 260-261: Co-occurrence network analysis: which of the dataset was used to

construct this network? It is not interesting to just list that there were negative and positive correlations. this is obvious. I would just drop this figure entirely.

Answer: Thank you for your suggestion and this figure has been omitted from the revised manuscript.

line 265: "changes in cultivable microbiome function"

Answer: Done.

again, briefly state, what is the goal of this experiment? how was it performed?

Answer: "In order to provide a more detailed understanding on the mechanisms behind these results, we further performed transcriptome analysis of cultivable microbiome from monocropping and rotation rhizosphere to understand the associated functional variation for *F. oxysporum* inhibition (Fig. 1C). We extracted mRNA of cultivable microbiome from monocropping and rotation rhizosphere after the VOCs-mediated antagonism assay, and transcribed it into cDNA for sequencing. Pathways enriched in different Kyoto Encyclopedia of Genes and Genomes (KEGG) orthology functional categories in cultivable rhizosphere microbiomes from monocropping and rotation were analyzed." has been added. (lines 230-238)

lines 292-295: Good! this is exactly what I mean. Each new section should start with the question or hypothesis and then explain briefly how the experiment was performed.

Answer: Thank you for your suggestion and we have revised the manuscript according to your suggestion.

REVIEWER COMMENTS

Reviewer #3 (Remarks to the Author):

I find that this revised manuscript is much improved. In particular, I appreciate that each experiment is introduced with a sentence to explain the goal, this much improves the readability.

I like the new Figure 1.

Results from Figure 4.E and F and associated paragraphs are very confusing to me. Please explain what was done and what was the motivation. The interpretation of the results do not make sense to me.

Similarly, the interpretation of the results from Extended Data Fig. 7 is very confusing. On this figure, we see that the bacteria do not affect each other growth. So there is no evidence for a growth inhibition. But I do not agree with the conclusion that the the depletion of the strains if intensified.

Personally, I am not a big fan of the merged results and discussion section. Is that a requirement of Nature Communications? I prefer when the two sections are separate. In the current state of the manuscript, each result is related to one or two references but a more general discussion is missing.

There are still some places where the English could be improved.

For example:

lines 52-54: maybe split the sentence in two to improve readability?

In general, I would avoid using "we" in the abstract and introduction and keep the personal pronoun for the Material and Methods and Results sections.

Specific comments

lines 154-155. I see on Fig. 2E that *F. oysporum* causes more disease than *F. solani*, but I do not agree with the conclusion of line 154-155. The disease could also be caused by a complex of microorganisms including other the isolated *Fusarium* strains.

line 167-170: I do not see how this sentence relates to the results described in Fig. 2F. What do you mean with "early in the plant growth"? You did not observe any difference at the seedling stage.

Line 175-176: why suddenly investigating the bacterial microbiome? this needs to be justified since you have been focusing on fungi for the first part of the study.

Line 181: I believe the test is called ANOSIM (<https://rdrr.io/rforge/vegan/man/anosim.html>)

Line 181: please provide P value if larger than 0.001

Lines 189-191: there is a problem with this sentence which is missing a verb. Also not sure what is the added value of this sentence compared to the sentence on lines 185-187.

Line 209: please provide P value if larger than 0.001

line 208-211: this sentence sounds a bit strange, not sure what the authors mean with the word "entanglement"

lines 220-221 what do you mean with "were not determined"? do you mean "not detected"?

lines 287-288: I do not understand this statement. What is the evidence from your experiment

supporting this statement? please provide figure number.

Fig. 4E: what do the colors stand for? How is the relative abundance calculated?

Line 300: please provide P value if larger than 0.001

lines 298-300: again, I not understand this sentence. I see on the Fig. 4F that all the microbes which have a red bar in Figure E grew less when treated with root exudates from rotation (and the reviewse is tree for Burkholderia and Ptreotrophomas). So these microbes are sensitive to something present in the two different types of exudates. But I do not understand the connection of this result to the conclusion sentences at the end of the paragraph.

Line 336: ANOVA statement is redundant with ANOVA statement presented on line 333.

Lines 344-246: I do not see and inactivated SynCom on that Figure 5D.

Line 359: please provide P value if larger than 0.001

line 361: Is this really a field experiment? This design is quite bad as all the 5 controls at the top left and field experiments are usually performed with a randomised block design. A caveat about this design needs to be added to the manuscript.

Figure 3 legend: according to the material and methods, deseq2 is used to identify differentially abundant OTUs but in the figure legend, a Wilcoxon test is mentioned.

Figure 4 legend: what does "TPM" stand for? What about "Rich Factor"? Please provide more information in the figure legend. Also please state what the bars and standard deviation stand for? I assume mean and standard deviation, but this needs to be stated. N=30 is provided for Figure A and B, what about C, D, E, and F?

Figure 5 legend: define "RhiCom"

Fig. 5A: "s" is missing at the end of the word "combination" since there are 35 or 21, it is the plural form!

Reference 9 (Feussner et al.), is missing the journal, issue number, pages and date

[Editor's note: as Reviewer #1 and #2 were not available this time, Reviewer 3 agreed to examine the responses to their main points]

REVIEWER 1

regarding the question about reference databases, I believe the authors have answered this question adequately.

Same for differential abundance analysis, although the figure legend of Fig. 3B mentions a Wilcoxon's test, the material and methods mention deseq2.

Regarding the question about the exudate assay (Figure 4A from the original manuscript), I find the new Fig 4E and lines 295-308 very very confusing and do not agree with the interpretation of the results. This whole section needs more work.

Regarding the question about the RNAseq, I believe the new methods section describes better the procedure.

Regarding the question about BLAST, I believe the Extended Data Fig. 1 provides the required information.

Figure 2c from the original manuscript is now Figure 2D and the figure legend has been updated as requested by the reviewer.

"Lines 139-141: this is not a valid interpretation as no rhizosphere data support it".
In my opinion the new sentence is still not a valid interpretation.

"Lines 151-156: I could not understand this experiment. " This was Fig. 2D in the original manuscript. I am not sure where results of these experiments are now shown, maybe extended data table 2? I don't understand if the authors answered the reviewer's concerns.

"Line 188: the differential abundance analysis was performed with a t-test"
I agree with the authors that `deseq2` is a valid approach.

"Line 716: Metatranscriptome. "
I believe the authors answered that question in the text.

REVIEWER 2

line 292 "heat kill the mixture of bacteria used in the 7-member SynCom."
The new experiment is presented in Figure 5A, but I do not see the heat-killed control suggested by the reviewer.

Line 160: "really strong statement that VOCs can affect infection"

The authors present new results in Extended Data Fig. 4 which are quite convincing.

"Line 311: The authors could consider the relevance of evaluating the interactions between these 7 microbes? "

The authors present a new experiment in Fig. 5A which looks good to me.

Reviewer #4 (Remarks to the Author):

This is well revised manuscript! Interesting and novel findings regarding assembly of rhizosphere microbiome in monocropping and rotation and its link to disease suppressiveness.

My suggestion is to move the first part of the results and discussion (ln97-109) to the end as it reads more like concluding remark.

As the results and discussion are combined in one part- I missed the discussion of this manuscript. I would suggest to elaborate the discussion.

Response to reviewers' comments

Reviewer #3 (Remarks to the Author):

I find that this revised manuscript is much improved. In particular, I appreciate that each experiment is introduced with a sentence to explain the goal, this much improves the readability.

I like the new Figure 1.

Answer: We appreciate your positive feedback on our study and have carefully considered your comments and concerns for refining our manuscript. We are confident that we have addressed all the concerns raised by the reviewer, as outlined below.

Results from Figure 4.E and F and associated paragraphs are very confusing to me. Please explain what was done and what was the motivation. The interpretation of the results do not make sense to me.

Answer: We apologize for any confusion regarding the explanation of the similarity comparison between isolated strains and depleted or enriched OTUs. To enhance clarity, we have included an additional description of the motivation: "When the sequence similarity between isolated strains and depleted or enriched OTUs exceeds 97%, the strain is considered representative of the corresponding OTU.". (lines 258-260).

Similarly, the interpretation of the results from Extended Data Fig. 7 is very confusing. On this figure, we see that the bacteria do not affect each other growth. So there is no evidence for a growth inhibition. But I do not agree with the conclusion that the depletion of the strains if intensified.

Answer: Many thanks for your helpful comment. The conclusion has been revised to "No significant enhancement or inhibition was observed between any two strains (Extended Data Fig. 7), indicating independent growth and stable coexistence among depleted strains". (lines 292-295)

Personally, I am not a big fan of the merged results and discussion section. Is that a requirement of Nature Communications? I prefer when the two sections are separate. In the current state of the manuscript, each result is related to one or two references but a more general discussion is missing.

Answer: Thanks for your comment. Referring to your suggestion, results and discussion have been separated, and the discussion section was expanded

correspondingly as well.

There are still some places where the English could be improved.

For example:

lines 52-54: maybe split the sentence in two to improve readability?

Answer: Thanks for your comment. This has been revised to “Some soils have a larger capacity to support disease suppression than others helping to prevent the establishment of pathogens in the rhizosphere of plants. However, the characteristic defining these soils are largely unknown”. (lines 55-58)

In general, I would avoid using "we" in the abstract and introduction and keep the personal pronoun for the Material and Methods and Results sections.

Answer: Thanks for your suggestion. We have avoided using "we" in the abstract and introduction.

Specific comments

lines 154-155. I see on Fig. 2E that *F. oxysporum* causes more disease than *F. solani*, but I do not agree with the conclusion of line 154-155. The disease could also be caused by a complex of microorganisms including other the isolated *Fusarium* strains.

Answer: Firstly, through Illumina sequencing of healthy and diseased roots, we found that OTU177 (taxonomically assigned to *F. oxysporum*) and OTU90 (taxonomically assigned to *F. solani*) were significantly enriched in diseased peanut root. However, other isolated *Fusarium* strains did not significantly enrich in the diseased peanut roots. Based on your comment, the sentence has been revised to “Since other *Fusarium* were not enriched and cultivable, we identified our isolated strain (*F. oxysporum*) as the most likely organism behind the root rot disease observed in the studied peanut fields”. (lines 145-147)

line 167-170: I do not see how this sentence relates to the results described in Fig. 2F. What do you mean with "early in the plant growth"? You did not observe any difference at the seedling stage.

Answer: We are sorry for not clearly explaining. Now it has been omitted.

Line 175-176: why suddenly investigating the bacterial microbiome? this needs to be justified since you have been focusing on fungi for the first part of the study.

Answer: One sentence has been added: “The bacterial microbiome plays a crucial role in influencing the severity of soil-borne diseases compared to fungi. Plant roots can effectively recruit a higher diversity and richness of the rhizosphere bacterial

community to suppress the invasion of pathogens, particularly those originating from fungal sources¹⁶⁻¹⁸". (lines 164-167)

Line 181: I believe the test is called ANOSIM (<https://rdr.io/rforge/vegan/man/anosim.html>)

Answer: We are sorry for the mistake. This has been corrected. (line 173)

Line 181: please provide P value if larger than 0.001

Answer: Done as suggested. (line 173)

Lines 189-191: there is a problem with this sentence which is missing a verb. Also not sure what is the added value of this sentence compared to the sentence on lines 185-187.

Answer: We are sorry for the mistake. Here, we found that crop management had an important impact on low abundance OTUs in the rhizosphere of peanuts. This sentence has been revised to "Further, the average relative abundance of 122 depleted OTUs (account for 82.4% of the discriminating OTUs), and 148 enriched OTUs (account for 83.1% of the discriminating OTUs) in monocropping was less than 0.1%.". (lines 177-180)

Line 209: please provide P value if larger than 0.001

Answer: Done as suggested. (line 194)

line 208-211: this sentence sounds a bit strange, not sure what the authors mean with the word "entanglement"

Answer: The word has been revised to "participation". (line 420)

lines 220-221 what do you mean with "were not determined"? do you mean "not detected"?

Answer: Yes, this has been revised to "not detected". (line 203)

lines 287-288: I do not understand this statement. What is the evidence from your experiment supporting this statement? please provide figure number.

Answer: We are sorry for unclear description. Sequence similarity between isolated strains and depleted/enriched OUTs was compared to determine the consistency of the two strains. Among them, sequence similarity between 9 OTUs (7 monocropping depleted OTUs, OTU9456, OTU8675, OTU9041, OTU9439, OTU8770, OTU9294 and OTU10342; 2 monocropping enriched OTUs, OTU6966 and OTU12223) and isolates

are greater than 97%. Then, the bar chart in Fig.4E shows the relative abundance of these OTUs. Results indicated that the relative abundance of matched OTUs are relatively low. “(Fig. 4E)” has been added. (line 269)

Fig. 4E: what do the colors stand for? How is the relative abundance calculated?

Answer: “Blue and red represent monocropping enriched and depleted OTUs in culturable microbiome, respectively.” has been added in lines 1091-1092. Relative abundance was calculated by dividing reads per OTU by the total reads per treatment in the culturable microbiome.

Line 300: please provide P value if larger than 0.001

Answer: Done as suggested. (line 278)

lines 298-300: again, I not understand this sentence. I see on the Fig. 4F that all the microbes which have a red bar in Figure E grew less when treated with root exudates from rotation (and the reviewse is tree for Burkholderia and Ptrenotrophomas). So these microbes are sensitive to something present in the two different types of exudates. But I do not understand the connection of this result to the conclusion sentences at the end of the paragraph.

Answer: Blue and red represent monocropping enriched and depleted OTUs in Fig. 4E, respectively. Then, in Fig. 4F, we tested the response of these representative OTUs to root exudations from monocropping (blue) and rotation (red) peanuts. This has been revised to “Compared to the enriched strains, the growth of depleted strains increased less in responding to root exudates from monocropping peanut.”. (lines 281-282)

Line 336: ANOVA statement is redundant with ANOVA statement presented on line 333.

Answer: It has been omitted.

Lines 344-246: I do not see and inactivated SynCom on that Figure 5D.

Answer: In the Figure 5D, 0 represent the inactivated SynCom which was indicated in lines 1117.

Line 359: please provide P value if larger than 0.001

Answer: Done as suggested. (line 328)

line 361: Is this really a field experiment? This design is quite bad as all the 5 controls

at the top left and field experiments are usually performed with a randomised block design. A caveat about this design needs to be added to the manuscript.

Answer: We are sorry for not correctly describing field experiment. A randomized block design was used in the field experiment. Figure 5G represents the design model of the four treatments rather than the actual field layout. This has been revised to “Design of four experimental treatments (does not represent the plots distribution)”. (lines 1110-1111)

Figure 3 legend: according to the material and methods, deseq2 is used to identify differentially abundant OTUs but in the figure legend, a Wilcoxon test is mentioned.

Answer: We are sorry for error description. “Wilcoxon's test” has been revised to “DESeq2” in Fig.3B legend. (line 1075)

Figure 4 legend: what does "TPM" stand for? What about "Rich Factor"? Please provide more information in the figure legend. Also please state what the bars and standard deviation stand for? I assume mean and standard deviation, but this needs to be stated. N=30 is provided for Figure A and B, what about C, D, E, and F?

Answer: “Rich Factor, the ratio of the number of differential genes in the metabolic pathway to the number of all genes annotated to the pathway. (n=3)”, “TPM, transcripts Per million. (n=3)”, and “Each bars represents the mean \pm SD.” has been added. (lines 1084-1196)

Figure 5 legend: define "RhiCom"

Answer: “RhiCom, treatment with monocropping rhizosphere community” has been added. (line 1115)

Fig. 5A: "s" is missing at the end of the word "combination" since there are 35 or 21, it is the plural form!

Answer: Thanks. This has been corrected.

Reference 9 (Feussner et al.), is missing the journal, issue number, pages and date

Answer: We are very sorry for the error. “*PNAS*. **115**, E5213-E5222 (2018).”has been added. (lines 917)

[Editor's note: as Reviewer #1 and #2 were not available this time, Reviewer 3 agreed to examine the responses to their main points]

REVIEWER 1

regarding the question about reference databases, I believe the authors have answered this question adequately.

Answer: Many thanks.

Same for differential abundance analysis, although the figure legend of Fig. 3B mentions a Wilcoxon's test, the material and methods mention *deseq2*.

Answer: We are sorry for error description. "Wilcoxon's test" has been revised to "DESeq2" in Fig.3B legend.

Regarding the question about the exudate assay (Figure 4A from the original manuscript), I find the new Fig 4E and lines 295-308 very confusing and do not agree with the interpretation of the results. This whole section needs more work.

Answer: We are sorry for unclear description. It has been revised to "Compared to the enriched strains, the growth of depleted strains increased less in responding to root exudates from monocropping peanut.". (lines 281-282)

Regarding the question about the RNAseq, I believe the new methods section describes better the procedure.

Answer: Thanks.

Regarding the question about BLAST, I believe the Extended Data Fig. 1 provides the required information.

Answer: Thanks.

Figure 2c from the original manuscript is now Figure 2D and the figure legend has been updated as requested by the reviewer.

Answer: Thanks.

"Lines 139-141: this is not a valid interpretation as no rhizosphere data support it".

In my opinion the new sentence is still not a valid interpretation.

Answer: Thanks for your comment. The sentence has been revised to "Several taxa from *Fusarium* sp. have been reported with ability to cause wilting symptoms in peanuts, but specific pathogen lying behind peanut root rots remains to be discovered." (Lines 130-132)

"Lines 151-156: I could not understand this experiment. " This was Fig. 2D in the original manuscript. I am not sure where results of these experiments are now shown, maybe extended data table 2? I don't understand if the authors answered the

reviewer's concerns.

Answer: These experimental results are now shown in Figures 4A and 4B, and the relevant description is shown in lines 186-195.

"Line 188: the differential abundance analysis was performed with a t-test"

I agree with the authors that deseq2 is a valid approach.

Answer: Thanks.

"Line 716: Metatranscriptome. "

I believe the authors answered that question in the text.

Answer: Thanks.

REVIEWER 2

line 292 "heat kill the mixture of bacteria used in the 7-member SynCom."

The new experiment is presented in Figure 5A, but I do not see the heat-killed control suggested by the reviewer.

Answer: We are sorry for misunderstood of reviewer 2's meaning and only added inactivated control in Fig. 5C and 5E. Now, we supplemented the experiment with synthetic community inactivation. Related results have been added in Fig. 5B.

Line 160: "really strong statement that VOCs can affect infection"

The authors present new results in Extended Data Fig. 4 which are quite convincing.

Answer: Thanks.

"Line 311: The authors could consider the relevance of evaluating the interactions between these 7 microbes? "

The authors present a new experiment in Fig. 5A which looks good to me.

Answer: Thanks.

Reviewer #4 (Remarks to the Author):

This is well revised manuscript! Interesting and novel findings regarding assembly of rhizosphere microbiome in monocropping and rotation and its link to disease suppressiveness.

My suggestion is to move the first part of the results and discussion (ln97-109) to the end as it reads more like concluding remark.

As the results and discussion are combined in one part- I missed the discussion of this manuscript. I would suggest to elaborate the discussion.

Answer: We appreciate your compliments on our study. The first part of the results

and discussion in the original manuscript has been moved to the end. Results and discussion have been separated, and the discussion section has been expanded as well in the revised manuscript.